# MIN-MAX MULTI-OBJECTIVE BILEVEL OPTIMIZATION WITH APPLICATIONS IN ROBUST MACHINE LEARNING

**Alex Gu**[†]**, Songtao Lu**[‡]**, Parikshit Ram**[‡]**, Tsui-Wei Weng**[*]
[†]MIT CSAIL, [‡]IBM Research, [*]UCSD
[†]gua@mit.edu, [‡]{songtao, parikshit.ram}@ibm.com, [*]lweng@ucsd.edu

## ABSTRACT

We consider a generic min-max multi-objective bilevel optimization problem with applications in robust machine learning such as representation learning and hyperparameter optimization. We design `MORBiT`, a novel single-loop gradient descent-ascent bilevel optimization algorithm, to solve the generic problem and present a novel analysis showing that `MORBiT` converges to the first-order stationary point at a rate of $\widetilde{\mathcal{O}}(n^{1/2}K^{-2/5})$ for a class of weakly convex problems with $n$ objectives upon $K$ iterations of the algorithm. Our analysis utilizes novel results to handle the non-smooth min-max multi-objective setup and to obtain a sublinear dependence in the number of objectives $n$. Experimental results on robust representation learning and robust hyperparameter optimization showcase (i) the advantages of considering the min-max multi-objective setup, and (ii) convergence properties of the proposed `MORBiT`. Our code is at https://github.com/minimario/MORBiT.

## 1 INTRODUCTION

We begin by examining the classic bilevel optimization (BLO) problem as follows:

$$\min_{x\in\mathcal{X}\subseteq\mathbb{R}^{d_x}} f(x, y^\star(x)) \quad \text{subject to} \quad y^\star(x) \in \arg\min_{y\in\mathcal{Y}=\mathbb{R}^{d_y}} g(x, y) \tag{1}$$

where $f : \mathcal{X} \times \mathcal{Y} \to \mathbb{R}$ is the upper-level (UL) objective function and $g : \mathcal{X} \times \mathcal{Y} \to \mathbb{R}$ is the lower-level (LL) objective function. $\mathcal{X}$ and $\mathcal{Y}$, respectively, denote the domains for the UL and LL optimization variables $x$ and $y$, incorporating any respective constraints. Equation 1 is called BLO because the UL objective $f$ depends on both $x$ and the solution $y^\star(x)$ of the LL objective $g$. BLO is well-studied in the optimization literature (Bard, 2013; Dempe, 2002). Recently, *stochastic* BLO has found various applications in machine learning (Liu et al., 2021; Chen et al., 2022a), such as hyperparameter optimization (Franceschi et al., 2018), reinforcement learning or RL (Hong et al., 2020), multi-task representation learning (Arora et al., 2020), model compression (Zhang et al., 2022), adversarial attack generation (Zhao et al., 2022) and invariant risk minimization (Zhang et al., 2023).

In this work, we focus on a *robust* generalization of equation 1 to the multi-objective setting, where there are $n$ different objective function pairs $(f_i, g_i)$. Let $[n] \triangleq \{1, 2, \cdots, n\}$ and $f_i : \mathcal{X} \times \mathcal{Y}_i \to \mathbb{R}$, $g_i : \mathcal{X} \times \mathcal{Y}_i \to \mathbb{R}$ denote the $i^{\text{th}}$ UL and LL objectives respectively. We study the following problem:

$$\min_{x\in\mathcal{X}\subseteq\mathbb{R}^{d_x}} \max_{i\in[n]} f_i(x, y_i^\star(x)) \quad \text{subject to} \quad y_i^\star(x) \in \arg\min_{y_i\in\mathcal{Y}_i=\mathbb{R}^{d_{y_i}}} g_i(x, y_i), \forall i \in [n]. \tag{2}$$

Here, the optimization variable $x$ is shared across all objectives $f_i, g_i, i \in [n]$, while the variables $y_i, i \in [n]$ are only involved in their corresponding objectives $f_i, g_i$. The goal is to find a *robust solution* $x \in \mathcal{X}$, such that, the worst-case across all objectives is minimized. This is a generic problem which reduces to equation 1 if we have a single objective pair, that is $n = 1$. Such a robust optimization problem is useful in various applications, and especially necessary in any safety-critical ones. For example, in decision optimization, the different objectives $(f_i, g_i)$ can correspond to different "scenarios" (such as plans for different scenarios), with $x$ being the shared decision variable and $y_i$'s being scenario-specific decision variables. The goal of equation 2 is to find the robust shared decision $x$ which provides robust performance across all the $n$ considered scenarios, so that such a robust assignment of decision variables will generalize well on other scenarios. In machine

learning, robust representation learning is important in object recognition and facial recognition where we desire robust worst-case performance across different groups of objects or different population demographics. In RL applications with multiple agents (Busoniu et al., 2006; Li et al., 2019; Gronauer & Diepold, 2022), our robust formulation in equation 2 would generate a shared model of the world – the UL variable $x$ – such that the worst-case utility, $\max_i f_i(x, y_i^\star(x))$, of the agent-specific optimal action – the LL variable $y_i^\star(x)$ – is optimized, ensuring robust performance across all agents.

An additional technical advantage of the general multi-objective problem in equation 2 is that it allows the objective-specific variables $y_i \in \mathcal{Y}_i$ to come from different domains, that is, $\mathcal{Y}_i \neq \mathcal{Y}_j, i, j \in [n]$; as stated in equation 2, this implies that the dimensionality $d_{y_i}$ for the per-objective $y_i$ need not be the same across all objectives. This allows for a larger class of problems where each objective can then have different number of objective specific variables but we still require a robust shared variable $x$. For example, in multi-agent RL, different agents can have different action spaces because they need to operate in different mediums (land, water, air, etc).

Focusing on stochastic objectives common in ML, the main contributions of this work are as follows:

▶ (New algorithm design) We present a single loop **M**ulti-**O**bjective **R**obust **Bi**level **T**wo-timescale optimization algorithm, MORBiT, which uses (i) SGD for the unconstrained strongly convex LL problem, and (ii) projected SGD for the *constrained weakly convex* UL problem.

▶ (Theoretical convergence guarantees) We demonstrate that, under standard smoothness and regularity conditions, MORBiT with $n$ objectives converges to a $\widetilde{\mathcal{O}}(n^{1/2}K^{-5/2})$-stationary point with $K$ iterations, matching the best convergence rate for single-loop single-objective ($n = 1$) BLO algorithms with the constrained UL problem while using vanilla SGD for the LL problem, and providing a *sublinear $n^{1/2}$-dependence* on the number of objective pairs $n$.

▶ (Two sets of applications) We present two applications involving min-max multi-objective bilevel problems, robust representation learning and robust hyperparameter optimization (HPO), and demonstrate the effectiveness of our proposed algorithm MORBiT.

**Paper Outline.** In the following section 2, we further discuss the different aspects of the problem in equation 2 and compare that to the problems and solutions considered in existing literature. We present our novel algorithm, MORBiT, and analyse its convergence properties in section 3, and empirically evaluate it in section 4. We conclude with future directions in section 5.

## 2 PROBLEM AND RELATED WORK

We first discuss the different aspects of the robust multi-objective BLO problem with constrained UL in equation 2. While BLO is used in machine learning (Liu et al., 2021; Chen et al., 2022a), **multi-objective** BLO has not received much attention. In multi-task learning (MTL), the optimization problem is a multi-objective problem in nature, but is usually solved by summing the objectives and using a single-objective solver, that is, optimizing the objective $\sum_i f_i$. The robust **min-max** extension of MTL (Mehta et al., 2012; Collins et al., 2020) and RL (Li et al., 2019) have been shown to improve generalization performance, supporting the need for a more complex multi-objective optimization problem that replaces the objective $\sum_i f_i$ with the objective $\max_i f_i$.

For SGD-based solutions to stochastic BLO, one critical aspect is whether the algorithm is **single-loop** (a single update for both $x$ and $y$ in each iteration) or **double-loop** (multiple updates for the LL $y$ between each update of the UL $x$). Double-loop algorithms can have faster empirical convergence, but are more computationally intensive, and their performance is extremely sensitive to the step-sizes and termination criterion for the LL updates. Double-loop algorithms are not applicable when the (stochastic) gradients of the LL and UL problems are only provided sequentially, such as in logistics, motion planning and RL problems. Hence, we develop and analyse a single-loop algorithm.

A final aspect of BLO is the *constrained UL problem*. When the UL variable $x$ corresponds to some decision variable in a decision optimization problem or a hyperparameter in HPO, we must consider a constrained form, $x \in \mathcal{X} \subset \mathbb{R}^{d_x}$. To capture a more general form of the bilevel problem, we focus on the constrained UL setup. In the remainder of this section, we will review existing literature on single-objective and multi-objective BLO and robust optimization, especially in the context of machine learning. Table 1 provides a snapshot of the properties of the problems and algorithms (with rigorous convergence analysis) studied in recent machine learning literature.

Table 1: The problem studied here relative to representative related work. If the studied problem is not a BLO, the notion of single-loop or constrained UL ($\mathcal{X} \subset \mathbb{R}^{d_x}$) is not applicable. The 1$^{\text{st}}$ row block lists general problems. The 2$^{\text{nd}}$ block lists algorithms with analyses. The final row is MORBiT. †: This problem has been viewed both as single-level and bilevel. □: The problem can be multi-objective but is treated as single-objective by summing the objectives. △: In bilevel adversarial learning, the UL is unconstrained but the LL is constrained.

| Problem/Method | Bilevel | Multi-objective | Min-max | Single-loop | $\mathcal{X} \subset \mathbb{R}^{d_x}$ |
|---|---|---|---|---|---|
| Distributionally Robust Learning | † | ✗ | ✓ | - | - |
| Adversarially Robust Learning | † | ✗ | ✓ | - | △ |
| Multi-task Learning (MTL) | † | □ | ✗ | - | - |
| Robust MTL (Mehta et al., 2012) | † | ✓ | ✓ | - | - |
| Meta-learning | † | □ | ✗ | - | - |
| BSA (Ghadimi & Wang, 2018) | ✓ | ✗ | ✗ | ✗ | ✓ |
| HiBSA (Lu et al., 2020) | ✗ | ✗ | ✓ | ✓ | ✓ |
| GDA (Lin et al., 2020) | ✗ | ✗ | ✓ | ✓ | ✗ |
| TR-MAML (Collins et al., 2020) | ✗ | ✓ | ✓ | ✓ | ✓ |
| TTSA (Hong et al., 2020) | ✓ | ✗ | ✗ | ✓ | ✓ |
| StocBio (Ji et al., 2021) | ✓ | ✗ | ✗ | ✗ | ✗ |
| MRBO (Yang et al., 2021) | ✓ | ✗ | ✗ | ✓ | ✗ |
| VRBO (Yang et al., 2021) | ✓ | ✗ | ✗ | ✗ | ✗ |
| ALSET (Chen et al., 2021) | ✓ | ✗ | ✗ | ✓ | ✗ |
| STABLE (Chen et al., 2022b) | ✓ | ✗ | ✗ | ✓ | ✓ |
| MMB (Hu et al., 2022) | ✓ | ✗ | ✓ | ✓ | ✗ |
| MORBiT (Ours) | ✓ | ✓ | ✓ | ✓ | ✓ |

**Single-Objective BLO.** Lately, many new algorithms have been proposed to solve the single-objective stochastic BLO problem in equation 1. Ghadimi & Wang (2018) proposed the first double-loop BSA approach. StocBio (Ji et al., 2021) and VRBO (Yang et al., 2021) are double-loop schemes that improve upon the convergence rate of BSA but do not consider constrained UL problems. TTSA (Hong et al., 2020) is a single-loop algorithm that handles UL constraints. MRBO (Yang et al., 2021) and ALSET (Chen et al., 2021) are single-loop algorithms improving TTSA's convergence rate but do not consider UL constraints. STABLE (Chen et al., 2022b) improves upon TTSA by leveraging an additive *correction term* in the LL update step (beyond a basic SGD step) while still handling UL constraints. In contrast to the above single-objective bilevel setup, our formulation in equation 2 gives flexibility for inherently multi-objective problems in a robust manner to obtain stronger guarantees, ensuring convergence of each individual objective, rather than the average objective.

**Multi-Objective BLO.** There has been a limited number of works analyzing multi-objective BLO schemes (Sinha et al., 2015; Deb & Sinha, 2009; Ji et al., 2017). All of these works analyze the multi-objective BLO problem from a game-theoretic point of view, using a vector-valued objective with the notion of Pareto optimality. In contrast, we are the first to study the multi-objective BLO problem from a traditional optimization perspective in terms of convergence properties and consider a min-max robust version of the multi-objective problem which produces *a single solution that ensures the convergence of each individual objective* instead of generating multiple Pareto-optimal solutions which trade-off the optimality of the different objectives. See further discussion in Appendix D.4.

**Min-max Robust Optimization in Machine Learning.** Min-max optimization is commonly used to achieve robustness, such as in distributionally robust learning (DRL) and adversarially robust learning (ARL). In DRL, Duchi & Namkoong (2018) and Shalev-Shwartz & Wexler (2016) showed that a min-max loss improves generalization due to variance regularization. HiBSA (Lu et al., 2020) and GDA (Lin et al., 2020) compute quasi-Nash equilibria with convergence guarantees. Robust optimization is shown to have strong generalization for new tasks in multi-task learning (Mehta et al., 2012) and meta-learning (Collins et al., 2020). While the classic MAML (Finn et al., 2017) can be formulated as a BLO problem (Rajeswaran et al., 2019), the precise problem analysed in Collins et al. (2020) is a single-level one. In fact, we consider the bilevel form of the TR-MAML problem as one of our applications for empirical evaluation. In ARL, the minimum is over the loss and the maximum is over the worst-case perturbation to inputs (Madry et al., 2017; Wang et al., 2019). In both DRL and ARL, the min-max objective is in the form $\min_x \max_y f(x, y)$ with a single-objective. In contrast, we study general robust multi-objective BLO where the UL objective is dependent on the LL solutions, and where the minimization is over the variable $x$ shared across all objectives, and the maximization is over the multiple objectives, ensuring that each individual objective converges fast.

**Closely related and concurrent work.** Since our goals align with the properties of TTSA (Hong et al., 2020) – the single-loop nature and the ability to handle UL constraints – our proposed MORBiT is inspired by TTSA and can be viewed as a robust multi-objective version. Beyond this advancement,

our contribution also lies in the convergence analysis of MORBiT, which significantly diverges from that of TTSA. After our MORBiT was developed and released (Gu et al., 2021), STABLE (Chen et al., 2022b) was recently presented as an improvement of TTSA, and we wish to explore similar improvements to MORBiT in future work. A very recent work (Hu et al., 2022) studies a problem that appears to be quite similar to equation 2, with common elements such as bilevel and min-max, and proposes a single-loop multi-block min-max bilevel (MMB) algorithm. However, there are significant differences: (i) Firstly, in their setup, they consider an extension of a min-max single level problem to a min-max BLO, and min-max is **not** meant to provide "robustness" among objectives. The applications in Hu et al. (2022) are restricted to problems such as multi-task AUC maximization instead of the common bilevel applications of representation learning and HPO. (ii) Also, Hu et al. (2022) do not consider a constrained UL problem. The problem in equation 2 is *not* a generalization of their problem – both our work and theirs are considering different setups with high-level commonalities. For more details, see Appendix D.2. Table 1 shows how our setup compares to existing literature. To the best of our knowledge, *the precise problem in equation 2 has not been studied in ML literature*.

## 3  ALGORITHM AND ANALYSIS

In this section, we propose a simple single-loop algorithm MORBiT to solve equation 2, and establish a rigorous convergence rate and sample complexity for this algorithm. For the theoretical results, we defer the precise assumptions, statements and proofs to Appendix A and present the high-level theoretical results and critical novel proof steps here. In the sequel, we will always use the subscript $i \in [n]$ to denote the objective index and the superscript $(k)$ to denote the iteration index, with $x^{(k)}$ denoting the $k^{\text{th}}$ iterate of the shared variable $x \in \mathcal{X} \subseteq \mathbb{R}^{d_x}$ and $y_i^{(k)}$ denoting the $k^{\text{th}}$ iterate of the $i^{\text{th}}$-objective-specific variable $y_i \in \mathbb{R}^{d_{y_i}}$. We will also use the shorthand $\boldsymbol{y}$ to denote all the per-objective variables $[y_1, y_2, \ldots, y_n]$, with $\boldsymbol{y}^{(k)}$ denoting the $k^{\text{th}}$ iterate of all the per-objective variables $[y_1^{(k)}, y_2^{(k)}, \ldots, y_n^{(k)}]$. Given our assumption that the LL objectives $g_i$ are strongly convex, we define $y_i^\star(x) \triangleq \arg\min_{y_i \in \mathbb{R}^{d_{y_i}}} g_i(x, y)$, and use the shorthand $\ell_i(x) \triangleq f_i(x, y_i^\star(x))$.

### 3.1  MORBiT ALGORITHM

We begin with a standard reformulation of robust min-max problems (Duchi et al., 2008). We can rewrite the non-smooth min-max problem in equation 2 as

$$\min_{x \in \mathcal{X}} \max_{\lambda \in \Delta_n} \sum_{i \in [n]} \lambda_i f_i(x, y_i^\star(x)) \quad \text{subject to} \quad y_i^\star(x) = \arg\min_{y_i \in \mathbb{R}^{d_{y_i}}} g_i(x, y_i), \, \forall i \in [n] \quad (3)$$

where $\Delta_n \in \mathbb{R}_+^n$ is the $n$-simplex defined as $\Delta_n \coloneqq \{\lambda \in \mathbb{R}_+^n : \lambda_i \geq 0, \forall i \in [n], \sum_{i \in [n]} \lambda_i = 1\}$. This problem is equivalent to the min-max problem in equation 2, but allows us to solve the problem with (projected) gradient based methods. The gradient for $y_i, i \in [n]$ is the straightforward $\nabla_{y_i} g_i(x, y_i)$ and we denote $h_i$ as its stochastic estimate, with $\boldsymbol{h}$ as the shorthand for the per-objective stochastic gradient estimates $[h_1, h_2, \ldots, h_n]$. The gradient for the $x$-update is more involved because of the hierarchical structure of the BLO problem. Then, we consider the following weighted objectives utilizing the simplex variable $\lambda \in \Delta_n$ to define the necessary gradients:

$$F(x, \lambda) = \sum_{i \in [n]} \lambda_i \ell_i(x), \quad F(x, y, \lambda) = \sum_{i \in [n]} \lambda_i f_i(x, y_i). \quad (4)$$

Note that $F(x, \lambda)$ is the UL objective in equation 3, and the UL gradients can be defined as:

$$\nabla_x F(x, \lambda) = \sum_{i \in [n]} \lambda_i \nabla \ell_i(x), \quad \nabla_\lambda F(x, \lambda) = [\ell_1(x), \cdots, \ell_n(x)]^\top, \quad (5)$$

where $\nabla_x \ell_i(x)$ for any $i \in [n]$ can be defined as follows utilizing the strong convexity of the LL problem and implicit gradients (Gould et al., 2016):

$$\nabla \ell_i(x) = \nabla_x f_i(x, y_i^\star(x)) - \nabla_{xy_i}^2 g_i(x, y_i^\star(x)) \left[\nabla_{y_i y_i}^2 g_i(x, y_i^\star(x))\right]^{-1} \nabla_{y_i} f_i(x, y_i^\star(x)). \quad (6)$$

Note that in general, $y_i^\star(x)$ cannot be computed exactly. Following Ghadimi & Wang (2018), we use an approximation of $\nabla_x \ell_i(x)$ as a surrogate, denoted by $\overline{\nabla}_x f_i(x, y_i)$, by replacing $y_i^\star(x)$ in equation 6 with any $y_i \in \mathbb{R}^{d_{y_i}}$ as follows:

$$\overline{\nabla}_x f_i(x, y_i) = \nabla_x f_i(x, y_i) - \nabla_{xy_i}^2 g_i(x, y_i) \left[\nabla_{y_i y_i}^2 g_i(x, y_i)\right]^{-1} \nabla_{y_i} f_i(x, y_i). \quad (7)$$

Consequently, we define our approximate gradients for the UL variables $x$ (and $\lambda$) as:

$$\overline{\nabla}_x F(x, \boldsymbol{y}, \lambda) = \sum_{i \in [n]} \lambda_i \overline{\nabla}_x f_i(x, y_i), \quad \overline{\nabla}_\lambda F(x, \boldsymbol{y}, \lambda) = [f_1(x, y_i), \cdots, f_n(x, y_n)]^\top. \quad (8)$$

We denote the (possibly biased) stochastic estimates of $\overline{\nabla}_x F(x, \boldsymbol{y}, \lambda)$ as $h_x$ and $\overline{\nabla}_\lambda F(x, \boldsymbol{y}, \lambda)$ as $h_\lambda$.

---

**Algorithm 1:** MORBiT with learning rates $\alpha$, $\beta$ and $\gamma$ for $x, \boldsymbol{y}, \lambda$ respectively

---

1 **for** $k = 1, 2, \cdots, K$ **do**
2     $\boldsymbol{y}^{(k+1)} \leftarrow \boldsymbol{y}^{(k)} - \beta \boldsymbol{h}^{(k)}$
3     $x^{(k+1)} \leftarrow \text{proj}_{\mathcal{X}}(x^{(k)} - \alpha h_x^{(k)})$
4     $\lambda^{(k+1)} \leftarrow \text{proj}_{\Delta_n}(\lambda^{(k)} + \gamma h_\lambda^{(k)})$
5 **end**
6 Sample $\tau \sim \mathcal{U}(\{1, \cdots, K\})$
7 **return** $\bar{x} \leftarrow x^{(\tau)}, \bar{y}_i \leftarrow y_i^{(\tau-1)}, \bar{\lambda} \leftarrow \lambda^{(\tau)}$

---

Given the gradients and their stochastic estimates, we present our single-loop algorithm MORBiT in algorithm 1, where we utilize learning rates $\alpha, \beta, \gamma > 0$ for the UL variable $x$, LL variables $y_i, i \in [n]$ and the simplex variable $\lambda$ respectively. The algorithm tracks three sets of variables $x^{(k)}, \boldsymbol{y}^{(k)} = [y_1^{(k)}, y_2^{(k)}, \ldots, y_n^{(k)}]$ and $\lambda^{(k)}$ through a total of $K$ iterations. The per-iterate gradient estimates $\boldsymbol{h}^{(k)}$ of the LL variables $\boldsymbol{y}$ is defined as the collection of the per-objective gradient estimate $h_i^{(k)}$ evaluated at $(x^{(k)}, y_i^{(k)})$ for all $i \in [n]$. The gradient estimates $h_x^{(k)}$ and $h_\lambda^{(k)}$ of the UL variables $x$ and $\lambda$ are evaluated at $(x^{(k)}, \boldsymbol{y}^{(k+1)}, \lambda^{(k)})$. We perform a standard gradient descent update for the objective specific variables $y_i, i \in [n]$ from $y_i^{(k)}$ to $y_i^{(k+1)}$. For the shared UL variable $x$, we perform a **projected gradient descent** to satisfy the UL constraints, where $\text{proj}_{\mathcal{X}}(\cdot)$ denotes the projection operation onto the constrained set $\mathcal{X}$. We update the simplex variable $\lambda$ via **projected gradient ascent**, where we project the variable back onto the $n$-simplex after a gradient ascent step with $\text{proj}_{\Delta_n}(\cdot)$. Given the learning rates $(\alpha, \beta, \gamma)$, MORBiT is quite straightforward in terms of implementation. When $n = 1$, the problem reduces to single-objective BLO, $\lambda^{(k)} = 1$, and MORBiT reduces to TTSA (Hong et al., 2020).

## 3.2 ANALYSIS

Given the single-loop MORBiT, we establish conditions under which MORBiT has finite-horizon convergence. The coupling of the stochastic errors due to the sampling process makes the convergence analysis of this three-sequence-based algorithm much more challenging than existing BLO algorithms.

**Assumptions.** We summarize the following typical assumptions (detailed in Appendix A.1) for all objective pairs $(f_i, g_i), i \in [n]$. Focusing on the smoothness and regularity properties of the objectives, we assume that (i) the LL objective $g_i$ is strongly convex in $y_i$, twice-differentiable, and has sufficiently smooth first and second order gradients (Assumption 2 in Appendix A.1), (ii) the UL objective $f_i$ has sufficiently smooth first order gradients, and (iii) the function $\ell_i(x) \triangleq f_i(x, y_i^\star(x))$ is weakly convex, bounded and has bounded first-order gradients (Assumption 1 in Appendix A.1, also see Appendix D.1). Regarding the quality of the gradient estimates $h_i^{(k)}, i \in [n]$, $h_x^{(k)}$ and $h_\lambda^{(k)}$, we assume that, for all $k > 0$, (i) $h_i^{(k)}$ is an unbiased estimate with bounded variance, (ii) $h_\lambda^{(k)}$ is an unbiased estimate, and (iii) $h_x^{(k)}$ has bounded variance, and can be a biased estimate of the $\overline{\nabla}_x F(x^{(k)}, \boldsymbol{y}^{(k+1)}, \lambda^{(k)})$ term defined in equation 8, but the bias norm at iteration $k$ is bounded by $b_k \geq 0$, with $\{b_k, k \geq 0\}$ forming a non-increasing sequence. These gradient estimate quality assumptions are detailed in Assumption 3 in Appendix A.1. While the assumptions on $h_i^{(k)}$ and $h_\lambda^{(k)}$ are standard (Hong et al., 2020; Lu et al., 2022), the assumption on $h_x^{(k)}$ actually can be easily satisfied when a Hessian inverse approximation (HIA) based mini-batch sampling strategy is adopted, which can also avoid the matrix inversion by leveraging the Neumann series (Agarwal et al., 2017; Ghadimi & Wang, 2018; Hong et al., 2020).

**Optimality and Stationarity of Solutions.** To quantify the convergence properties of the solutions $\bar{x}, \bar{y}_i, i \in [n], \bar{\lambda}$ generated by MORBiT, we use the following optimality properties of the optimal solutions $x^\star, y_i^\star, i \in [n], \lambda^\star$ of the problem in equation 3. (i) The per-objective optimal LL variable $y_i^\star = y_i^\star(x^\star) = \arg\min_{y_i \in \mathbb{R}^{d_{y_i}}} g_i(x^\star, y_i)$; (ii) The optimal simplex variable $\lambda^\star$: $F(x^\star, \lambda^\star) = \max_{\lambda \in \Delta_n} F(x^\star, \lambda)$. Given the constrained UL, the first-order stationarity condition is satisfied if $\langle \nabla_x F(x^\star, \lambda^\star), x - x^\star \rangle \geq 0 \, \forall x \in \mathcal{X}$. (iii) For establishing near-stationarity of UL variable $x$, the **proximal map** $\hat{x}(z) \in \mathcal{X}$, defined below,

$$\hat{x}(z) \triangleq \arg\min_{x \in \mathcal{X}} \frac{\rho}{2} \|x - z\|^2 + F(x, \lambda), \quad \rho > 0 \text{ is a fixed constant}. \quad (9)$$

is employed (Davis & Drusvyatskiy, 2018; Hong et al., 2020) to quantify the convergence for a constrained variable $x$ in the stochastic setting. If $\|\hat{x}(x^{(k)}) - x^{(k)}\|^2$ is small, then, near-stationarity of $x^{(k)}$ is achieved at iteration $k$. Therefore, we need to bound $\|\hat{x}(\bar{x}) - \bar{x}\|$ to guarantee the convergence of the UL solution $\bar{x}$ returned by MORBiT. Given the convergence of the UL $\bar{x}$, we also need to bound $\|\bar{y}_i - y_i^\star(\bar{x})\|^2$ *for each* $i \in [n]$ *simultaneously* to quantify the convergence of the LL variables. Finally, the convergence of $\bar{\lambda}$ requires us to bound the difference between $F(\bar{x}, \bar{\lambda})$ and $\max_{\lambda \in \Delta_n} F(\bar{x}, \lambda)$.

**Theoretical Convergence Rate.** Now, we are ready to state our main theoretical result: a rigorous convergence rate for the solution returned by MORBiT (algorithm 1). We state an abbreviated version of the result, deferring details to Theorem 2 in Appendix A.2:

**Theorem 1** (MORBiT convergence). *Suppose that the previously stated assumptions holds and learning rates are set as $\alpha = \mathcal{O}(K^{-3/5})$, $\beta = \mathcal{O}(K^{-2/5})$ and $\gamma = \mathcal{O}(n^{-1/2}K^{-3/5})$. Then, if $b_k^2 \leq \alpha$, the solutions $\bar{x}, \bar{y}_i, i \in [n], \bar{\lambda}$ generated by algorithm 1 satisfy:*

$$\mathbb{E}[\|\hat{x}(\bar{x}) - \bar{x}\|^2] \leq \widetilde{\mathcal{O}}(\sqrt{n}K^{-2/5}), \tag{10a}$$

$$\mathbb{E}\left[\max_{i \in [n]} \|\bar{y}_i - y_i^\star(\bar{x})\|^2\right] \leq \widetilde{\mathcal{O}}(\sqrt{n}K^{-2/5}), \tag{10b}$$

$$\max_\lambda \mathbb{E}[F(\bar{x}, \lambda)] - \mathbb{E}[F(\bar{x}, \bar{\lambda})] \leq \widetilde{\mathcal{O}}(\sqrt{n}K^{-2/5}), \tag{10c}$$

*with expectation over the stochastic gradient estimates and the random index $\tau$ (algorithm 1, line 6).*

This result establishes the $\widetilde{\mathcal{O}}(n^{1/2}K^{-2/5})$-stationarity achieved by $K$ iterations of MORBiT for both the UL and LL variables if all the assumptions are satisfied and the learning rates are selected appropriately. Note that, if the UL problem is unconstrained, that is $x \in X = \mathbb{R}^{d_x}$, the definition of the proximal map (equation 9) implies that $\mathbb{E}\|\nabla_x F(\bar{x}, \bar{\lambda})\| \leq \widetilde{\mathcal{O}}(n^{1/2}K^{-2/5})$, providing the convergence of $\bar{x}$ to a $\widetilde{\mathcal{O}}(n^{1/2}K^{-2/5})$-stationary point if the UL problem is unconstrained.

**Comparison with Related Work.** We would like to further highlight the differences between the convergence results of TTSA and MORBiT to highlight the major novelties in our analyses and theorem proving techniques. First, we consider a more general proximal map in equation 9 involving *a weighted sum of weakly convex functions $\ell_i$ instead of a single weakly convex function* in TTSA, requiring new construction of potential functions for establishing the convergence of the UL variable $\bar{x}$ in equation 10a. Secondly, even though TTSA provides a convergence rate for a single LL variable (equivalent to bounding $\mathbb{E}[\|\bar{y}_i - y_i^\star(\bar{x})\|^2]$ for a single $i \in [n]$), we provide a much stronger result for multiple LL optimization objectives, in the sense that *simultaneously establishing convergence for all* LL *variables in equation 10b* through measuring the convergence rate of $\mathbb{E}[\max_{i \in [n]} \|\bar{y}_i - y_i^\star(\bar{x})\|^2]$. This is especially challenging since a bounded $\mathbb{E}[\|\bar{y}_i - y_i^\star(\bar{x})\|]$ for each $i \in [n]$ does not directly imply a bounded $\mathbb{E}[\max_{i \in [n]} \|\bar{y}_i - y_i^\star(\bar{x})\|^2]$; in fact this can be generally unbounded. Finally, to satisfy the requirements of the min-max problem in equation 3, we have to additionally establish convergence for the simplex solution $\bar{\lambda}$ in equation 10c while TTSA does not have any such analysis.

Given the convergence rate, another related quantity of interest is the **sample complexity** which pertains to the number of queries to the stochastic gradient oracle required to achieve a desired level of stationarity. For example, for an iterative algorithm that converges to a $\mathcal{O}(K^{-\mu})$-stationary point with $K$ iterations for some $\mu > 0$, requiring $\mathcal{O}(1)$ queries to the stochastic gradient oracle in each iteration, the sample complexity to find an $\epsilon$-optimal solution is $\mathcal{O}(\epsilon^{-1/\mu})$. The number of stochastic gradient oracle queries required is directly related to the conditions in the gradient estimate quality assumptions (Assumption 3 in Appendix A.1 in our case). While the conditions on the per-iterate gradient estimates $h_i^{(k)}$ (for the per-objective LL variables) and $h_\lambda^{(k)}$ (for the simplex variable $\lambda$) both only require $\mathcal{O}(1)$ stochastic gradient oracle queries from each of the $n$ objective pairs in each iteration, the condition $b_k^2 \leq \alpha$ on the non-increasing squared norm of the per-iterate bias of the gradient estimate $h_x^{(k)}$ (for the UL variable) require $\mathcal{O}(\log K)$ stochastic gradient oracle queries for each of the $n$ objective pairs leveraging the HIA sampling techniques in Ghadimi & Wang (2018) and Hong et al. (2020) using the Neumann series (Agarwal et al., 2017). This gives us the following sample complexity bound for MORBiT (see Appendix D.3 on potential improvements):

**Corollary 1.** *Under the conditions of Theorem 1, MORBiT converges to $\epsilon$-(near)-stationarity with $\mathcal{O}(n^{5/4}\epsilon^{-5/2} \log(1/\epsilon))$ queries to the stochastic gradient oracle for each of the $n$ objective pairs.*

**Proof Sketch of Theorem 1.** We now give a proof sketch of our main theorem, with constant terms abstracted away with $\mathcal{O}$ notation. In order to show equation 10a of Theorem 1 (convergence of $x$), we will derive a descent lemma comparing successive iterates $x^{(k)}$ and $x^{(k+1)}$. Descent lemmas often contain a quadratic term $\|x^{(k+1)} - x^{(k)}\|^2$, so it is natural that we must bound $\|h_x^{(k)}\|^2$. In Lemma 1, we bound the expected squared norm of the stochastic gradient estimate:

**Lemma 1.** *Under our regularity assumptions,* $\mathbb{E}[\|h_x^{(k)}\|^2] \leq \mathcal{O}\left( \sum_{i \in [n]} \lambda_i^{(k)} \|y_i^\star(x^{(k)}) - y_i^{(k+1)}\|^2 \right).$

Turning to equation 10b of Theorem 1, we use a descent relation on $y_i^{(k)} - y_i^\star(x^{(k-1)})$. While ideally we would obtain a descent relation purely involving $y_i^{(k)}$ terms themselves, the intricate coupling of the $x$ and $y_i$ terms result in an extra $x^{(k-1)} - x^{(k)}$ term. The resulting relation is shown in Lemma 2. Here, we have that $c_1 = 1 - \frac{\mu_g \beta}{2}, c_2 = \frac{2}{\mu_g \beta} - 1$, where $\mu_g \beta < 1$.

**Lemma 2.** $\mathbb{E}[\|y_i^{(k+1)} - y_i^\star(x^{(k)})\|^2] \leq \mathcal{O}\left( (1 - c_1)\mathbb{E}[\|y_i^{(k)} - y_i^\star(x^{(k-1)})\|^2] + c_2 \mathbb{E}[\|x^{(k-1)} - x^{(k)}\|^2] \right).$

From this lemma, intuitively, we know that $\mathbb{E}[\|y_i^{(k)} - y_i^\star(x^{(k-1)})\|^2]$ is decreasing as $k$ increases, *as long as the* $\mathbb{E}[\|x^{(k-1)} - x^{(k)}\|^2]$*'s are not too large*. Therefore, it is important to have another descent relation that upper bounds this quantity, which we do next in Lemma 3. The lemma naturally involves the objective $F(x^{(k)}, \lambda^{(k)})$, which will telescope. Here, we have $c_3 = \frac{1}{4\alpha} - \frac{L_f}{2}, c_4 = 4\alpha L^2$. As $k \to \infty$, $\alpha \to 0$, and $c_3$ is positive.

**Lemma 3.** *Let* $\mathcal{L}^{(k)} \triangleq \mathbb{E}[F(x^{(k)}, \lambda^{(k)})] = \sum_{i \in [n]} \lambda_i^{(k)} \mathbb{E}[\ell_i(x^{(k)})]$. *Then, the* $\mathcal{L}^{(k)}$ *satisfies:*

$$\mathcal{L}^{(k+1)} - \mathcal{L}^{(k)} \leq \mathcal{O}\left( -c_3 \mathbb{E}[\|x^{(k+1)} - x^{(k)}\|^2] + c_4 \max_{i \in [n]} \mathbb{E}[\|y_i^{(k+1)} - y_i^\star(x^{(k)})\|^2] + \sqrt{n}\gamma + \alpha \right).$$

Following the intuition previously described, we then use Lemmas 2 and 3 to show that the $\mathbb{E}[\|x^{(k-1)} - x^{(k)}\|^2]$ terms are small enough and that the $y_i^{(k)}$ iterates converge in Lemma 4.

**Lemma 4.** $\frac{1}{K} \sum_{k=1}^{K} \max_{i \in [n]} \mathbb{E}[\|y_i^{(k)} - y_i^\star(x^{(k-1)})\|^2] \leq \mathcal{O}(\sqrt{n}K^{-2/5}).$

Finally, we can use the convergence of $y_i$ to prove convergence of $\lambda$ and $x$. Theorem 1 then follows. A more detailed proof plan can be found in Appendix A.3.

## 4 EXPERIMENTAL RESULTS

In this section, we consider two applications where the min-max multi-objective bilevel formulation in equation 2 enhances robustness – multi-task representation learning and hyperparameter optimization. We will highlight the advantage of the min-max formulation and the convergence of MORBiT on these applications. We use PyTorch (Paszke et al., 2019), and implementation details are in Appendix C. All results are aggregated over 10 trials.

**Representation Learning.** In this setup, each objective pair corresponds to a learning "task" $i \in [n]$, with its own training and validation dataset pair $D_i^{\mathrm{t}}, D_i^{\mathrm{v}}$. We consider a shared representation network $\phi_x$ with ReLU nonlinearity (making the UL problem weakly convex) parameterized with $x \in \mathbb{R}^{d_x}$ and a per-task linear model $w_{y_i}$ parameterized with $y_i \in \mathbb{R}^{d_{y_i}}$. Here the UL is unconstrained. Using $L(f, D)$ to denote the loss of a model $f$ on data $D$, we consider the problem in equation 2 with

$$f_i(x, y_i) \triangleq L(w_{y_i} \circ \phi_x, D_i^{\mathrm{v}}), \quad g_i(x, y_i) \triangleq L(w_{y_i} \circ \phi_x, D_i^{\mathrm{t}}) + \rho \|w_{y_i}\|_2^2, \tag{11}$$

with $\rho > 0$ as a regularization penalty (ensuring that the LL problem is strongly convex). We first consider a multi-task setup with $n = 10$ binary classification tasks from the FashionMNIST dataset (Xiao et al., 2017). The goal is to learn a shared representation and per-task models so that each of the tasks generalizes well. Usually, this problem is solved as a single-objective BLO by minimizing $1/n \sum_i f_i$; we call this **min-avg**. We theoretically show that solving the min-max multi-objective BLO in equation 2 provides a *tighter* generalization guarantee (Proposition 2, Appendix B).

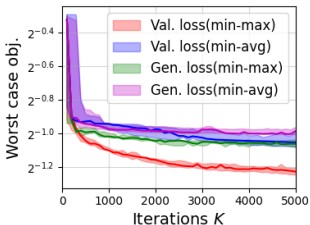 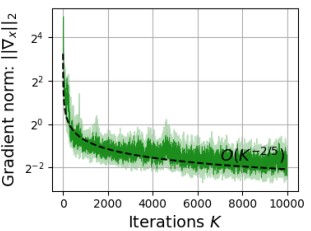 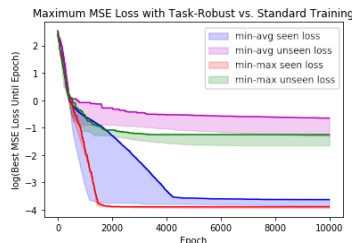

(a) Quality of min-avg vs min-max.  (b) Convergence of $\|\nabla_x\|^2$.  (c) Task-gen. of min-avg vs min-max.

Figure 1: Numerical results for representation learning application.

Here we demonstrate the same in figure 1a – we plot the worst-case `UL` objective (the validation loss) and the worst-case generalization loss across all tasks/objectives throughout the optimization trajectory, comparing the behaviour of the solution of min-avg problem to that of the min-max problem. The results indicate **solving the min-max problem** significantly reduces the worst-case validation loss and this also **results in a significant reduction of the worst-case generalization loss**, highlighting the utility of solving the min-max multi-objective bilevel problem in equation 2.

We study the convergence of the `UL` variable for the min-max problem in the form of the trajectory of the (stochastic) gradient norm $\|\nabla_x\|^2$ in figure 1b, comparing it to the theoretical $\widetilde{\mathcal{O}}(K^{-2/5})$ rate. We see that **the empirical trajectory of the gradient norm closely tracks the theoretical rate**.

We also consider a bilevel extension of the robust meta-learning application (Collins et al., 2020) for a sinusoid regression task, a common meta-learning application introduced by Finn et al. (2017)[1]. Here the goal of solving the problem in equation 2 with the objectives defined in equation 11 would be to learn a robust representation network such that we not only improve generalization on tasks seen during the optimization but also improve generalization for related unseen tasks.

We theoretically show that that solving the min-max multi-objective bilevel problem in equation 2 also provides a tighter generalization guarantee for the unseen tasks (Proposition 3, Appendix B). Note that these results are similar in spirit to those of Mehta et al. (2012) and Collins et al. (2020), but our results are the first for a general bilevel setup. The results in figure 1c support this, showing that solving the min-max problem not only improves the generalization on seen tasks, but significantly improves the generalization on unseen tasks when compared to solving the min-avg problem. These results are also consistent with the results for robust MTL in figure 1a.

**Hyperparameter Optimization.** In this setup, each objective pair again corresponds to a learning "task" $i \in [n]$, each with its own $d$ dimensional training/validation dataset pair $D_i^{\mathrm{t}}, D_i^{\mathrm{v}}$. We consider a shared hyperparameter optimization problem for kernel logistic regression (Zhu & Hastie, 2001) with $K$ random Fourier features (RFFs) (Rahimi & Recht, 2007), where $x = \{x_\rho \in \mathbb{R}_+^{2K}, x_\sigma \in \mathbb{R}_+^d\}$ are the regularization penalty and the bandwidth hyperparameters respectively, with $\phi_{x_\sigma}$ denoting the RFF[2]. The per-task linear model $w_{y_i}$ on top of the RFFs are parameterized with $y_i \in \mathbb{R}^{2K}$. In this setup, we have a weakly convex *constrained* `UL` *problem* (the hyperparameters need to be positive), and an unconstrained strongly convex `LL` problem. Again using $L(f, D)$ to denote the learning loss of a model $f$ on a dataset $D$, we consider the problem in equation 2 with

$$f_i(x, y_i) \triangleq L(w_{y_i} \circ \phi_{x_\sigma}, D_i^{\mathrm{v}}), \quad g_i(x, y_i) \triangleq L(w_{y_i} \circ \phi_{x_\sigma}, D_i^{\mathrm{t}}) + \|x_\rho \odot w_{y_i}\|_2^2, \quad (12)$$

where $\odot$ denotes the elementwise vector multiplication, and we consider a weighted regression penalty[3]. We generate $n = 16$ binary classification tasks from the Letter dataset (Frey & Slate, 1991) and compare the generalization of the min-max solution of equation 2 to that of the min-avg.

The results in figure 2a indicate that the solution of equation 2 provides a robust solution $x$ (hyper-parameters), significantly improving not only the worst-case validation loss but also the worst-case generalization loss for the supervised learning problems. This result highlights the advantage of

---

[1]We use the formulation of Raghu et al. (2020) to separate the representation and the model parameters.

[2]For a $d$-dimension point $p$, the RFF $\phi_{x_\sigma}(p) = [\sin(W(x_\sigma \odot p))^\top, \cos(W(x_\sigma \odot p))^\top]^\top \in \mathbb{R}^{2K}$, where $W \in \mathbb{R}^{K \times d}$ is a random normal matrix and the $\sin(\cdot)$ and $\cos(\cdot)$ are applied elementwise.

[3]The weighted regression penalty mitigates bias especially in the high-dimensional learning setting (Candes et al., 2008; Gasso et al., 2009; Šehić et al., 2022), which is common when using RFFs.

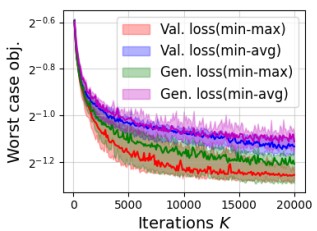 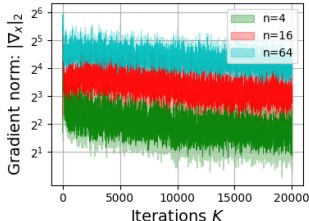 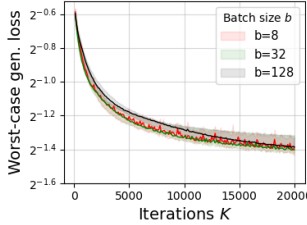

(a) Quality of min-avg vs min-max.     (b) Effect of $n$ on convergence.     (c) Effect of batch size.

Figure 2: Numerical results for hyperparameter optimization application.

solving the min-max problem in equation 2 and the ability of the single-loop `MORBiT` to handle a weakly convex constrained `UL` problem.

We study the effect of the number of objective pairs $n$ on the convergence. We consider $n \in \{4, 16, 64\}$, increasing $n$ with a factor of 4 (implying a theoretical convergence slow down by a factor of 2) to check how the convergence matches the $\sqrt{n}$-dependence in our theoretical result.

In this case, we consider the trajectory of the (stochastic) gradient norm $\|\nabla_x\|^2$ (as in figure 1b). The results in figure 2b display such a behaviour – for a fixed $K$ (outer iterations), as the number of tasks is increased 4-fold, the gradient norm approximately increases 2-fold (note the $\log_2$-scale on the vertical axis). This validates our theoretical dependence on the number of objective pairs $n$.

We also study the effect of the batch size on the generalization performance of the min-max solution. In the previous experiments, we considered a batch size of $8$ for both the `UL` and `LL` stochastic gradients. Here, we will consider batch sizes from $\{8, 32, 128\}$, using the same batch size for gradients of both levels and variables.

Note that, in this problem, each of the 16 learning tasks (and hence, objective pairs) has a training set size of around 900 samples (for the `LL` loss), with 300 samples each for the `UL` loss and for computing the generalization loss. Unlike figures 1a and 2a, we only show the generalization loss (dropping the validation loss) in figure 2c. The results indicate that increasing the batch size improves the stability and reduces the variance of the overall generalization. However, the convergence follows a similar trend for all batch sizes, and converges to a very similar level of generalization, supporting the $\widetilde{\mathcal{O}}(1)$ batch size requirement for convergence.

> **Empirical conclusion**
>
> The empirical evaluations highlight that considering the more robust min-max problem in equation 2 does provide improved generalization in multiple applications (representation learning for MTL and meta-learning, and for hyperparameter optimization).
>
> - - - - - - - - - - - - - - - - - - - - - - - - - - - - - - - - - - - - - - - - - - - -
>
> The results also highlight the validity of our theoretical convergence analysis both in terms of the number of iterations $K$ and the number of objective pairs $n$.

## 5 CONCLUDING REMARKS

Motivated by the desiderata of robustness in bilevel learning applications, we study a new min-max multi-objective BLO framework (equation 2) that provides full flexibility and generality. We propose `MORBiT` (algorithm 1), a single-loop gradient descent-ascent based algorithm for finding an solution to our proposed min-max multi-objective framework. We establish its convergence rate (Theorem 1) and sample complexity (Corollary 1), demonstrating both the advantage of the min-max multi-objective BLO framework and the validity of our theoretical analyses on robust representation learning and hyperparameter optimization applications. We wish to explore further applications where robustness would be beneficial such as in RL, federated learning and domain generalization. On the theoretical side, we wish to develop single-loop algorithms with improved convergence rates (for example, exploring techniques in Chen et al. (2022b)) and double-loop algorithms with convergence guarantees for applications where a single-loop algorithm is not feasible (e.g., federated learning). Finally, we also wish to develop algorithms for large $n$ (the number of objective pairs) or even $n \to \infty$ where `MORBiT` is not computationally feasible.

## REPRODUCIBILITY STATEMENT

The formal definitions, assumptions, precise theorem statments, high level proof outline and detailed proofs for our main theoretical results are presented in Appendix A. We provide appropriate citations for the datasets used in our experiments and the experimental setup and details are presented in Appendix C. Our implementation is available at `https://github.com/minimario/MORBiT`.

## ACKNOWLEDGEMENTS

A.G. is supported by the National Science Foundation (NSF) Graduate Research Fellowship under Grant No. 2141064, and T.-W. Weng is supported by NSF under Grant No. 2107189. We would like to thank the MIT-IBM Watson AI Lab (`https://mitibmwatsonailab.mit.edu/`) and the MIT-UROP program (`https://urop.mit.edu/`) for their support. We would also like to thank the organizers of the "Beyond First-order Methods in ML Systems" workshop at ICML'21 and the "Bilevel Stochastic Methods for Optimization and Learning" session at INFORMS'22 for giving us the opportunity to present various iterations of our work (Gu et al., 2021; 2022). Finally, we would like to thank Soumyadip Ghosh and Mark Squillante for some insightful discussions.

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

## A  CONVERGENCE ANALYSIS OF MORBiT

### A.1  ASSUMPTIONS

First, we begin by listing the assumptions we make:

**Assumption 1** (Regularity of the outer functions). *For all $i \in [n]$, assume that outer functions $f_i(x, y)$ and $\ell_i(x) = f_i(x, y_i^\star(x))$ satisfy the following properties:*

▶ *For any $x \in \mathcal{X}$, $f_i(x, \cdot)$ is Lipschitz (w.r.t. $y$) with constant $G_f > 0$.*

▶ *For any $x \in \mathcal{X}$, $\nabla_x f_i(x, \cdot)$ and $\nabla_{y_i} f_i(x, \cdot)$ are Lipschitz continuous (w.r.t. $y_i$) with constants $L_{f_x} > 0$ and $L_{f_y} > 0$.*

▶ *For any $y_i \in \mathbb{R}^{d_{y_i}}$, $\nabla_{y_i} f_i(\cdot, y_i)$ is Lipschitz continuous (w.r.t. $x$) with constant $\bar{L}_{f_y} > 0$.*

▶ *For any $x \in \mathcal{X}, y_i \in \mathbb{R}^{d_{y_i}}$, we have $\|\nabla_{y_i} f_i(x, y_i)\| \leq C_{fy}$ for $C_{fy} > 0$.*

▶ *The function $\ell_i(\cdot)$ is $\mu_\ell$ weakly convex (in $x$), so that for all $v, w \in \mathcal{X}$,*

$$\ell_i(w) \geq \ell_i(v) + \langle \nabla \ell_i(v), w - v \rangle - \mu_\ell \|w - v\|^2. \tag{13}$$

▶ *For all $x \in \mathcal{X}$, $\|\ell_i(x)\| \leq B_\ell$ for $B_\ell > 0$.*

▶ *For all $x \in \mathcal{X}$, $\|\nabla \ell_i(x)\| \leq C_\ell$, for $C_\ell > 0$.*

**Assumption 2** (Regularity of the inner functions). *Assume that inner functions $g_i(x, y_i), \forall i \in [n]$ satisfy:*

▶ *For any $x \in \mathcal{X}$ and $y_i \in \mathbb{R}^{d_{y_i}}$, $g_i(x, y_i)$ is twice continuously differentiable in $(x, y_i)$.*

▶ *For any $x \in \mathcal{X}$, $\nabla_{y_i} g_i(x, \cdot)$ is Lipschitz continuous (w.r.t. $y_i$) with constant $L_g$.*

▶ *For any $x \in \mathcal{X}$, $g_i(x, \cdot)$ is $\mu_g$-strongly convex in $y_i$, so that for all $v, w \in \mathbb{R}^{d_{y_i}}$,*

$$g_i(x, w) \geq g_i(x, v) + \langle \nabla_v g_i(x, v), w - v \rangle + \mu_g \|w - v\|^2. \tag{14}$$

▶ *For any $x \in \mathcal{X}$, $\nabla_{xy_i}^2 g_i(x, \cdot)$ and $\nabla_{y_i}^2 g_i(x, \cdot)$ are Lipschitz continuous (w.r.t. $y_i$) with constants $L_{gxy} > 0$ and $L_{gyy} > 0$, respectively.*

▶ *For any $x \in \mathcal{X}$ and $y_i \in \mathbb{R}^{d_{y_i}}$, we have $\|\nabla_{xy_i}^2 g_i(x, y_i)\| \leq C_{gxy}$ for some $C_{gxy} > 0$.*

▶ *For any $y_i \in \mathbb{R}^{d_{y_i}}$, $\nabla_{xy_i}^2 g_i(\cdot, y_i)$ and $\nabla_{y_i}^2 g_i(\cdot, y_i)$ are Lipschitz continuous (w.r.t. $x$) with constants $\bar{L}_{gxy} > 0$ and $\bar{L}_{gyy} > 0$, respectively.*

From these assumptions, we can show a few additional regularity-type conditions. Since these conditions can also be found in Ghadimi & Wang (2018, Lemma 2.2) and Hong et al. (2020, Lemma 2), we state these results without proof.

**Lemma 5** (Corollary of Assumptions). *Under Assumptions 1 and 2 stated above, for all $x, x_1, x_2 \in \mathcal{X} \subseteq \mathbb{R}^{d_x}, y \in \mathbb{R}^{d_{y_i}}, i \in [n]$, we have*

$$\|\bar{\nabla}_x f_i(x, y) - \nabla \ell_i(x)\| \leq L\|y_i^\star(x) - y\|, \tag{15a}$$

$$\|y_i^\star(x_1) - y_i^\star(x_2)\| \leq G_y\|x_1 - x_2\|, \tag{15b}$$

$$\|\nabla \ell_i(x_1) - \nabla \ell_i(x_2)\| \leq L_f\|x_1 - x_2\|, \tag{15c}$$

*where we define*

$$L \triangleq L_{f_x} + \frac{L_{f_y} C_{gxy}}{\mu_g} + C_{f_y}\left(\frac{L_{gxy}}{\mu_g} + \frac{L_{gyy} C_{gxy}}{\mu_g^2}\right), \tag{16a}$$

$$L_f \triangleq L_{f_x} + \frac{(\bar{L}_{f_y} + L) C_{gxy}}{\mu_g} + C_{f_y}\left(\frac{\bar{L}_{gxy}}{\mu_g} + \frac{\bar{L}_{gyy} C_{gxy}}{\mu_g^2}\right), \tag{16b}$$

$$G_y \triangleq \frac{C_g}{\mu_g}. \tag{16c}$$

**Assumption 3** (Quality of stochastic gradient estimates). *For any iteration $k > 0$ and all $i \in [n]$, the gradient estimates $h_i^{(k)}$ for the LL variable $y_i$ satisfy the following for some $\sigma_g > 0$ (Hong et al., 2020; Lu et al., 2022):*

$$\mathbb{E}[h_i^{(k)}] = \nabla_{y_i} g_i(x^{(k)}, y_i^{(k)}), \tag{17}$$

$$\mathbb{E}[\|h_i^{(k)} - \nabla_{y_i} g_i(x^{(k)}, y_i^{(k)})\|^2] \leq \sigma_g^2(1 + \|\nabla_{y_i} g_i(x^{(k)}, y_i^{(k)})\|^2). \tag{18}$$

*For any iteration $k > 0$, the gradient estimate $h_\lambda^{(k)}$ for the simplex variable $\lambda$ satisfies:*

$$\mathbb{E}[h_\lambda^{(k)}] = \nabla_\lambda F(x^{(k)}, \boldsymbol{y}^{(k+1)}, \lambda^{(k)}) = \left[f_1(x^{(k)}, y_1^{(k+1)}), \cdots, f_n(x^{(k)}, y_n^{(k+1)})\right]^\top. \tag{19}$$

*For any $k \geq 0$ and a $\sigma_f > 0$, we assume that there exists a non-increasing sequence $\{b_k\}_{k \geq 0}$ such that*

$$\mathbb{E}[h_x^{(k)}] = \overline{\nabla}_x F(x^{(k)}, \boldsymbol{y}^{(k+1)}, \lambda^{(k)}) + B_k, \|B_k\| \leq b_k, \tag{20}$$

$$\mathbb{E}[\|h_x^{(k)} - \mathbb{E}[h_x^{(k)}]\|^2] \leq \sigma_f^2. \tag{21}$$

### A.2 MAIN THEOREM AND REMARKS

We now state our main theorem, in full. Recall our notation from equation 9: for a fixed constant $\rho > 0$, we defined the proximal map to be

$$\hat{x}(z) \triangleq \arg\min_{x \in \mathcal{X}} \frac{\rho}{2}\|x - z\|^2 + F(x, \lambda). \tag{22}$$

We also define the Moreau envelope as

$$\Phi_{1/\rho}(z) \triangleq \min_x \frac{\rho}{2}\|x - z\|^2 + \sum_{i=1}^n \lambda_i \ell_i(x). \tag{23}$$

In addition, we use the notation $\Delta_{y_i}^{(k)} \triangleq \mathbb{E}[\|y_i^{(k)} - y_i^\star(x^{(k-1)})\|^2]$ and $i^{(k)} \triangleq \arg\max_{i \in [n]} \Delta_{y_i}^{(k)}$. Finally, we define $\tilde{\sigma}_f^2 = \sigma_f^2 + 3C_\ell^2$ (see Lemma 12). Now, we are ready to state the full version of our main theorem:

**Theorem 2** (Convergence of MORBiT). *Under Assumptions 1, 2 and 3, and the terms defined in Lemma 5, when step sizes are chosen as*

$$\alpha = \min\left(\frac{\mu_g}{16G_y L}\nu, \frac{K^{-3/5}}{4G_y L}\right), \tag{24}$$

$$\beta = \min\left(\nu, \frac{4K^{-2/5}}{\mu_g}\right), \tag{25}$$

$$\gamma = \frac{2K^{-3/5}}{B_\ell n^{1/2}}, \tag{26}$$

*where $\nu = \min\left(\mu_g/L_g^2(1+\sigma_g^2), 1/\mu_g\right)$, Algorithm 1 produces $\bar{x}, \bar{\lambda}, \bar{y}_i, i \in [n]$ satisfying:*

$$\mathbb{E}\left[\max_{i \in [n]} \|\bar{y}_i - y_i^\star(\bar{x})\|^2\right] \leq A, \tag{27}$$

$$\max_\lambda \mathbb{E}[F(\bar{x}, \lambda)] - \mathbb{E}[F(\bar{x}, \bar{\lambda})] \leq \sqrt{2}B_\ell\sqrt{n}K^{-2/5} + G_f A, \tag{28}$$

$$\mathbb{E}[\|\hat{x}(\bar{x}) - \bar{x}\|^2] \leq \frac{16\Phi_{1/\rho}(x^{(0)})G_y L}{(-\mu_\ell + \rho)\rho K^{2/5}} + \frac{8(b_0^2 + L^2 A)}{(-\mu_\ell + \rho)^2} + \frac{2\alpha(\tilde{\sigma}_f^2 + 3b_0^2 + 3L^2 A)}{-\mu_\ell + \rho}, \tag{29}$$

*where*

$$A = \frac{\Delta_{y_{i(0)}}^{(0)}/\mu_g}{K^{3/5}} + \frac{16\sigma_g^2/\mu_g^2}{K^{7/5}} + \frac{G_y/L}{K^{4/5}} + \frac{2\sqrt{n}B_\ell G_y/L}{K^{2/5}} + \frac{(b_0^2 + \frac{1}{2}\sigma_f^2)/(L^2)}{K^{2/5}} + \frac{16\sigma_g^2/\mu_g^2}{K^{2/5}}. \tag{30}$$

**Connection of** TTSA **(Hong et al., 2020).** As we generalize Hong et al. (2020), our proof follows a similar structure. In particular, Lemma 6 is a generalization of Hong et al. (2020, Equation (14)), and our Lemma 7 combines Hong et al. (2020, Lemma 3 and 4). Lemmas 8, 9, and 11 in our work parallel Hong et al. (2020, Lemmas 6, 5, and 7), respectively. Lemma 10 in our work deals with the maximization problem w.r.t. $\lambda$, so there is no analogue in Hong et al. (2020). However, it borrows techniques from Collins et al. (2020, Theorem 1).

We also discuss the convergence rate. Note that $A$ in equation 30 is dominated by the fourth term, $\sqrt{n}/K^{2/5}$, so it is clear that equation 27 and equation 28 converge at a rate of $\mathcal{O}(\sqrt{n}K^{-2/5})$. We give special attention to equation 29). Apart from the $8b_0^2/(-\mu_\ell+\rho)^2$ term, we see that the RHS of equation 29 converges at a rate of $\mathcal{O}(\sqrt{n}K^{-2/5})$. To understand the convergence of this term, we turn to (20) from Assumption 3. As discussed in section 3.2, $b_k$ can be made arbitrarily small by running more iterations of the subroutine for estimating $h_x^{(k)}$ (for example utilizing the HIA sampling scheme (Agarwal et al., 2017; Ghadimi & Wang, 2018; Hong et al., 2020)). Therefore, as long as we run enough iterations ($O(\log K)$ for HIA) such that $b_k^2 \leq \alpha$, (29) will also converge at a rate of $\mathcal{O}(\sqrt{n}K^{-2/5})$.

### A.3 PROOF PLAN

**Overall Roadmap:** In what follows, we give a proof sketch of our main theorem, with constant terms abstracted away with $\mathcal{O}$ notation. $c_1, c_2, c_3, c_4$ are positive constants depending on $L_f$ (defined in Lemma 5), $\mu_g$ (the LL objective convexity defined in Assumption 2) and the learning rates $\alpha, \beta$ (in algorithm 1).

In order to show equation 10a of Theorem 1 (the convergence of $x$), we will derive a descent lemma comparing successive iterates $x^{(k)}$ and $x^{(k+1)}$. Descent lemmas often contain a quadratic term $\|x^{(k+1)} - x^{(k)}\|^2$, so it is natural that we will have to bound $\|h_x^{(k)}\|^2$. As such, in Lemma 6, we bound the averaged squared norm of the stochastic gradient estimate, $\mathbb{E}[\|h_x^{(k)}\|^2]$:

**Lemma 6.** *Under our regularity assumptions, the average squared norm of $h_x^{(k)}$ can be bounded as follows, where the expectation is over the filtration $\mathcal{F}_i' \triangleq \{y_i^{(0)}, x^{(0)}, \cdots, y_i^{(k)}, x^{(k)}, y_i^{(k+1)}\}$:*

$$\mathbb{E}[\|h_x^{(k)}\|^2] \leq \mathcal{O}\left(\sum_{i=1}^n \lambda_i^{(k)} \|y_i^\star(x^{(k)}) - y_i^{(k+1)}\|^2\right). \tag{31}$$

Turning to equation 10a of Theorem 1, we'll use a descent relation on $y_i^{(k)} - y_i^\star(x^{(k-1)})$. While ideally we would obtain a descent relation purely involving $y_i^{((k))}$ terms themselves, the intricate coupling of the $x$ and $y$ terms result in an extra $x^{(k-1)} - x^{(k)}$ term. The resulting relation is shown in Lemma 7. Here, we have that $c_1 = 1 - \frac{\mu_g\beta}{2}, c_2 = \frac{2}{\mu_g\beta} - 1$, where $\mu_g\beta < 1$.

**Lemma 7.** *The distance between the algorithm's iterates $y_i^{(k)}$ and the true inner optimum $y_i^\star(x^{(k)})$ satisfies the following descent equation,*

$$\mathbb{E}[\|y_i^{(k+1)} - y_i^\star(x^{(k)})\|^2] \leq \mathcal{O}((1 - c_1)\mathbb{E}[\|y_i^{(k)} - y_i^\star(x^{(k-1)})\|^2] + c_2\mathbb{E}[\|x^{(k-1)} - x^{(k)}\|^2]). \tag{32}$$

From this lemma, intuitively, we know that $\mathbb{E}[\|y_i^{(k)} - y_i^\star(x^{(k-1)})\|^2]$ is decreasing as $k$ increases, *as long as the $\mathbb{E}[\|x^{(k-1)} - x^{(k)}\|^2]$'s are not too large*. Therefore, it is important to have another descent relation that upper bounds this quantity, which we do next in Lemma 8. The lemma naturally involves the objective $\mathcal{L}^{(k)} = F(x^{(k)}, \lambda^{(k)})$, which will telescope. Here, we have $c_3 = \frac{1}{4\alpha} - \frac{L_f}{2}, c_4 = 4\alpha L^2$. As $k \to \infty$, $\alpha \to 0$, and $c_3$ is positive.

**Lemma 8.** *Let $\mathcal{L}^{(k)} \triangleq \mathbb{E}[F(x^{(k)}, \lambda^{(k)})] = \sum_{i=1}^n \lambda_i^{(k)}\mathbb{E}[\ell_i(x^{(k)})]$. Then, the $\mathcal{L}^{(k)}$ satisfies the descent equation*

$$\mathcal{L}^{(k+1)} - \mathcal{L}^{(k)} \leq \mathcal{O}\left(-c_3\mathbb{E}[\|x^{(k+1)} - x^{(k)}\|^2] + c_4 \max_{i \in [n]}\mathbb{E}[\|y_i^{(k+1)} - y_i^\star(x^{(k)})\|^2] + \sqrt{n}\gamma + \alpha\right). \tag{33}$$

Following the intuition previously described, we then use Lemmas 7 and 8 to show that the $\mathbb{E}[\|x^{(k-1)} - x^{(k)}\|^2]$ terms are small enough and that the $y_i^{(k)}$ iterates converge:

**Lemma 9** (Informal, see Appendix A.7 for precise statement)**.**

$$\frac{1}{K} \sum_{k=1}^{K} \max_{i \in [n]} \mathbb{E}\left[\|y_i^{(k)} - y_i^\star(x^{(k-1)})\|^2\right] \leq \mathcal{O}(\sqrt{n} K^{-2/5}). \tag{34}$$

Lemma 10 then leverages the convergence of $y_i, i \in [n]$ to bound the convergence of $\lambda$, and Lemma 11 shows the bound on $x$. By plugging in our step-sizes into Lemmas 9, 10, and 11, Theorem 1 directly follows.

**Lemma 10.** *For any $\lambda \in \Delta_n$, the iterates of Algorithm 1 satisfy*

$$\frac{1}{K} \mathbb{E}\left[\sum_{k=1}^{K} F(x^{(k)}, \lambda) - F(x^{(k)}, \lambda^{(k)})\right] \leq \mathcal{O}(\sqrt{n} K^{-2/5}). \tag{35}$$

**Lemma 11.** *The iterates of Algorithm 1 satisfy*

$$\frac{1}{K} \sum_{k=1}^{K} \mathbb{E}[\|\hat{x}(x^{(k)}) - x^{(k)}\|^2] \leq \mathcal{O}(\sqrt{n} K^{-2/5}). \tag{36}$$

### A.4 PROOF OF LEMMA 1 (LEMMA 6)

Stating Lemma 1 more precisely:

**Lemma 12.** *Under Assumptions 1, 2 and 3, the average squared norm of $h_x^{(k)}$ can be bounded as follows, where $\tilde{\sigma}_f^2 = \sigma_f^2 + 3C_\ell^2$:*

$$\mathbb{E}[\|h_x^{(k)}\|^2] \leq \tilde{\sigma}_f^2 + 3b_k^2 + 3L^2 \sum_{i=1}^{n} \lambda_i^{(k)} \|y_i^\star(x^{(k)}) - y_i^{(k+1)}\|^2, \tag{37}$$

*where the expectation is over the filtration $\mathcal{F}_i' \triangleq \{y_i^{(0)}, x^{(0)}, \cdots, y_i^{(k)}, x^{(k)}, y_i^{(k+1)}\}$.*

Note that here the expectation is over $\mathcal{F}_i' \triangleq \{y_i^{(0)}, x^{(0)}, \cdots, y_i^{(k)}, x^{(k)}, y_i^{(k+1)}\}$, so no expectation is needed in the last term.

*Proof.* We can derive the following:

$$\mathbb{E}[\|h_x^{(k)}\|^2] \stackrel{(1)}{=} \mathbb{E}[\|h_x^{(k)} - \mathbb{E}[h_x^{(k)}]\|^2] + \|\mathbb{E}[h_x^{(k)}]\|^2 \tag{38}$$

$$\stackrel{(2)}{=} \mathbb{E}[\|h_x^{(k)} - \mathbb{E}[h_x^{(k)}]\|^2] + \|\overline{\nabla}_x F(x^{(k)}, \boldsymbol{y}^{(k+1)}, \lambda^{(k)}) + B_k\|^2 \tag{39}$$

$$\stackrel{(3)}{\leq} \sigma_f^2 + \|\overline{\nabla}_x F(x^{(k)}, \boldsymbol{y}^{(k+1)}, \lambda^{(k)}) + B_k\|^2 \tag{40}$$

$$\stackrel{(4)}{=} \sigma_f^2 + \left\|\sum_{i=1}^{n} \lambda_i^{(k)} \overline{\nabla}_x f_i(x^{(k)}, y_i^{(k+1)}) + B_k\right\|^2 \tag{41}$$

$$\stackrel{(5)}{\leq} \sigma_f^2 + 3b_k^2 + \frac{3}{2} \left\|\sum_{i=1}^{n} \lambda_i^{(k)} \overline{\nabla}_x f_i(x^{(k)}, y_i^{(k+1)})\right\|^2. \tag{42}$$

(1) is true because

$$\mathbb{E}[\|h_x^{(k)} - \mathbb{E}[h_x^{(k)}]\|^2] + \|\mathbb{E}[h_x^{(k)}]\|^2 = \mathbb{E}[\|h_x^{(k)}\|^2] + \|\mathbb{E}[h_x^{(k)}]\|^2 - 2\mathbb{E}\langle h_x^{(k)}, \mathbb{E}[h_x^{(k)}]\rangle + \|\mathbb{E}[h_x^{(k)}]\|^2 \tag{43}$$

$$= \mathbb{E}[\|h_x^{(k)}\|^2] + \|\mathbb{E}[h_x^{(k)}]\|^2 - \langle\mathbb{E}[h_x^{(k)}], \mathbb{E}[h_x^{(k)}]\rangle + \|\mathbb{E}[h_x^{(k)}]\|^2 \tag{44}$$

$$= \mathbb{E}[\|h_x^{(k)}\|^2]. \tag{45}$$

(2) follows from (20), (3) follows from (21), (4) follows from definition of $\overline{\nabla}_x$, and (5) follows from Young's inequality, $\|a + b\|^2 \leq 3\|a\|^2 + \frac{3}{2}\|b\|^2$. Next, we bound the last term in (42). We start by using the fact that

$$
\left\| \sum_{i=1}^{n} \lambda_i^{(k)} \overline{\nabla}_x f_i(x^{(k)}, y_i^{(k+1)}) \right\|^2
$$

$$
\overset{(1)}{\leq} 2 \left\| \sum_{i=1}^{n} \lambda_i^{(k)} (\overline{\nabla}_x f_i(x^{(k)}, y_i^{(k+1)}) - \nabla \ell_i(x^{(k)})) \right\|^2 + 2 \left\| \sum_{i=1}^{n} \lambda_i^{(k)} \nabla \ell_i(x^{(k)}) \right\|^2 \tag{46}
$$

$$
\overset{(2)}{\leq} 2 \sum_{i=1}^{n} \lambda_i^{(k)} \|\overline{\nabla}_x f_i(x^{(k)}, y_i^{(k+1)}) - \nabla \ell_i(x^{(k)})\|^2 + 2 \sum_{i=1}^{n} \lambda_i^{(k)} \|\nabla \ell_i(x^{(k)})\|^2, \tag{47}
$$

where (1) follows from $\|a + b\|^2 \leq 2\|a\|^2 + 2\|b\|^2$, and (2) follows from $\left\| \sum_{i=1}^{n} p_i a_i \right\|^2 \leq \sum_{i=1}^{n} p_i \|a_i\|^2$.
Next, we bound the first term in (47). From Lemma 5, we have

$$
\|\overline{\nabla}_x f_i(x^{(k)}, y_i^{(k+1)}) - \nabla \ell_i(x^{(k)})\|^2 \leq L \|y_i^{\star}(x^{(k)}) - y_i^{(k+1)}\|^2. \tag{48}
$$

Therefore, we can obtain

$$
\mathbb{E}[\|h_x^{(k)}\|^2] \overset{(1)}{\leq} \sigma_f^2 + 3b_k^2 + 3L^2 \sum_{i=1}^{n} \lambda_i^{(k)} \|y_i^{\star}(x^{(k)}) - y_i^{(k+1)}\|^2 + 3 \sum_{i=1}^{n} \lambda_i^{(k)} \|\nabla \ell_i(x^{(k)})\|^2 \tag{49}
$$

$$
\overset{(2)}{\leq} \tilde{\sigma}_f^2 + 3b_k^2 + 3L^2 \sum_{i=1}^{n} \lambda_i^{(k)} \|y_i^{\star}(x^{(k)}) - y_i^{(k+1)}\|^2, \tag{50}
$$

where (1) comes from plugging (48) into (47) and (47) into (42), (2) comes from the definition of $\tilde{\sigma}_f^2$ using $\|\nabla \ell_i(x^{(k)})\|^2 \leq C_\ell^2$ and $\lambda^{(k)} \in \Delta_n$. $\qquad \square$

## A.5 PROOF OF LEMMA 2 (LEMMA 7)

We state the precise version of Lemma 2 here:

**Lemma 13.** *Under Assumptions 1, 2, and 3, when $\beta^2(1 + \sigma_g^2)L_g^2 \leq \beta\mu_g$ and $\mu_g\beta < 1$, the iterates $y_i^{(k)}$ satisfy the descent equation:*

$$
\mathbb{E}[\|y_i^{(k+1)} - y_i^{\star}(x^{(k)})\|^2] \tag{51}
$$
$$
\leq \left(1 - \frac{\mu_g\beta}{2}\right)\|y_i^{(k)} - y_i^{\star}(x^{(k-1)})\|^2 + \left(\frac{2}{\mu_g\beta} - 1\right)G_y^2\|x^{(k-1)} - x^{(k)}\|^2 + \beta^2\sigma_g^2.
$$

*Proof.* For a particular (fixed) realization of the iterates $x^{(1)}, \cdots, x^{(k)}, y_i^{(1)}, \cdots, y_i^{(k)}$ for some $i \in [n]$, we have

$$
\mathbb{E}[\|h_i^{(k)}\|^2] \overset{(1)}{=} \mathbb{E}[\|h_i^{(k)} - \mathbb{E}[h_i^{(k)}]\|^2] + \|\mathbb{E}[h_i^{(k)}]\|^2 \tag{52}
$$

$$
\overset{(2)}{=} \mathbb{E}[\|h_i^{(k)} - \nabla_{y_i} g_i(x^{(k)}, y_i^{(k)})\|^2] + \|\nabla_{y_i} g_i(x^{(k)}, y_i^{(k)})\|^2 \tag{53}
$$

$$
\overset{(3)}{\leq} \sigma_g^2 + (1 + \sigma_g^2)\|\nabla_{y_i} g_i(x^{(k)}, y_i^{(k)})\|^2 \tag{54}
$$

$$
\overset{(4)}{\leq} \sigma_g^2 + (1 + \sigma_g^2)\|\nabla_{y_i} g_i(x^{(k)}, y_i^{(k)}) - \nabla_{y_i} g_i(x^{(k)}, y_i^{\star}(x^{(k)}))\|^2 \tag{55}
$$

$$
\overset{(5)}{\leq} \sigma_g^2 + (1 + \sigma_g^2)L_g^2 \|y_i^{(k)} - y_i^{\star}(x^{(k)})\|^2, \tag{56}
$$

where (1) follows from algebra, (2) follows from $\mathbb{E}[h_i^{(k)}] = \nabla_{y_i} g_i(x^{(k)}, y_i^{(k)})$ in equation 17 in Assumption 3, (3) is from equation 18 in Assumption 3, (4) is from $\nabla_{y_i} g_i(x^{(k)}, y_i^{\star}(x^{(k)})) = 0$ due to the optimality of $y_i^{\star}(x^{(k)})$, and (5) is due to the $L_g$-Lipschitz continuity of $\nabla_{y_i} g_i(x, \cdot)$.

Next, we can bound the difference between $y_i^{(k+1)}$ and $y_i^\star(x^{(k)})$ as the following, where again we assume that $x^{(1)}, \cdots, x^{(k)}, y^{(1)}, \cdots, y_i^{(k)}$ is fixed and the expectation is over the stochasticity of the gradient estimates:

$$\mathbb{E}[\|y_i^{(k+1)} - y_i^\star(x^{(k)})\|^2]$$

$$\stackrel{(1)}{=} \mathbb{E}[\|y_i^{(k)} - \beta h_i^{(k)} - y_i^\star(x^{(k)})\|^2] \tag{57}$$

$$\stackrel{(2)}{=} \|y_i^{(k)} - y_i^\star(x^{(k)})\|^2 + \beta^2 \mathbb{E}[\|h_i^{(k)}\|^2] - 2\beta \langle y_i^{(k)} - y_i^\star(x^{(k)}), \nabla_{y_i} g_i(x^{(k)}, y_i^{(k)}) \rangle \tag{58}$$

$$\stackrel{(3)}{\leq} (1 - 2\beta\mu_g)\|y_i^{(k)} - y_i^\star(x^{(k)})\|^2 + \beta^2 \mathbb{E}[\|h_i^{(k)}\|^2] \tag{59}$$

$$\stackrel{(4)}{\leq} (1 - 2\beta\mu_g)\|y_i^{(k)} - y_i^\star(x^{(k)})\|^2 + \beta^2 \sigma_g^2 + \beta^2(1 + \sigma_g^2)L_g^2 \|y_i^{(k)} - y_i^\star(x^{(k)})\|^2 \tag{60}$$

$$\stackrel{(5)}{\leq} (1 - \beta\mu_g)\|y_i^{(k)} - y_i^\star(x^{(k)})\|^2 + \beta^2 \sigma_g^2 \tag{61}$$

$$\stackrel{(6)}{\leq} (1 - \beta\mu_g) \left[ (1 + c)\|y_i^{(k)} - y_i^\star(x^{(k-1)})\|^2 + \left(1 + \frac{1}{c}\right) \|y_i^\star(x^{(k-1)}) - y_i^\star(x^{(k)})\|^2 \right] + \beta^2 \sigma_g^2 \tag{62}$$

$$\stackrel{(7)}{\leq} (1 - \beta\mu_g) \left[ (1 + c)\|y_i^{(k)} - y_i^\star(x^{(k-1)})\|^2 + \left(1 + \frac{1}{c}\right) G_y^2 \|x^{(k-1)} - x^{(k)}\|^2 \right] + \beta^2 \sigma_g^2, \tag{63}$$

where (1) is true by definition, and (2) holds by direct algebra and the unbiasedness assumption $\mathbb{E}[h_i^{(k)}] = \nabla_{y_i} g_i(x^{(k)}, y_i^{(k)})$ in equation 17 in Assumption 3, (3) is from strong convexity, $\beta \left\langle \nabla_{y_i} g_i(x^{(k)}, y_i^{(k)}), y_i^{(k)} - y_i^\star(x^{(k)}) \right\rangle \geq \beta\mu_g \|y_i^{(k)} - y_i^\star(x^{(k)})\|^2$, (4) is from equation 56, (5) is from the assumption $\beta^2(1 + \sigma_g^2)L_g^2 \leq \beta\mu_g$, (6) is from the inequality $\|a + b\|^2 \leq (1 + 1/c)\|a\|^2 + (1 + c)\|b\|^2$, and (7) is from the $G_y$-lipschitzness of $y_i^\star(\cdot)$ in Lemma 5.

Then, we choose $c = \frac{\mu_g \beta}{2(1 - \mu_g \beta)}$, so that $(1 - \beta\mu_g)(1 + c) = 1 - \beta\mu_g/2$ and $1 + 1/c = \frac{2}{\mu_g \beta} - 1$. We have $c > 0$ because $\mu_g\beta < 1$. Plugging these expressions into (63), we get

$$\|y_i^{(k+1)} - y_i^\star(x^{(k)})\|^2 \tag{64}$$
$$\leq \left(1 - \frac{\mu_g \beta}{2}\right) \|y_i^{(k)} - y_i^\star(x^{(k-1)})\|^2 + \left(\frac{2}{\mu_g \beta} - 1\right) G_y^2 \|x^{(k-1)} - x^{(k)}\|^2 + \beta^2 \sigma_g^2,$$

which completes the proof. $\qquad\square$

### A.6 PROOF OF LEMMA 3 (LEMMA 8)

We state the precise version of Lemma 3 here:

**Lemma 14.** *Let* $\mathcal{L}^{(k)} \triangleq \mathbb{E}[F(x^{(k)}, \lambda^{(k)})] = \sum\limits_{i=1}^{n} \lambda_i^{(k)} \mathbb{E}[\ell_i(x^{(k)})]$. *Under Assumptions 1, 2 and 3, assume that the iterates* $\{x^{(k)}, y_i^{(k)}, i \in [n], \lambda^{(k)}, \forall k\}$ *are generated by* MORBiT, *then,* $\mathcal{L}^{(k)}$ *satisfies the descent equation*

$$\mathcal{L}^{(k+1)} - \mathcal{L}^{(k)}$$
$$\leq 4\alpha L^2 \max_{i \in [n]} \Delta_{y_i}^{(k+1)} + \left(\frac{L_f}{2} - \frac{1}{4\alpha}\right) \mathbb{E}[\|x^{(k+1)} - x^{(k)}\|^2] + \gamma n B_\ell^2 + 4\alpha b_0^2 + 2\alpha\sigma_f^2. \tag{65}$$

*Proof.* First, since $\ell_i$ is $L_f$-smooth, we know that for all $i \in [n]$,

$$\ell_i(x^{(k+1)}) \leq \ell_i(x^{(k)}) + \langle x^{(k+1)} - x^{(k)}, \nabla \ell_i(x^{(k)}) \rangle + \frac{L_f}{2} \|x^{(k+1)} - x^{(k)}\|^2. \tag{66}$$

Taking $\lambda_i^{(k)}$ times the equation for $i$ in (66) and summing, we can get

$$
\begin{aligned}
&\sum_{i=1}^{n} \lambda_i^{(k)} \ell_i(x^{(k+1)}) \\
&\leq \sum_{i=1}^{n} \lambda_i^{(k)} \ell_i(x^{(k)}) + \left\langle x^{(k+1)} - x^{(k)}, \sum_{i=1}^{n} \lambda_i^{(k)} \nabla \ell_i(x^{(k)}) \right\rangle + \frac{L_f}{2} \|x^{(k+1)} - x^{(k)}\|^2.
\end{aligned}
\tag{67}
$$

Therefore, we have

$$
\begin{aligned}
&\sum_{i=1}^{n} \lambda_i^{(k+1)} \ell_i(x^{(k+1)}) - \sum_{i=1}^{n} \lambda_i^{(k)} \ell_i(x^{(k)}) \\
&\leq \underbrace{\sum_{i=1}^{n} \lambda_i^{(k+1)} \ell_i(x^{(k+1)}) - \sum_{i=1}^{n} \lambda_i^{(k)} \ell_i(x^{(k+1)})}_{\triangleq (A)} + \underbrace{\langle x^{(k+1)} - x^{(k)}, \sum_{i=1}^{n} \lambda_i^{(k)} \nabla \ell_i(x^{(k)}) \rangle}_{\triangleq (B)} \\
&\quad + \frac{L_f}{2} \|x^{(k+1)} - x^{(k)}\|^2.
\end{aligned}
\tag{68}
$$

Next, we bound $(A)$ and $(B)$ respectively as follows. First, we upper bound term $(A)$. First, from the non-expansiveness of projections and $\lambda^{(k+1)} = \mathrm{proj}_{\Delta_n}(\lambda^{(k)} + \gamma h_\lambda^{(k)})$, we have $\|\lambda^{(k+1)} - \lambda^{(k)}\| \leq \|\gamma h_\lambda^{(k)}\|$. Since $\lambda^{(k+1)}, \lambda^{(k)} \in \Delta_n$, $\|\lambda^{(k+1)} - \lambda^{(k)}\| \leq \sqrt{2}$. Therefore, we know that $\|\lambda^{(k+1)} - \lambda^{(k)}\| \leq \Lambda \triangleq \min\{\sqrt{2}, \gamma \|h_\lambda^{(k)}\|\}$. Based on these facts, we can have

$$
(A) = \sum_{i=1}^{n} \lambda_i^{(k+1)} \ell_i(x^{(k+1)}) - \sum_{i=1}^{n} \lambda_i^{(k)} \ell_i(x^{(k+1)})
\tag{69}
$$

$$
\overset{(1)}{=} \sum_{i=1}^{n} \left( \lambda_i^{(k+1)} - \lambda_i^{(k)} \right) \ell_i(x^{(k+1)})
\tag{70}
$$

$$
\overset{(2)}{\leq} \left\| \lambda^{(k+1)} - \lambda^{(k)} \right\| \left\| \left[ \ell_1(x^{(k+1)}), \ell_2(x^{(k+1)}), \cdots, \ell_n(x^{(k+1)}) \right]^\top \right\|
\tag{71}
$$

$$
\overset{(3)}{\leq} \Lambda B_\ell \sqrt{n} \quad \overset{(4)}{\leq} \sqrt{n} \min\{\sqrt{2}, \|\gamma h_\lambda^{(k)}\|\} B_\ell \quad \overset{(5)}{\leq} \sqrt{n} \gamma \|h_\lambda^{(k)}\| B_\ell,
\tag{72}
$$

where (1) is straightforward, (2) follows from Cauchy-Schwarz, (3) follows from the update rule for $\lambda$ and the fact that $|\ell_i(\cdot)| \leq B_\ell$ from Assumption 1, (4) is from plugging in the definition of $\Lambda$, and (5) follows from $\gamma \|h_\lambda^k\| \leq \sqrt{2}$.

Then, we upper bound $(B)$. First, from the non-expansiveness of projection and the update rule $x^{(k+1)} = \mathrm{proj}_\mathcal{X}(x^{(k)} - \alpha h_x^{(k)})$, we know that

$$
\|x^{(k+1)} - x^{(k)} + \alpha h_x^{(k)}\|^2 \leq \| - \alpha h_x^{(k)}\|^2,
\tag{73}
$$

$$
\Rightarrow \|x^{(k+1)} - x^{(k)}\|^2 + 2\alpha \langle x^{(k+1)} - x^{(k)}, h_x^{(k)} \rangle \leq 0,
\tag{74}
$$

$$
\Rightarrow \frac{1}{2\alpha} \|x^{(k+1)} - x^{(k)}\|^2 + \langle x^{(k+1)} - x^{(k)}, h_x^{(k)} \rangle \leq 0.
\tag{75}
$$

Therefore, we can have

$$(B) = \left\langle \left( \sum_{i=1}^{n} \lambda_i^{(k)} \nabla \ell_i(x^{(k)}) \right), \left( x^{(k+1)} - x^{(k)} \right) \right\rangle \overset{(1)}{=} \left\langle \nabla_x F(x^{(k)}, \lambda^{(k)}), \left( x^{(k+1)} - x^{(k)} \right) \right\rangle \tag{76}$$

$$\overset{(2)}{=} \left\langle \left( \nabla_x F(x^{(k)}, \lambda^{(k)}) - \overline{\nabla}_x F(x^{(k)}, \boldsymbol{y}^{(k+1)}, \lambda^{(k)}) - B_k \right), \left( x^{(k+1)} - x^{(k)} \right) \right\rangle$$
$$+ \left\langle \left( \overline{\nabla}_x F(x^{(k)}, \boldsymbol{y}^{(k+1)}, \lambda^{(k)}) + B_k \right), \left( x^{(k+1)} - x^{(k)} \right) \right\rangle \tag{77}$$

$$\overset{(3)}{\leq} \left\langle \left( \nabla_x F(x^{(k)}, \lambda^{(k)}) - \overline{\nabla}_x F(x^{(k)}, \boldsymbol{y}^{(k+1)}, \lambda^{(k)}) - B_k \right), \left( x^{(k+1)} - x^{(k)} \right) \right\rangle \tag{78}$$
$$+ \left\langle \left( \overline{\nabla}_x F(x^{(k)}, \boldsymbol{y}^{(k+1)}, \lambda^{(k)}) + B_k - h_x^{(k)} \right), \left( x^{(k+1)} - x^{(k)} \right) \right\rangle - \frac{1}{2\alpha} \|x^{(k+1)} - x^{(k)}\|^2$$

$$\overset{(4)}{\leq} \frac{1}{2c} \|\nabla_x F(x^{(k)}, \lambda^{(k)}) - \overline{\nabla}_x F(x^{(k)}, \boldsymbol{y}^{(k+1)}, \lambda^{(k)}) - B_k\|^2 + \frac{c}{2} \|x^{(k+1)} - x^{(k)}\|^2$$
$$+ \frac{1}{2d} \|\overline{\nabla}_x F(x^{(k)}, \boldsymbol{y}^{(k+1)}, \lambda^{(k)}) + B_k - h_x^{(k)}\|^2 + \frac{d}{2} \|x^{(k+1)} - x^{(k)}\|^2$$
$$- \frac{1}{2\alpha} \|x^{(k+1)} - x^{(k)}\|^2 \tag{79}$$

$$\overset{(5)}{\leq} \frac{1}{2c} \|\nabla_x F(x^{(k)}, \lambda^{(k)}) - \overline{\nabla}_x F(x^{(k)}, \boldsymbol{y}^{(k+1)}, \lambda^{(k)}) - B_k\|^2 + \frac{c}{2} \|x^{(k+1)} - x^{(k)}\|^2$$
$$+ \frac{\sigma_f^2}{2d} + \frac{d}{2} \|x^{(k+1)} - x^{(k)}\|^2 - \frac{1}{2\alpha} \|x^{(k+1)} - x^{(k)}\|^2 \tag{80}$$

$$\overset{(6)}{=} \frac{1}{2c} \|\nabla_x F(x^{(k)}, \lambda^{(k)}) - \overline{\nabla}_x F(x^{(k)}, \boldsymbol{y}^{(k+1)}, \lambda^{(k)}) - B_k\|^2$$
$$+ \left( \frac{c+d}{2} - \frac{1}{2\alpha} \right) \|x^{(k+1)} - x^{(k)}\|^2 + \frac{\sigma_f^2}{2d}, \tag{81}$$

where (1) is by definition of $F(x^{(k)}, \lambda^{(k)})$, (2) is from adding and subtracting $\left\langle \left( \overline{\nabla}_x F(x^{(k)}, \boldsymbol{y}^{(k+1)}, \lambda^{(k)}) + B_k \right), \left( x^{(k+1)} - x^{(k)} \right) \right\rangle$, (3) is from adding (75) to the previous inequality, (4) is from applying the inequality $\langle a, b \rangle \leq \frac{1}{2c} \|a\|^2 + \frac{c}{2} \|b\|^2$ to both inner product terms, (5) is from equation 20 and equation 21 in Assumption 3, and (6) is from algebra.

Plugging in our expressions for (A) and (B) into (68), we get

$$\sum_{i=1}^{n} \lambda_i^{(k+1)} \ell_i(x^{(k+1)}) - \sum_{i=1}^{n} \lambda_i^{(k)} \ell_i(x^{(k)})$$
$$\leq \gamma \|h_\lambda^{(k)}\| \sqrt{n} B_\ell + \frac{1}{2c} \|\nabla_x F(x^{(k)}, \lambda^{(k)}) - \overline{\nabla}_x F(x^{(k)}, \boldsymbol{y}^{(k+1)}, \lambda^{(k)}) - B_k\|^2$$
$$+ \left( \frac{c+d+L_f}{2} - \frac{1}{2\alpha} \right) \|x^{(k+1)} - x^{(k)}\|^2 + \frac{\sigma_f^2}{2d}. \tag{82}$$

Next, we work on bounding $\|\nabla_x F(x^{(k)}, \lambda^{(k)}) - \overline{\nabla}_x F(x^{(k)}, \boldsymbol{y}^{(k+1)}, \lambda^{(k)}) - B_k\|^2$ in equation 81. Observe that

$$\|\nabla_x F(x^{(k)}, \lambda^{(k)}) - \overline{\nabla}_x F(x^{(k)}, \boldsymbol{y}^{(k+1)}, \lambda^{(k)}) - B_k\|^2$$

$$\overset{(1)}{\leq} 2\|\nabla_x F(x^{(k)}, \lambda^{(k)}) - \overline{\nabla}_x F(x^{(k)}, \boldsymbol{y}^{(k+1)}, \lambda^{(k)})\|^2 + 2\|B_k\|^2 \tag{83}$$

$$\overset{(2)}{\leq} 2\|\nabla_x F(x^{(k)}, \lambda^{(k)}) - \overline{\nabla}_x F(x^{(k)}, \boldsymbol{y}^{(k+1)}, \lambda^{(k)})\|^2 + 2b_k^2 \tag{84}$$

$$\overset{(3)}{\leq} 2\left\|\sum_{i=1}^n \left[\lambda_i^{(k)}\left(\nabla\ell_i(x^{(k)}) - \overline{\nabla}_x f_i(x^{(k)}, y_i^{(k+1)})\right)\right]\right\|^2 + 2b_k^2 \tag{85}$$

$$\overset{(4)}{\leq} 2\left(\sum_{i=1}^n \lambda_i^{(k)} L\|y_i^\star(x^{(k)}) - y_i^{(k+1)}\|\right)^2 + 2b_k^2 \tag{86}$$

$$\overset{(5)}{\leq} 2L^2 \max_{i\in[n]}\|y_i^\star(x^{(k)}) - y_i^{(k+1)}\|^2 + 2b_k^2, \tag{87}$$

where (1) comes from $\|a + b\|^2 \leq 2\|a\|^2 + 2\|b\|^2$, (2) comes from $\|B_k\| \leq b_k$ in Assumption 3, (3) is from expanding the definitions of $\nabla_x F(\cdot, \cdot)$ and $\overline{\nabla}_x F(\cdot, \cdot, \cdot)$, (4) is from Lemma 5. Therefore, plugging in equation 87 into equation 82, using $c = d = \frac{1}{4\alpha}$, and taking expectation over the stochasticity of the gradient estimates, we get:

$$\sum_{i=1}^n \lambda_i^{(k+1)}\ell_i(x^{(k+1)}) - \sum_{i=1}^n \lambda_i^{(k)}\ell_i(x^{(k)})$$

$$\leq \sqrt{n}\gamma\|h_\lambda^{(k)}\|B_\ell + 4\alpha L^2 \max_{i\in[n]}\mathbb{E}\|y_i^\star(x^{(k)}) - y_i^{(k+1)}\|^2$$

$$+ 4\alpha b_k^2 + \left(\frac{L_f}{2} - \frac{1}{4\alpha}\right)\mathbb{E}[\|x^{(k+1)} - x^{(k)}\|^2] + 2\alpha\sigma_f^2. \tag{88}$$

Now, observe that the LHS looks like a telescoping sum. To make this more apparent, define $\mathcal{L}^{(k)} = \sum_{i=1}^n \lambda_i^{(k)}\mathbb{E}[\ell_i(x^{(k)})]$ and $\Delta_{y_i}^{(k)} = \mathbb{E}[\|y_i^{(k)} - y_i^\star(x^{(k-1)})\|^2]$. Therefore, with the assumption that $b_k \leq b_0$, we have

$$\mathcal{L}^{(k+1)} - \mathcal{L}^{(k)}$$

$$\leq 4\alpha L^2 \max_{i\in[n]}\Delta_{y_i}^{(k+1)} + \left(\frac{L_f}{2} - \frac{1}{4\alpha}\right)\mathbb{E}[\|x^{(k+1)} - x^{(k)}\|^2] + \sqrt{n}\gamma\|h_\lambda^{(k)}\|B_\ell + 4\alpha b_0^2 + 2\alpha\sigma_f^2$$

$$\leq 4\alpha L^2 \max_{i\in[n]}\Delta_{y_i}^{(k+1)} + \left(\frac{L_f}{2} - \frac{1}{4\alpha}\right)\mathbb{E}[\|x^{(k+1)} - x^{(k)}\|^2] + \gamma n B_\ell^2 + 4\alpha b_0^2 + 2\alpha\sigma_f^2. \tag{89}$$

$$\square$$

### A.7 Proof of Lemma 9

We restate Lemma 9 in more general terms here:

**Lemma 15.** *Assume that* $\Omega^{(k)}, \Theta^{(k)}, \Upsilon_i^{(k)}, \lambda_i, c_0, c_1, c_2, d_0, d_1, d_2$ *are real numbers such that for all* $0 \leq k \leq K - 1$,

$$\Omega^{(k+1)} \leq \Omega^{(k)} - c_0\Theta^{(k+1)} + c_1 \max_{i\in[n]}\Upsilon_i^{(k+1)} + c_2 \tag{90}$$

*and also for all* $1 \leq k \leq K, 1 \leq i \leq N$,

$$\Upsilon_i^{(k+1)} \leq (1 - d_0)\Upsilon_i^{(k)} + d_1\Theta^{(k)} + d_2. \tag{91}$$

*In addition, assume that $1 - d_0 > 0$, $d_0 - d_1 c_1 c_0^{-1} > 0$ and $c_0 - c_1 d_1 d_0^{-1}$, and that $\Upsilon_i^{(k)}, \Omega^{(k)} \geq 0$ for all $k, i \in [n]$. Then, if $i^{(0)} = \arg\max_{i \in [n]} \Upsilon_i^{(0)}$, we have*

$$\frac{1}{K} \sum_{k=1}^{K} \max_{i \in [n]} \Upsilon_i^{(k)} \leq (d_0 - d_1 c_0^{-1} c_1)^{-1} \left( \frac{\Upsilon_{i^{(0)}}^{(0)} + d_1 \Theta^{(0)} + d_2 + d_1 c_0^{-1} \Omega^{(0)}}{K} + d_1 c_0^{-1} c_2 + d_2 \right).$$

(92)

*Proof.* First, let $i^{(k)} \triangleq \arg\max_{i \in [n]} \Upsilon_i^{(k)}$, so that $\Upsilon_{i^{(k)}}^{(k)} = \max_{i \in [n]} \Upsilon_i^{(k)}$. Summing (90) from $k = 0, 1, \cdots, K-1$, we get:

$$c_0 \sum_{k=1}^{K} \Theta^{(k)} \leq \Omega^{(0)} - \Omega^{(k)} + c_1 \sum_{k=1}^{K} \Upsilon_{i^{(k)}}^{(k)} + c_2 K.$$

(93)

Next, we apply (91) for $i = i^{(k+1)}$. Noting that $1 - d_0 > 0$ and $\Upsilon_{i^{(k+1)}}^{(k)} \leq \Upsilon_{i^{(k)}}^{(k)}$ by definition of $i^{(k)}$, we have

$$\Upsilon_{i^{(k+1)}}^{(k+1)} \leq (1 - d_0) \Upsilon_{i^{(k+1)}}^{(k)} + d_1 \Theta^{(k)} + d_2$$

(94)

$$\leq (1 - d_0) \Upsilon_{i^{(k)}}^{(k)} + d_1 \Theta^{(k)} + d_2.$$

(95)

Then summing for $k = 1$ to $K$, we get

$$d_0 \sum_{k=1}^{K} \Upsilon_{i^{(k)}}^{(k)} \leq \Upsilon_{i^{(1)}}^{(1)} - \Upsilon_{i^{(K+1)}}^{(K+1)} + d_1 \sum_{k=1}^{K} \Theta^{(k)} + d_2 K.$$

(96)

Now, we have

$$d_0 \sum_{k=1}^{K} \Upsilon_{i^{(k)}}^{(k)} \overset{(1)}{\leq} \Upsilon_{i^{(1)}}^{(1)} - \Upsilon_{i^{(K+1)}}^{(K+1)} + d_1 c_0^{-1} \left( \Omega^{(0)} - \Omega^{(k)} + c_1 \sum_{k=1}^{K} \Upsilon_{i^{(k)}}^{(k)} + c_2 K \right) + d_2 K$$

(97)

$$\overset{(2)}{\leq} \Upsilon_{i^{(1)}}^{(1)} + d_1 c_0^{-1} \left( \Omega^{(0)} + c_1 \sum_{k=1}^{K} \Upsilon_{i^{(k)}}^{(k)} + c_2 K \right) + d_2 K$$

(98)

$$\overset{(3)}{\leq} \Upsilon_{i^{(1)}}^{(1)} + d_1 c_0^{-1} \Omega^{(0)} + d_1 c_0^{-1} c_1 \sum_{k=1}^{K} \Upsilon_{i^{(k)}}^{(k)} + d_1 c_0^{-1} c_2 K + d_2 K,$$

(99)

where (1) holds from plugging (93) into (96), (2) is true because $\Upsilon, \Omega \geq 0$, and (3) follows from the distributive property. We can rewrite this equation as

$$(d_0 - d_1 c_0^{-1} c_1) \sum_{k=1}^{K} \Upsilon_{i^{(k)}}^{(k)} \leq \Upsilon_{i^{(1)}}^{(1)} + d_1 c_0^{-1} \Omega^{(0)} + d_1 c_0^{-1} c_2 K + d_2 K$$

(100)

$$\Rightarrow \frac{1}{K} \sum_{k=1}^{K} \Upsilon_{i^{(k)}}^{(k)} \leq (d_0 - d_1 c_0^{-1} c_1)^{-1} \left( \frac{\Upsilon_{i^{(1)}}^{(1)} + d_1 c_0^{-1} \Omega^{(0)}}{K} + d_1 c_0^{-1} c_2 + d_2 \right).$$

(101)

By plugging in $\Upsilon_{i^{(1)}}^{(1)} = \Upsilon_{i^{(0)}}^{(0)} + d_1 \Theta^{(0)} + d_2$ into (101), we get the statement of the lemma. $\qquad \square$

Plugging the following values into Lemma 15 and utilizing Lemmas 13 and 14 and the learning rates $\alpha, \beta, \gamma$ from Theorem 2, we get the result in Lemma 9 in precise terms:

$$\Omega^{(k)} = \mathcal{L}^{(k)}, \qquad \Theta^{(k)} = \mathbb{E}[\|x^{(k)} - x^{(k-1)}\|^2], \qquad \Upsilon_i^{(k)} = \Delta_{y_i}^{(k)},$$

$$c_0 = \frac{1}{4\alpha} - \frac{L_f}{2}, \qquad c_1 = 4\alpha L^2, \qquad c_2 = \gamma n B_\ell^2 + 4\alpha b_0^2 + 2\alpha \sigma_f^2,$$

$$d_0 = \mu_g \beta / 2, \qquad d_1 = \left( \frac{2}{\mu_g \beta} - 1 \right) G_y^2, \qquad d_2 = \beta^2 \sigma_g^2.$$

(102)

Next, recall that our step sizes were

$$\alpha = \min\left(\frac{\mu_g}{16G_yL}\nu, \frac{K^{-3/5}}{4G_yL}\right), \beta = \min\left(\nu, \frac{4K^{-2/5}}{\mu_g}\right), \gamma = \frac{2K^{-3/5}}{B_\ell n^{1/2}}, \tag{103}$$

where $\nu = \min(\frac{\mu_g}{L_g^2(1+\sigma_g^2)}, \frac{1}{\mu_g})$. Note that the choice of $\nu$ was motivated by the conditions of Lemma 5. First, observe that $1 - d_0 > 0$ is true because we chose $\beta < 2/\mu_g$. Now, observe that $\frac{\alpha}{\beta} \leq \frac{\mu_g}{16G_yL}$. Finally, will next show that $d_0 - d_1c_1(c_0)^{-1} > 0$ and $c_0 - c_1d_1(d_0)^{-1} > 0$, completing the set of conditions in Lemma 5. By direct algebraic manipulation, we have

$$\begin{aligned}
d_0 - d_1c_1(c_0)^{-1} &= \frac{\mu_g\beta}{2} - \frac{\left(\frac{2}{\mu_g\beta} - 1\right)G_y^2 \cdot 4L^2\alpha}{\left(\frac{1}{4\alpha} - \frac{L_f}{2}\right)} \geq \frac{\mu_g\beta}{2} - \frac{\frac{2}{\mu_g\beta} \cdot G_y^2 \cdot 4L^2\alpha}{\left(\frac{1}{4\alpha} - \frac{L_f}{2}\right)} \\
&\geq \frac{\mu_g\beta}{2} - \frac{8L^2G_y^2\alpha}{\beta\mu_g\left(\frac{1}{4\alpha} - \frac{L_f}{2}\right)} \geq \frac{\mu_g\beta}{2} - \frac{64L^2G_y^2}{\mu_g^2} \cdot \frac{\alpha^2}{\beta^2} \cdot \mu_g\beta \\
&\geq \frac{\mu_g\beta}{2} - \frac{64L^2G_y^2}{\mu_g^2} \cdot \frac{\mu_g^2}{256G_y^2L^2} \cdot \mu_g\beta = \frac{\mu_g\beta}{2} - \frac{\mu_g\beta}{4} = \frac{\mu_g\beta}{4}.
\end{aligned} \tag{104}$$

Similarly, we also have

$$\begin{aligned}
&c_0 - c_1d_1(d_0)^{-1} \\
&= \left(\frac{1}{4\alpha} - \frac{L_f}{2}\right) - \frac{4L^2\alpha \cdot \left(\frac{2}{\mu_g\beta} - 1\right)G_y^2}{\frac{\mu_g\beta}{2}} = \left(\frac{1}{4\alpha} - \frac{L_f}{2}\right) - \frac{8L^2\alpha\left(\frac{2}{\mu_g\beta} - 1\right)G_y^2}{\mu_g\beta} \\
&\geq \left(\frac{1}{4\alpha} - \frac{L_f}{2}\right) - \frac{8L^2\alpha\left(\frac{2}{\mu_g\beta}\right)G_y^2}{\mu_g\beta} \geq \left(\frac{1}{4\alpha} - \frac{L_f}{2}\right) - \frac{16L^2\alpha G_y^2}{\mu_g^2} \cdot \frac{\alpha^2}{\beta^2} \cdot \frac{1}{\alpha} \\
&\geq \frac{1}{8\alpha} - \frac{16L^2\alpha G_y^2}{\mu_g^2} \cdot \frac{\mu_g^2}{256G_y^2L^2} \cdot \frac{1}{\alpha} = \frac{1}{8\alpha} - \frac{1}{16\alpha} = \frac{1}{16\alpha}.
\end{aligned} \tag{105}$$

Now, we can bound the optimality of $y$ by bounding the maximum difference $\Delta_{y_i}^{(k)}$:

$$
\frac{1}{K} \sum_{k=1}^{K} \max_{i \in [n]} \Delta_{y_i}^{(k)}
$$

$$
\stackrel{(1)}{\leq} (d_0 - d_1 c_0^{-1} c_1)^{-1} \left( \frac{\Upsilon_{i^{(0)}}^{(0)} + d_1 \Theta^{(0)} + d_2 + d_1 c_0^{-1} \Omega^{(0)}}{K} + d_1 c_0^{-1} c_2 + d_2 \right) \tag{106}
$$

$$
\stackrel{(2)}{=} \frac{4}{\mu_g \beta} \left( \frac{\Upsilon_{i^{(0)}}^{(0)} + d_1 \Theta^{(0)} + d_2 + d_1 c_0^{-1} \Omega^{(0)}}{K} + d_1 c_0^{-1} c_2 + d_2 \right) \tag{107}
$$

$$
\stackrel{(3)}{\leq} \frac{4}{\mu_g \beta} \left( \frac{\Delta_{y_{i^{(0)}}}^{(0)} + \beta^2 \sigma_g^2 + \frac{2 G_y^2}{\mu_g \beta}(8\alpha) \mathcal{L}^{(0)}}{K} \right.
$$

$$
\left. + \frac{2 G_y^2}{\mu_g \beta}(8\alpha)(n \gamma B_\ell^2 + 4\alpha b_0^2 + 2\alpha \sigma_f^2) + \beta^2 \sigma_g^2 \right) \tag{108}
$$

$$
\stackrel{(4)}{=} \frac{\frac{4 \Delta_{y_{i^{(0)}}}^{(0)}}{\mu_g \beta} + 4\beta \frac{\sigma_g^2}{\mu_g} + \frac{64 G_y^2 \alpha}{\mu_g^2 \beta^2}}{K} + \frac{64 G_y^2 \alpha}{\mu_g^2 \beta^2} \left( n \gamma B_\ell + 4\alpha b_0^2 + 2\alpha \sigma_f^2 \right) + \frac{4\beta \sigma_g^2}{\mu_g} \tag{109}
$$

$$
\stackrel{(5)}{=} \frac{4 \Delta_{y_{i^{(0)}}}^{(0)}}{\mu_g} \frac{1}{\beta K} + \frac{4 \sigma_g^2}{\mu_g} \frac{\beta}{K} + \frac{64 G_y^2}{\mu_g^2} \frac{\alpha}{\beta^2 K} + \frac{64 G_y^2}{\mu_g^2} \gamma B_\ell^2 \frac{n\alpha}{\beta^2}
$$

$$
+ \frac{64 G_y^2 (4 b_0^2 + 2\sigma_f^2)}{\mu_g^2} \frac{\alpha^2}{\beta^2} + \frac{4\sigma_g^2}{\mu_g} \beta \tag{110}
$$

$$
\stackrel{(6)}{\leq} \frac{\Delta_{y_{i^{(0)}}}^{(0)}/\mu_g}{K^{3/5}} + \frac{16 \sigma_g^2/\mu_g^2}{K^{7/5}} + \frac{G_y/L}{K^{4/5}} + \frac{2\sqrt{n} B_\ell G_y/L}{K^{2/5}} + \frac{(b_0^2 + \frac{1}{2}\sigma_f^2)/(L^2)}{K^{2/5}} + \frac{16 \sigma_g^2/\mu_g^2}{K^{2/5}}. \tag{111}
$$

Here, (1) follows directly from plugging $\Upsilon_i^{(k)}$ from (102) into Lemma 4, (2) follows from (104), (3) comes from plugging in the rest of (102), (4) is direct algebraic manipulation, (5) separates the step sizes and $n, K$ factors from the rest of the constants, and (6) applies the definition of the step sizes. This gives us the $\mathcal{O}(\sqrt{n} K^{-2/5})$ bound in Lemma 9.

## A.8 PROOF OF LEMMA 10

We present a precise form of Lemma 10 here:

**Lemma 16.** *For any $\lambda \in \Delta_n$, under Assumptions 1, 2, and 3, assume that the iterates $\{x^{(k)}, y_i^{(k)}, i \in [n] \lambda^{(k)}, \forall k\}$ generated by MORBiT, then we have*

$$
\frac{1}{K} \mathbb{E} \left[ \sum_{k=1}^{K} F(x^{(k)}, \lambda) - F(x^{(k)}, \lambda^{(k)}) \right]
$$

$$
\leq \frac{1}{\sqrt{2}} \left( B_\ell \sqrt{n} K^{-3/5} + B_\ell \sqrt{n} K^{-2/5} \right) + \frac{G_f}{K} \sum_{k=1}^{K} \max_{i \in [n]} \Delta_{y_i}^{(k)}. \tag{112}
$$

*Proof.* Recall that we defined

$$
F(x, \lambda) := \sum_{i=1}^{n} \lambda_i f_i(x, y_i^\star(x)) = \sum_{i=1}^{n} \lambda_i \ell_i(x). \tag{113}
$$

For a fixed realization of $x^{(1)}, \cdots, x^{(k)}, y_i^{(1)}, \cdots, y_i^{(k)}, i \in [n]$, we have

$F(x^{(k)}, \lambda) - F(x^{(k)}, \lambda^{(k)})$

$$\overset{(1)}{=} \sum_{i=1}^{n} (\lambda_i - \lambda_i^{(k)}) f_i(x^{(k)}, y_i^\star(x^{(k)})) \tag{114}$$

$$\overset{(2)}{=} \sum_{i=1}^{n} (\lambda_i - \lambda_i^{(k)})(f_i(x^{(k)}, y_i^\star(x^{(k)})) - f_i(x^{(k)}, y_i^{(k+1)}) + f_i(x^{(k)}, y_i^{(k+1)}))) \tag{115}$$

$$\overset{(3)}{=} \left\langle \left(\lambda - \lambda^{(k)}\right), \left[f_1(x^{(k)}, y_1^{(k+1)}), \cdots, f_n(x^{(k)}, y_n^{(k+1)})\right]^\top \right\rangle$$
$$+ \sum_{i=1}^{n} (\lambda_i - \lambda_i^{(k)})(f_i(x^{(k)}, y_i^\star(x^{(k)})) - f_i(x^{(k)}, y_i^{(k+1)})) \tag{116}$$

$$\overset{(4)}{=} \left\langle \left(\lambda - \lambda^{(k)}\right), h_\lambda^{(k)} \right\rangle + \sum_{i=1}^{n} (\lambda_i - \lambda_i^{(k)})(f_i(x^{(k)}, y_i^\star(x^{(k)})) - f_i(x^{(k)}, y_i^{(k+1)})) \tag{117}$$

$$\overset{(5)}{=} \frac{\|\lambda - \lambda^{(k)}\|^2 + \gamma^2 \|h_\lambda^{(k)}\|^2 - \|\lambda - \lambda^{(k)} - \gamma h_\lambda^{(k)}\|^2}{2\gamma}$$
$$+ \sum_{i=1}^{n} (\lambda_i - \lambda_i^{(k)})(f_i(x^{(k)}, y_i^\star(x^{(k)})) - f_i(x^{(k)}, y_i^{(k+1)})) \tag{118}$$

$$\overset{(6)}{\leq} \frac{\|\lambda - \lambda^{(k)}\|^2 + \gamma^2 \|h_\lambda^{(k)}\|^2 - \|\lambda - \lambda^{(k+1)}\|^2}{2\gamma}$$
$$+ \sum_{i=1}^{n} (\lambda - \lambda^{(k)})_i(f_i(x^{(k)}, y_i^\star(x^{(k)})) - f_i(x^{(k)}, y_i^{(k+1)})) \tag{119}$$

$$\overset{(7)}{\leq} \frac{\|\lambda - \lambda^{(k)}\|^2 + \gamma^2 \|h_\lambda^{(k)}\|^2 - \|\lambda - \lambda^{(k+1)}\|^2}{2\gamma} + G_f \sum_{i=1}^{n} (\lambda_i - \lambda_i^{(k)})(y_i^\star(x^{(k)}) - y_i^{(k+1)}), \tag{120}$$

where (1) comes from the definition of $F$, (2) follows from adding and subtracting $(\lambda_i - \lambda_i^{(k)}) f_i(x^{(k)}, y_i^{(k+1)})$ terms, (3) follows from splitting the preceding sum and writing the first term as a dot product, (4) follows from definition of $h_\lambda^{(k)}$, (5) uses $\langle a, b \rangle = \frac{\|a\|^2 + \|b\|^2 - \|a-b\|^2}{2}$, (6) follows from the update $\lambda^{(k+1)} = \text{proj}_{\Delta_n}(\lambda^{(k)} - \gamma h_\lambda^{(k)})$ and the projection property, and (7) follows from Lipschitzness of $f$.

Therefore, applying the telescoping sum by adding the preceding inequality over $k = 1, 2, \ldots, K$, and taking expectation, we get:

$$\mathbb{E}\left[\sum_{k=1}^{K} F(x^{(k)}, \lambda) - F(x^{(k)}, \lambda^{(k)})\right]$$

$$\overset{(1)}{\leq} \frac{\gamma}{2} \sum_{k=1}^{K} \mathbb{E}\|h_\lambda^{(k)}\|^2 + \frac{\mathbb{E}[\|\lambda - \lambda^{(1)}\|^2]}{2\gamma} + G_f \sum_{k=1}^{K} \sum_{i=1}^{n} (\lambda_i - \lambda_i^{(k)}) \Delta_{y_i}^{(k)} \tag{121}$$

$$\overset{(2)}{\leq} \frac{\gamma}{2} \sum_{k=1}^{K} \mathbb{E}\|h_\lambda^{(k)}\|^2 + \frac{1}{\gamma} + G_f \sum_{k=1}^{K} \max_{i \in [n]} \Delta_{y_i}^{(k)} \tag{122}$$

$$\overset{(3)}{\leq} \frac{nKB^2\gamma}{2} + \frac{1}{\gamma} + G_f \sum_{k=1}^{K} \max_{i \in [n]} \Delta_{y_i}^{(k)} \tag{123}$$

$$\overset{(4)}{\leq} \frac{\sqrt{2}}{2}\left(B_\ell \sqrt{n} K^{2/5} + B_\ell \sqrt{n} K^{3/5}\right) + G_f \sum_{k=1}^{K} \max_{i \in [n]} \Delta_{y_i}^{(k)} \tag{124}$$

where (1) follows directly from (120) and the telescoping sum, (2) follows from $\lambda \in \Delta_n$ and $\|\lambda - \lambda^{(1)}\|^2 \leq 2$, (3) follows from $\|h_\lambda^{(k)}\|^2 \leq nB^2$, and (4) follows from selecting $\gamma = \frac{\sqrt{2}}{B_\ell \sqrt{n} K^{3/5}}$. $\qquad \square$

Therefore, we obtain

$$
\begin{aligned}
&\frac{1}{K}\mathbb{E}\left[\sum_{k=1}^{K} F(x^{(k)}, \lambda) - F(x^{(k)}, \lambda^{(k)})\right] \\
&\leq \frac{1}{\sqrt{2}}\left(B_\ell\sqrt{n}K^{-3/5} + B_\ell\sqrt{n}K^{-2/5}\right) + G_f\mathcal{O}(\sqrt{n}K^{-2/5}).
\end{aligned}
\tag{125}
$$

### A.9 PROOF OF LEMMA 11

We state Lemma 11 here in precise terms:

**Lemma 17.** *Under Assumptions 1, 2, and 3 with the iterates $\{x^{(k)}, y_i^{(k)}, i \in [n], \lambda^{(k)}, \forall k\}$ generated by* MORBiT*, then we have*

$$
\begin{aligned}
&\frac{1}{K}\sum_{k=1}^{K}\mathbb{E}[\|\hat{x}(x^{(k)}) - x^{(k)}\|^2] \\
&\leq \frac{4}{-\mu_\ell + \rho}\left(\frac{\Phi_{1/\rho}(x_0)}{\rho\alpha K} + \frac{2}{-\mu_\ell + \rho}b_0^2 + \right. \\
&\qquad\left.\left(\frac{2L^2}{-\mu_\ell + \rho} + \frac{3L^2\alpha}{2}\right)\left(\frac{1}{K}\sum_{k=1}^{K}\max_{i\in[n]}\Delta_{y_i}^{(k)}\right) + \frac{\alpha}{2}\left(\tilde{\sigma}_f^2 + 3b_0^2\right)\right).
\end{aligned}
\tag{126}
$$

*Proof.* Recall that we defined the Moreau envelope and proximal map as follows:

$$
\Phi_{1/\rho}(z) \triangleq \min_x \frac{\rho}{2}\|x - z\|^2 + \sum_{i=1}^{n}\lambda_i\ell_i(x), \qquad \hat{x}(z) \triangleq \arg\min_x \frac{\rho}{2}\|x - z\|^2 + \sum_{i=1}^{n}\lambda_i\ell_i(x). \tag{127}
$$

Therefore, we have

$$
\Phi_{1/\rho}(x^{(k+1)}) \overset{(1)}{=} \sum_i \lambda_i\ell_i(\hat{x}(x^{(k+1)})) + \frac{\rho}{2}\|x^{(k+1)} - \hat{x}(x^{(k+1)})\|^2 \tag{128}
$$

$$
\overset{(2)}{\leq} \sum_i \lambda_i\ell_i(\hat{x}(x^{(k)})) + \frac{\rho}{2}\|x^{(k+1)} - \hat{x}(x^{(k)})\|^2 \tag{129}
$$

$$
\overset{(3)}{=} \sum_i \lambda_i\ell_i(\hat{x}(x^{(k)})) + \frac{\rho}{2}\|x^{(k+1)} - x^{(k)} + x^{(k)} - \hat{x}(x^{(k)})\|^2 \tag{130}
$$

$$
\begin{aligned}
\overset{(4)}{=} &\sum_i \lambda_i\ell_i(\hat{x}(x^{(k)})) + \frac{\rho}{2}\|x^{(k+1)} - x^{(k)}\|^2 + \frac{\rho}{2}\|x^{(k)} - \hat{x}(x^{(k)})\|^2 \\
&+ \rho\left\langle\left(x^{(k+1)} - x^{(k)}\right), \left(x^{(k)} - \hat{x}(x^{(k)})\right)\right\rangle
\end{aligned}
\tag{131}
$$

where (1) is by definition of the proximal map, (2) comes from the optimality of the Moreau envelope, (3) is by adding and subtracting $x^{(k)}$, and (4) is from expanding out $\|a+b\|^2$ into $\|a\|^2 + \|b\|^2 + 2\langle a, b\rangle$.

Next, from the optimality condition of the update $x^{(k+1)} = \text{proj}_{\Delta_n}(x^{(k)} - \alpha h_x^{(k)})$, we have

$$\langle x^{(k+1)} - \hat{x}(x^{(k)}), x^{(k+1)} - x^{(k)} + \alpha h_x^{(k)} \rangle \leq 0 \tag{132}$$

$$\overset{(1)}{\Rightarrow} \langle x^{(k+1)} - x^{(k)} + x^{(k)} - \hat{x}(x^{(k)}) x^{(k+1)} - x^{(k)} + \alpha h_x^{(k)} \rangle \leq 0 \tag{133}$$

$$\overset{(2)}{\Rightarrow} \langle x^{(k+1)} - x^{(k)}, x^{(k)} - \hat{x}(x^{(k)}) \rangle$$
$$\leq -\|x^{(k+1)} - x^{(k)}\|^2 - \alpha \langle h_x^{(k)}, x^{(k+1)} - x^{(k)} \rangle - \alpha \langle h_x^{(k)}, x^{(k)} - \hat{x}(x^{(k)}) \rangle \tag{134}$$

$$\overset{(3)}{\Rightarrow} \rho \langle x^{(k+1)} - x^{(k)}, x^{(k)} - \hat{x}(x^{(k)}) \rangle$$
$$\leq -\rho \|x^{(k+1)} - x^{(k)}\|^2 - \rho\alpha \langle h_x^{(k)}, x^{(k+1)} - x^{(k)} \rangle - \rho\alpha \langle h_x^{(k)}, x^{(k)} - \hat{x}(x^{(k)}) \rangle \tag{135}$$

$$\overset{(4)}{\Rightarrow} \rho \langle x^{(k+1)} - x^{(k)}, x^{(k)} - \hat{x}(x^{(k)}) \rangle$$
$$\leq -\rho \|x^{(k+1)} - x^{(k)}\|^2 + \rho \langle \alpha h_x^{(k)}, x^{(k)} - x^{(k+1)} \rangle + \rho\alpha \langle h_x^{(k)}, \hat{x}(x^{(k)}) - x^{(k)} \rangle \tag{136}$$

$$\overset{(5)}{\Rightarrow} \rho \langle x^{(k+1)} - x^{(k)}, x^{(k)} - \hat{x}(x^{(k)}) \rangle$$
$$\leq -\rho \|x^{(k+1)} - x^{(k)}\|^2 + \frac{\rho}{2} \left( \|\alpha h_x^{(k)}\|^2 + \|x^{(k)} - x^{(k+1)}\|^2 \right)$$
$$+ \rho\alpha \langle h_x^{(k)}, \hat{x}(x^{(k)}) - x^{(k)} \rangle \tag{137}$$

$$\overset{(6)}{\Rightarrow} \rho \langle x^{(k+1)} - x^{(k)}, x^{(k)} - \hat{x}(x^{(k)}) \rangle$$
$$\leq -\frac{\rho}{2} \|x^{(k+1)} - x^{(k)}\|^2 + \frac{\rho\alpha^2}{2} \|h_x^{(k)}\|^2 + \rho\alpha \langle h_x^{(k)}, \hat{x}(x^{(k)}) - x^{(k)} \rangle \tag{138}$$

where (1) is by adding and subtracting $x^{(k)}$, (2) is from distributing the inner product $\langle (x^{(k+1)} - x^{(k)}) + (x^{(k)} - \hat{x}(x^{(k)})), (x^{(k+1)} - x^{(k)}) + \alpha h_x^{(k)} \rangle$, (3) is from multiplying both sides by $\rho$, (4) is from simple algebra, (5) is from rewriting $\langle a, b \rangle \leq \frac{\|a\|^2 + \|b\|^2}{2}$, and (6) is from combining terms. Therefore, substituting (138) into (131), we get

$$\Phi_{1/\rho}(x^{(k+1)})$$
$$\leq \sum_i \lambda_i \ell_i(\hat{x}(x^{(k)})) + \frac{\rho}{2} \|x^{(k)} - \hat{x}(x^{(k)})\|^2 + \frac{\rho\alpha^2}{2} \|h_x^{(k)}\|^2 + \rho\alpha \langle h_x^{(k)}, \hat{x}(x^{(k)}) - x^{(k)} \rangle \tag{139}$$

$$= \Phi_{1/\rho}(x^{(k)}) + \frac{\rho\alpha^2}{2} \|h_x^{(k)}\|^2 + \rho\alpha \langle h_x^{(k)}, \hat{x}(x^{(k)}) - x^{(k)} \rangle \tag{140}$$

where the second equality is by definition of the Moreau envelope. Now, we bound the last term in (140):

$$\langle h_x^{(k)}, \hat{x}(x^{(k)}) - x^{(k)} \rangle$$
$$\overset{(1)}{=} \left\langle \hat{x}(x^{(k)}) - x^{(k)}, h_x^{(k)} - \overline{\nabla}_x F(x^{(k)}, \boldsymbol{y}^{(k+1)}, \lambda^{(k)}) \right\rangle$$
$$+ \left\langle \hat{x}(x^{(k)}) - x^{(k)}, \overline{\nabla}_x F(x^{(k)}, \boldsymbol{y}^{(k+1)}, \lambda^{(k)}) - \nabla_x F(x^{(k)}, \lambda^{(k)}) \right\rangle$$
$$+ \left\langle \hat{x}(x^{(k)}) - x^{(k)}, \nabla_x F(x^{(k)}, \lambda^{(k)}) \right\rangle \tag{141}$$

$$\overset{(2)}{=} \underbrace{\langle \hat{x}(x^{(k)}) - x^{(k)}, B_k \rangle}_{(A)} + \underbrace{\langle \hat{x}(x^{(k)}) - x^{(k)}, \overline{\nabla}_x F(x^{(k)}, \boldsymbol{y}^{(k+1)}, \lambda^{(k)}) - \nabla_x F(x^{(k)}, \lambda^{(k)}) \rangle}_{(B)}$$
$$+ \underbrace{\langle \hat{x}(x^{(k)}) - x^{(k)}, \nabla_x F(x^{(k)}, \lambda^{(k)}) \rangle}_{(C)} \tag{142}$$

where (1) follows from adding and subtracting $\overline{\nabla}_x F(x^{(k)}, \boldsymbol{y}^{(k+1)}, \lambda^{(k)}), \nabla_x F(x^{(k)}, \lambda^{(k)})$ terms and (2) is from splitting the inner product and applying $h_x^{(k)} - \overline{\nabla}_x F(x^{(k)}, \boldsymbol{y}^{(k+1)}, \lambda^{(k)}) = B_k$ from equation 20 in Assumption 3. To bound $(A)$ and $(B)$, we simply apply $\langle a, b \rangle \leq \frac{c}{4} \|a\|^2 + \frac{1}{c} \|b\|^2$ to

both inner products:

$$(A) = \langle \hat{x}(x^{(k)}) - x^{(k)}, B_k \rangle \le \frac{c}{4}\|\hat{x}(x^{(k)}) - x^{(k)}\|^2 + \frac{1}{c}b_k^2 \tag{143}$$

$$(B) = \left\langle \left( \hat{x}(x^{(k)}) - x^{(k)} \right), \overline{\nabla}_x F(x^{(k)}, \boldsymbol{y}^{(k+1)}, \lambda^{(k)}) - \nabla_x F(x^{(k)}, \lambda^{(k)}) \right\rangle$$
$$\le \frac{1}{c}\|\overline{\nabla}_x F(x^{(k)}, \boldsymbol{y}^{(k+1)}, \lambda^{(k)}) - \nabla_x F(x^{(k)}, \lambda^{(k)})\|^2 + \frac{c}{4}\|\hat{x}(x^{(k)}) - x^{(k)}\|^2. \tag{144}$$

We proceed to bound $(C)$. First, from weak convexity of $\mu_\ell$, we have that for all $i \in [n]$,

$$\ell_i(\hat{x}(x^{(k)})) \ge \ell_i(x^{(k)}) + \langle \nabla \ell(x^{(k)}), \hat{x}(x^{(k)}) - x^{(k)} \rangle - \frac{\mu_\ell}{2}\|\hat{x}(x^{(k)}) - x^{(k)}\|^2. \tag{145}$$

Taking $\lambda_i$ times the $i$th of these equations, we get

$$\sum_{i=1}^n \lambda_i \ell_i(\hat{x}(x^{(k)})) \ge \sum_{i=1}^n \lambda_i \ell_i(x^{(k)}) + \left\langle \sum_{i=1}^n \nabla \ell_i(x^{(k)}), \hat{x}(x^{(k)}) - x^{(k)} \right\rangle - \frac{\mu_\ell}{2}\|\hat{x}(x^{(k)}) - x^{(k)}\|^2. \tag{146}$$

By definition of the Moreau envelope, we also have

$$\sum_{i=1}^n \lambda_i \ell_i(x^{(k)}) \ge \sum_{i=1}^n \lambda_i \ell_i(\hat{x}(x^{(k)})) + \frac{\rho}{2}\|\hat{x}(x^{(k)}) - x^{(k)}\|^2. \tag{147}$$

Adding (146) and (147), we have

$$\frac{\mu_\ell - \rho}{2}\|\hat{x}(x^{(k)}) - x^{(k)}\|^2 \ge \langle \nabla_x F(x^{(k)}, \lambda^{(k)}), \hat{x}(x^{(k)}) - x^{(k)} \rangle. \tag{148}$$

If we let $c = \frac{-\mu_\ell + \rho}{2}$ in (143) and (144), we can rewrite (142) as

$$\langle h_x^{(k)}, \hat{x}(x^{(k)}) - x^{(k)} \rangle$$
$$\le \frac{c}{2}\|\hat{x}(x^{(k)}) - x^{(k)}\|^2 + \frac{1}{c}b_k^2 + \frac{1}{c}\|\overline{\nabla}_x F(x^{(k)}, \boldsymbol{y}^{(k+1)}, \lambda^{(k)}) - \nabla_x F(x^{(k)}, \lambda^{(k)})\|^2$$
$$- \frac{-\mu_\ell + \rho}{2}\|\hat{x}(x^{(k)}) - x^{(k)}\|^2 \tag{149}$$

$$\le \frac{c}{2}\|\hat{x}(x^{(k)}) - x^{(k)}\|^2 + \frac{1}{c}b_k^2 + \frac{L^2}{c}\left( \sum_{i=1}^n \lambda_i \|y_i^\star(x^{(k)}) - y_i^{(k+1)}\|^2 \right)$$
$$- \frac{-\mu_\ell + \rho}{2}\|\hat{x}(x^{(k)}) - x^{(k)}\|^2 \tag{150}$$

$$\le \frac{2}{-\mu_\ell + \rho}b_k^2 + \frac{2L^2}{-\mu_\ell + \rho}\left( \sum_{i=1}^n \lambda_i \|y_i^\star(x^{(k)}) - y_i^{(k+1)}\|^2 \right) - \frac{-\mu_\ell + \rho}{4}\|\hat{x}(x^{(k)}) - x^{(k)}\|^2. \tag{151}$$

Taking the full expectation $\mathcal{F}_i \triangleq \{y_i^{(0)}, x^{(0)}, \cdots, y_i^{(k)}, x^{(k)}\}$, we have
$$\mathbb{E}[\langle h_x^{(k)}, \hat{x}(x^{(k)}) - x^{(k)} \rangle]$$
$$\le \frac{2}{-\mu_\ell + \rho}b_k^2 + \frac{2L^2}{-\mu_\ell + \rho}\mathbb{E}\left[ \sum_{i=1}^n \lambda_i \|y_i^\star(x^{(k)}) - y_i^{(k+1)}\|^2 \right] - \frac{-\mu_\ell + \rho}{4}\|\hat{x}(x^{(k)}) - x^{(k)}\|^2 \tag{152}$$

$$= \frac{2}{-\mu_\ell + \rho}b_k^2 + \frac{2L^2}{-\mu_\ell + \rho}\left[ \sum_{i=1}^n \lambda_i \mathbb{E}[\|y_i^\star(x^{(k)}) - y_i^{(k+1)}\|^2] \right] - \frac{-\mu_\ell + \rho}{4}\|\hat{x}(x^{(k)}) - x^{(k)}\|^2 \tag{153}$$

$$\le \frac{2}{-\mu_\ell + \rho}b_k^2 + \frac{2L^2}{-\mu_\ell + \rho}\left( \max_{i \in [n]} \mathbb{E}[\|y_i^\star(x^{(k)}) - y_i^{(k+1)}\|^2] \right) - \frac{-\mu_\ell + \rho}{4}\|\hat{x}(x^{(k)}) - x^{(k)}\|^2 \tag{154}$$

$$= \frac{2}{-\mu_\ell + \rho}b_k^2 + \frac{2L^2}{-\mu_\ell + \rho}\left( \max_{i \in [n]} \Delta_{y_i}^{(k)} \right) - \frac{-\mu_\ell + \rho}{4}\|\hat{x}(x^{(k)}) - x^{(k)}\|^2 \tag{155}$$

Therefore, rewriting everything into (146) and taking the full expectation over $\mathcal{F}_i$, we have (recall that the definition of $\Delta_{y_i}^{(k)}$ includes an expectation):

$$\mathbb{E}[\Phi_{1/\rho}(x^{(k+1)})]$$

$$\overset{(1)}{\leq} \mathbb{E}[\Phi_{1/\rho}(x^{(k)})] + \frac{\rho\alpha^2}{2}\mathbb{E}[\|h_x^{(k)}\|^2] + \rho\alpha\mathbb{E}[\langle h_x^{(k)}, \hat{x}(x^{(k)}) - x^{(k)}\rangle] \tag{156}$$

$$\overset{(2)}{=} \mathbb{E}[\Phi_{1/\rho}(x^{(k)})] + \frac{2\rho\alpha}{-\mu_\ell + \rho}b_k^2 + \frac{2L^2\rho\alpha}{-\mu_\ell + \rho}\left(\max_{i\in[n]}\Delta_{y_i}^{(k)}\right) - \frac{\rho\alpha(-\mu_\ell + \rho)}{4}\mathbb{E}[\|\hat{x}(x^{(k)}) - x^{(k)}\|^2]$$

$$+ \frac{\rho\alpha^2}{2}\left(\tilde{\sigma}_f^2 + 3b_k^2 + 3L^2\max_{i\in[n]}\Delta_{y_i}^{(k)}\right) \tag{157}$$

$$\overset{(3)}{=} \mathbb{E}[\Phi_{1/\rho}(x^{(k)})] + \frac{2\rho\alpha}{-\mu_\ell + \rho}b_k^2 + \left(\frac{2L^2\rho\alpha}{-\mu_\ell + \rho} + \frac{3L^2\rho\alpha^2}{2}\right)\left(\max_{i\in[n]}\Delta_{y_i}^{(k)}\right)$$

$$+ \frac{\rho\alpha(\mu_\ell - \rho)}{4}\mathbb{E}[\|\hat{x}(x^{(k)}) - x^{(k)}\|^2] + \frac{\rho\alpha^2}{2}(\tilde{\sigma}_f^2 + 3b_k^2), \tag{158}$$

where (1) is a copy of (140) and (2) is from (158), plugging in $\|h_x^{(k)}\|^2$ from Lemma 1, and doing the same expectation calculation from (152) to (155). Finally, (3) is combining terms via algebra. Summing up from $k = 0, 1, \cdots, K - 1$, we get the following bound:

$$\frac{1}{K}\sum_{k=1}^{K}\mathbb{E}[\|\hat{x}(x^{(k)}) - x^{(k)}\|^2]$$

$$\leq \frac{4}{-\mu_\ell + \rho}\left(\frac{\Phi_{1/\rho}(x_0)}{\rho\alpha K} + \frac{2}{-\mu_\ell + \rho}b_0^2\right.$$

$$\left. + \left(\frac{2L^2}{-\mu_\ell + \rho} + \frac{3L^2\alpha}{2}\right)\left(\frac{1}{K}\sum_{k=1}^{K}\max_{i\in[n]}\Delta_{y_i}^{(k)}\right) + \frac{\alpha}{2}\left(\tilde{\sigma}_f^2 + 3b_0^2\right)\right) \tag{159}$$

$$= \frac{4}{-\mu_\ell + \rho}\left(\underbrace{\frac{\Phi_{1/\rho}(x_0)}{\rho\alpha K}}_{K^{-2/5}} + \frac{2}{-\mu_\ell + \rho}b_0^2\right.$$

$$+ \underbrace{\frac{2L^2}{-\mu_\ell + \rho}\left(\frac{1}{K}\sum_{k=1}^{K}\max_{i\in[n]}\Delta_{y_i}^{(k)}\right)}_{\sqrt{n}K^{-2/5}}$$

$$\left. + \underbrace{\frac{\alpha}{2}\left(\tilde{\sigma}_f^2 + 3b_0^2 + \frac{3L^2}{K}\sum_{k=1}^{K}\max_{i\in[n]}\Delta_{y_i}^{(k)}\right)}_{K^{-3/5} + \sqrt{n}K^{-1}}\right). \tag{160}$$

$\square$

## B  GENERALIZATION BOUNDS

In addition to convergence, we also show the generalization abilities of the bilevel optimizer. The theorem in this section is inspired by Collins et al. (2020, Theorem 4), but our results hold for the fully general min-max multi-objective BLO setup while Collins et al. (2020) study a min-max multi-objective single-level problem. Assume that for a learning task $i$, we observe $m_i$ batches of train/test data, $D_{i,j}^{\mathrm{tr}}$ and $D_{i,j}^{\mathrm{v}}, j \in [m_i]$. Assume that each train and test batch has $K$ and $J$ input-output pairs, respectively, so that sets $D_{i,j}^{\mathrm{tr}}, D_{i,j}^{\mathrm{v}}$ are drawn from a common distribution $\mathcal{D}_i$. Also, let $y_i^\star(x; D_i^{\mathrm{tr}}) = \arg\min_{y_i} \frac{1}{m_i}\sum_{j'=1}^{m_i} g_i(x, y_i, D_{i,j'}^{\mathrm{tr}})$ be the value of $y_i$ that minimizes the empirical inner loss on some dataset $D_i^{\mathrm{tr}}$. For outer and inner objectives $f_i, g_i$, we consider the following

function class $\mathcal{F}_i$, where $x \in \mathcal{X}$ is the set of optimization parameters introduced in equation 2 and $(D_i^{\text{t}}, D_i^{\text{v}})$ are any train and test datasets sampled from $\mathcal{D}_i$:

$$\mathcal{F}_i = \left\{ f_i \left( x, \hat{y}_i^\star(x; D_i^{\text{t}}), D_i^{\text{v}} \right), x \in \mathcal{X} \right\}.$$

We use $f$ and $g$ to denote the empirical function values evaluated at the points in $D_i^{\text{t}}$ and $D_i^{\text{v}}$, and so the empirical Rademacher complexity of $\mathcal{F}_i$ on $m_i$ samples $\mathbf{D}_i \triangleq \{(D_{i,j}^{\text{t}}, D_{i,j}^{\text{v}}\}_{j=1}^{m_i} \sim (\mathcal{D}_i)^{m_i}$ is

$$\mathcal{R}_{m_i}^i(\mathcal{F}_i) = E_{\mathbf{D}_i} \mathbb{E}_{\epsilon_j} \left[ \sup_{x \in \mathcal{X}} \frac{1}{m_i} \sum_{j=1}^{m_i} \epsilon_j f_i \left( x, y_i^\star(x; D_{i,j}^{\text{t}}); D_{i,j}^{\text{v}} \right) \right],$$

where $\epsilon_j$s are Rademacher random variables ($\pm 1/2$ with equal probability). The empirical loss for fixed samples $D_{i,j}^{\text{t}}$ and $D_{i,j}^{\text{v}}$,

$$\hat{F}_i(x) \triangleq \frac{1}{m_i} \sum_{j=1}^{m_i} f_i \left( x, y_i^\star(x; D_{i,j}^{\text{t}}); D_{i,j}^{\text{v}} \right).$$

Similarly, define $F_i(x) \triangleq \mathbb{E}_{\mathbf{D}_i}[\hat{F}_i(x)]$. First, from classical generalization results such as Shalev-Shwartz & Ben-David (2014, Theorem 26.5) or Mohri et al. (2018, Theorem 3.3), we directly conclude the following proposition, which bounds the true loss of the classifier as a function of the empirical loss.

**Proposition 1.** *Assume the regularity assumptions considered in Appendix 3, specifically that the function $\ell$ is $B_\ell$-bounded. Then, with probability at least $1 - \delta$, we have*

$$F_i(x) \le \hat{F}_i(x) + 2\mathcal{R}_{m_i}^i(\mathcal{F}_i) + B_\ell \sqrt{\frac{\log 1/\delta}{2m_i}}. \tag{161}$$

This extends to the following in a straightforward manner, providing a guarantee for the worst-case generalization for any learning task $i$:

**Proposition 2.** *Assume the regularity assumptions given in Section 5, specifically that the function $\ell$ is $B_\ell$-bounded. Then, with probability at least $1 - \delta$, we have*

$$\max_{i \in [n]} F_i(x) \le \max_{i \in [n]} \hat{F}_i(x) + 2\mathcal{R}_m(\mathcal{F}) + B_\ell \sqrt{\frac{\log n/\delta}{2m}}. \tag{162}$$

*Here we assume that $\mathcal{F}_i = \mathcal{F}$ and $m_i = m$ for all $i \in [n]$ and hence $\mathcal{R}_{m_i}^i(\mathcal{F}_i) = \mathcal{R}_m(\mathcal{F})$ for all $i \in [n]$.*

Next, we proceed to bound the generalization on unseen tasks. Consider a new task with distribution $\mathcal{D}_{n+1} = \sum_{i=1}^n a_i \mathcal{D}_i$, for some $a \in \Delta_n$, meaning that the distribution of the new task is anywhere in the convex hull of the distribution of the old tasks. We make this assumption because if the new task is very dissimilar to the existing tasks, there is no reason to expect good generalization in the first place. We then show the following proposition:

**Proposition 3.** *For all $x \in \mathcal{X}$, with probability at least $1 - \delta$, we have*

$$F_{n+1}(x) \le \max_{p \in \Delta_n} p_i \hat{F}_i(x) + 2\sum_{i=1}^n a_i R_{m_i}^i(\mathcal{F}_i) + \sum_{i=1}^n a_i B_\ell \sqrt{\frac{\log(n/\delta)}{2m_i}}. \tag{163}$$

Notice that while Proposition 3 holds true for all $x$, the tightest upper bound is found when $x$ minimizes $\max_{p \in \Delta_n} p_i \hat{F}_i(x)$, which is precisely when $x = x^\star$, the optimal solution to problem we study in equation 2. This highlights another advantage of our formulation over TTSA: when we use the solution obtained by the single averaged objective $\sum_i f_i$ in equation 1, $x_{\text{min-avg}}^\star$, we will have a looser upper bound for $F_{n+1}(x_{\text{min-avg}}^\star)$ compared to $F_{n+1}(x^\star)$, showing that our formulation gives tighter robust (or worst case) generalization guarantees and this behaviour has been demonstrated empirically in section 4.

*Proof.* (of Proposition 3) First, by definition, for all $x$, we have that

$$F_{n+1}(x) = \mathbb{E}_{(\mathcal{D}^{\text{train}}_{n+1,j}, \mathcal{D}^{\text{test}}_{n+1,j}) \sim \mathcal{D}_{n+1}} \left[ \hat{f}_i \left( x, \arg\min_{y_i} \sum_{j'=1}^{m_i} \hat{g}_i(x, y_i, D^{\text{train}}_{i,j'}), D^{\text{test}}_{i,j} \right) \right] \tag{164}$$

$$= \sum_{i=1}^{n} a_i \mathbb{E}_{(\mathcal{D}^{\text{train}}_{n+1,j}, \mathcal{D}^{\text{test}}_{n+1,j}) \sim \mathcal{D}_i} \left[ \hat{f}_i \left( x, \arg\min_{y_i} \sum_{j'=1}^{m_i} \hat{g}_i(x, y_i, D^{\text{train}}_{i,j'}), D^{\text{test}}_{i,j} \right) \right] \tag{165}$$

$$= \sum_{i=1}^{n} a_i F_i(x). \tag{166}$$

Therefore, for all $x$, using a union bound over $1 \le i \le n$, we get that with probability at least $1 - n\delta'$, we have

$$F_{n+1}(x) = \sum_{i=1}^{n} a_i F_i(x) \le \sum_{i=1}^{n} a_i \hat{F}_i(x) + 2 \sum_{i=1}^{n} a_i R^i_{m_i}(\mathcal{F}) + \sum_{i=1}^{n} a_i B \sqrt{\frac{\log 1/\delta'}{2m_i}}. \tag{167}$$

Letting $\delta = n\delta'$ in (167), we have

$$F_{n+1}(x) \le \max_{p \in \Delta_n} p_i \hat{F}_i(x) + 2 \sum_{i=1}^{n} a_i R^i_{m_i}(\mathcal{F}) + \sum_{i=1}^{n} a_i B_\ell \sqrt{\frac{\log(n/\delta)}{2m_i}} \tag{168}$$

Therefore, plugging in $x^\star$ gives us

$$F_{n+1}(x^\star) \le \min_{x \in \mathcal{X}} \max_{p \in \Delta_n} p_i \hat{F}_i(x) + 2 \sum_{i=1}^{n} a_i R^i_{m_i}(\mathcal{F}) + \sum_{i=1}^{n} a_i B_\ell \sqrt{\frac{\log(n/\delta)}{2m_i}}. \tag{169}$$

$\square$

## C   IMPLEMENTATION DETAILS AND COMPUTE RESOURCES

We perform our experiments in Python 3.7.10 and PyTorch 1.8.1 with Intel(R) Core(TM) i5-8265U CPU @ 1.60GHz. The code is available at our repository https://github.com/minimario/bilevel. For our empirical evaluation, we first select $\alpha, \beta$ that give good performance/convergence for the min-avg problem (the baseline). We do a hyperparameter search to choose these parameters, specifically the initial learning rates. Then we fix $\alpha, \beta$ and only select $\gamma$ that provides good convergence for the min-max problem (our proposed scheme).

**Hypergradient computation.** We would like to note that the analysis does not require the actual hypergradient but rather a stochastic estimate with bounded bias. The standard Hessian inverse approximation using the Neumann series (Agarwal et al., 2017; Ghadimi & Wang, 2018; Hong et al., 2020) is one way of computing this estimate (as we have discussed in section 3.2 preceding Corollary 1). Since we are considering a single-loop algorithm, even a straightforward iterative differentiation (Ji et al., 2021) can provide an sufficiently useful estimate of the hypergradient. We utilize this for our empirical evaluations.

### C.1   SINUSOID REGRESSION TASK

We consider the sinusoid regression experiment (Finn et al., 2017), a multi-task representation learning problem where each task $\mathcal{T}_i$ is a regression problem $y = t_i(x) = a_i \sin(x - \phi_i)$. We uniformly sample the amplitude $a_i \in [0.1, 5]$, frequency and phase $\phi_i \in [0, \pi]$ for each task. We use $n = 3$ training tasks and 3 testing tasks, with 2 "easy tasks" ($a_i \in [0.1, 1.05]$) and one "hard tasks" ($a_i \in [4.95, 5]$) for each set. We use easy and hard tasks following the setup in (Collins et al., 2020). During training, for each task $i$, the learner is given samples $(x, y)$, $x \in [-5, 5]$. The goal is to learn a function approximating $t_i$ as best as possible in the mean squared error sense.

As described in Section 2, we use a neural network divided into two pieces, i.e., an embedding network and a task-specific network. The embedding network $f : \mathbb{R} \to \mathbb{R}^{10}$ consists of two hidden ReLU layers of size 80 and a final fully connected layer of size 10. Each task-specific network $g_i : \mathbb{R}^{10} \to \mathbb{R}, i \in [n]$ is a one-layer linear layer. Therefore, the loss on an input $x \in \mathbb{R}$ and $y = t_i(x)$ for task $i$ is $(g_i(f(x)) - y)^2$, and the true loss of the network with parameters $f, g_i$ are $\ell_i(f; g_i) = \mathbb{E}_{(x,y)}[(g_i(f(x)) - y)^2]$. The embedding network is as follows:

| **Input** ($\mathbb{R}$) |
| --- |
| Linear FC Layer (output in $\mathbb{R}^{80}$) |
| ReLU |
| Linear FC Layer (output in $\mathbb{R}^{80}$) |
| ReLU |
| Linear FC Layer (output in $\mathbb{R}^{10}$) |

**Training:** At each iteration, we first perform the inner loop optimization step (meta-training) by sampling 10 shots from each of the tasks in order to update each of the task-specific network weights. We use just 1 inner loop step. Pseudocode for the inner loop is shown below in PyTorch-style:

```
for task_id in task_list:
    xs, ys = sample_batch(task_id, n_shots)
    embedding = embedding_network(xs)
    for _ in range(n_inner):
        head_optimizers[task_id].zero_grad()
        total_loss = get_loss(task_id, xs, ys)
        total_loss.backward()
        head_optimizers[task_id].step()
```

For each outer loop optimization, we run a meta-validation batch again containing 10 shots from each of the tasks. We then take an outer-loop step, optimizing the embedding weights using the results of the meta-validation batch. The meta-validation batch is sampled in the exact same way as the meta-training batch shown above.

**Regularization:** First, as in (Ji et al., 2020), we add weight regualarization during inner loop training of the form $\epsilon_w \sum_{w \in \mathcal{W}} \|w\|$, where $\mathcal{W}$ denotes the set of weight parameters, where $\epsilon_w = 0.01$. In PyTorch, this is expressed as

```
l2_reg_constant = 0.01
for p in heads[task].parameters():
    l2_reg += p.norm(2)
total_loss += l2_reg * l2_reg_constant
```

Next, for the inner $\lambda$ updates, we add a regularization term to the overall loss, $-\epsilon_\lambda \sum_{i=1}^n (\lambda_i - \frac{1}{n})^2$, where $\epsilon_\lambda = 3$, which pulls the $\lambda$s closer to uniform. In PyTorch, the $\lambda$ update is expressed as

```
task_gradient = task_losses[i]
reg_gradient = -mu_lambda * (lambdas[task] - 1/n)
lambdas[task] += (task_update + reg_update) * gamma
```

**Parameters:** For the Task-Robust version of the algorithm, we use $\alpha = 0.007, \beta = 0.005, \gamma = 0.003$. For the standard version of the algorithm, we use $\alpha = 0.007, \beta = 0.011, \gamma = 0.003$.

**Loss curves:** To approximate the true loss for measurement purposes, we use 100 equally-spaced samples from $[-5, 5]$. After each iteration, we calculated the maximum loss among all the tasks. In 1c, we show the minimum of these maximum losses up until each epoch.

**Results with more tasks:** Finally, we show another figure similar to Figure 1, but with 20 training tasks and 20 test tasks. It can be observed that both the task-robust training loss and the task-robust testing loss greatly outperform their respective standard losses.

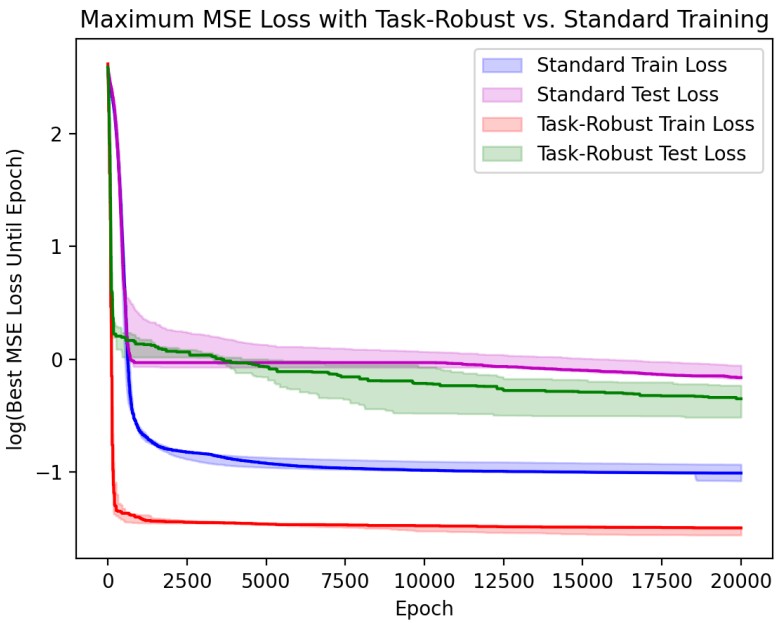

Figure 3: Comparison of standard (min-avg) training and robust (min-max) training using 20 tasks

## C.2 NONLINEAR REPRESENTATION LEARNING

We consider binary classification tasks generated from the FashionMNIST data set where we select 8 "easy" tasks (lowest log loss $\sim 0.3$ from independent training) and 2 "hard" tasks (lowest loss $\sim 0.45$ from independent training). We learn a shared representation network that maps the 784 dimensional (vectorized 28×28 images) to a 100 dimensional space. Each tasks then learns a binary classifier on top of this representation. The task specific objective $g_i$ for task $i$ corresponds to the cross-entropy loss on the training set, while the upper level objective $f_i$ corresponds to the loss of the $y_i^\star(x)$ with the learned representation $x$ on a validation set. We also maintain a heldout test set which we use to evaluate the generalization of the learned representation and per-task models.

For our data, we had $x \in \mathbb{R}^{784 \times 100}$ and $y \in \mathbb{R}^{100 \times 2}$. We used step sizes $\alpha = 0.01, \beta = 0.01$, and $\gamma = 0.3$. We used batch sizes of 8 and 128 to compute $g_i$ for each inner step and $f_i$ for each outer iteration, respectively. In addition, we included $\ell_2$-regularization of $y$ with regularization penalty 0.0005. We used vanilla SGD with a learning rate scheduler (ReduceLROnPlateau), invoked every 100 outer iterations, with patience of 10. Each optimization was executed for 10000 outer iterations. The results are generated by aggregation over runs with 10 different seeds.

## C.3 HYPERPARAMETER OPTIMIZATION

In this application, we use learning rates $\alpha = 0.0001$, $\beta = 0.001$, $\gamma = 0.001$ and 20000 outer iterations. We use a batch size of 8 for both the inner and outer steps for each $i \in [16]$ for the initial experiment in figure 2a. The optimizer was vanilla SGD with a learning rate scheduler (ReduceLROnPlateau), invoked every 100 outer iterations, with patience of 30. The results are generated by aggregating over 10 runs with different seeds. For the other HPO experiments, the number of tasks $n$ and the batch sizes are discussed in the main text.

## D ADDITIONAL TECHNICAL DETAILS

Here we provide further discussion on some technical aspects of the problem we are studying in this paper.

## D.1 WEAK-CONVEXITY AND NON-CONVEXITY

We consider weakly convex UL objective, and here we discuss how it is related to non-convexity. Weak convexity captures a class of non-convex problems. Weakly convex functions are not convex – note difference in the following definitions (also in Appendix A.1, Assumptions 1 and 2). For any convex function $\tau$, there exists a $\mu \geq 0$ such that, for any $x, x'$ ($x \neq x'$)

$$\tau(x') \geq \tau(x) + \langle \nabla_x \tau(x), x' - x \rangle + \mu \|x' - x\|^2, \tag{170}$$

whereas, for a weakly-convex function $\kappa$, there exists $\nu > 0$ such that, for any $x, x'$ ($x \neq x'$)

$$\kappa(x') \geq \kappa(x) + \langle \nabla_x \kappa(x), x' - x \rangle - \nu \|x' - x\|^2. \tag{171}$$

Note the "$-\nu$" for a weakly-convex $\kappa(\cdot)$ instead of the "$+\mu$" for a convex $\tau(\cdot)$ in the third term on the right hand side of the above two inequalities. So $\kappa(\cdot)$ is clearly not convex. Moreover, note that the $\|x' - x\|^2$ term on the right-hand side of the inequality for the weakly-convex function is strictly positive, implying that, for large enough $\nu$, the inequality will be true for **any** function. We provide convergence results which depend on the coefficient of weak-convexity (for our UL function in question, it is denoted as $\mu_\ell$), with slower rates for larger coefficients.

## D.2 COMPARISON WITH HU ET AL. (2022)

Hu et al. (2022) may *appear* similar to our work at a glance, but we would like to clarify that the differences are nontrivial as we are solving a different problem. We address this briefly in section 2 (**Closely related and Concurrent Work**), but we will elaborate further here to make the distinction clearer.

At a high level, the problem in Hu et al. (2022) is not multi-objective: the authors explicitly call it multi-block. They are still solving the single-objective min-max problem $\min_x \max_\alpha f(x, \alpha)$. Hence the problem setup in Hu et al. (2022) cannot solve standard bilevel learning applications such as representation learning and HPO; they choose AUC maximization as their motivating example instead.

Now, we explain what may be a source of confusion: why it seems like they are solving a multi-objective problem. Hu et al. (2022) start with the min-max problem $\min_x \max_\alpha f(x, \alpha)$ with strong concavity in $\alpha$, such as in AUC maximization. Then they make it bilevel to $\min_x \max_\alpha f(x, y^\star(x), \alpha)$ subject to $y^\star(x) = \arg\min_y g(x, y, \alpha)$ by splitting the $x$ variable and then further splitting into multi-block to $\min_x \max_{\alpha_i, i \in [n]} \sum_i f_i(x, y_i^\star(x), \alpha_i)$. Here, each $f_i$ is strongly concave in $\alpha_i$. This is a different problem setup than ours and does not include our problem formulation.

Therefore, the crucial difference is this: they study a single-objective problem $\min_x \max_\alpha f(x, \alpha)$, and we consider the robust multi-objective bilevel problem $\min_x \max_i f_i(x, y_i^\star(x))$. Their approach seems similar at first glance because they are solving the single-objective problem in a bilevel, multi-block way, but their problem class does not encompass the multi-objective one we consider.

## D.3 IMPROVING THE SAMPLE COMPLEXITY OF MORBiT

There is a potential room for improvement in the sample complexity of MORBiT. In the $n = 1$ case, our algorithm builds off of TTSA (Hong et al., 2020) with a $O(1/\epsilon^{2.5})$ complexity. The only existing work in the $n = 1$ case with a better sample complexity in a *single-loop constrained UL case* is the extremely recent STABLE (Chen et al., 2022b), achieving $O(1/\epsilon^2)$. STABLE, has a much more complex LL update than TTSA using variance reduction techniques. We are optimistic that more complex algorithms like STABLE can be extended to the robust multi-objective bilevel optimization setting with improved sample complexity.

## D.4 WHY ROBUST $\min\max$ INSTEAD OF PARETO MULTI-OBJECTIVE OPTIMIZATION?

Bilevel optimization problems are ubiquitous in machine learning applications such as representation learning and hyperparameter optimization, which is difficult to formulate as a single-level problem. We consider standard stochastic bilevel problems such as these, formulating a natural robust multi-objective version of these problems inspired by the benefits of robust multi-objective learning

highlighted in Mehta et al. (2012) and Collins et al. (2020). These papers consider the robust multi-objective view but do not study stochastic bilevel learning problems, which we do. Existing bilevel optimization problems, however, are all single-objective rather than multi-objective.

The advantages of taking single objective problems and formulating them as robust multi-objective problems have been highlighted in various works – see the literature cited in section 2 (**Min-max Robust Optimization in Machine Learning**). To summarize, the main advantage is that we can get guarantees on the worst-case performance instead of the usual average case performance (see for example our generalization guarantees in Appendix B). If we just summed the objectives and solved a single-objective problem, we would only be able to establish guarantees for the average-case performance: maybe we would find a solution that is good for most tasks, but might do extremely poorly on some. Moreover, at the lower level (LL) problem, there are different objectives for the learners as the individual problem structures and data distributions are different, again forming a natural multi-objective optimization (MOO) problem.

Much like our motivating existing literature on robust multi-objective learning, we focus on a single robust solution instead of a set of Pareto optimal solutions since, in various applications, we finally need select a single solution, and the robust (min max) solution provides stronger worst-case guarantees than any Pareto-optimal solution, which is our main motivation.

Pareto frontiers can be very useful and informative, potentially allowing us to understand the tradeoff between the multiple objectives. However, we would like to note that there are various forms of solutions in multi-objective optimization. There are Pareto optimal solutions, but also "possibly optimal" solutions (Wilson et al., 2015), convex coverage set of solutions (Yang et al., 2019), and min max robust solution (that we consider). The appropriate form of solution(s) would depend on the application, and we are focusing on min max applications, motivated by existing work such as Mehta et al. (2012) and Collins et al. (2020), since a min max solution can be shown to have good generalization guarantees (as we have also shown in Appendix B).

Furthermore, while the Pareto frontier can be more informative and the Pareto curves better demonstrate tradeoff between the objectives, it is important to note that, this curve is mostly intuitive with obvious tradeoffs for $n = 2$ objectives. With $n > 3$ objectives, the Pareto frontier cannot even be visualized, and one has to resort to pairwise comparisons, making it hard to reason about the tradeoffs between objectives even for moderately high $n$ since we will have to consider $n^2$ such comparisons (for example $n \sim O(10)$). Therefore, given a Pareto front of solutions, it is not clear which of the Pareto optimal solutions we should select.

One advantage of the min max formulation (equation 2) is that it tries to seek a single solution instead of a set of solutions. This allows us to use the solution for a new related problem (like for a new related task in representation learning application or hyperparameter optimization application in Franceschi et al. (2018)), we can use the robust min max solution – we select the robust solution for the shared UL variable $x$ (the representation network or the hyperparameter configuration). With a Pareto front, it is not clear which solution to pick for a new task since we would have a set of solutions, without the knowledge of which one would be useful for a new task/objective.

Furthermore, while a solution on the Pareto frontier implies that there is no other solution that "dominates" it, to the best of our knowledge, *there is no guarantee that some solution on the* **obtained Pareto frontier** *achieves the optimal value for the robust* min max *objective* $\max_i \min_x f_i(x)$ **unless** the Pareto frontier is completely dense, which is never the case. Multi-objective optimizers can return a set of solutions on the Pareto frontier, but even uniformly covering the Pareto frontier requires the size of the solution set to grow exponentially in the number of objectives $n$.

Finally, for nonconvex objective functions, the Pareto frontier refers to the Pareto stationarity rather than Pareto optimality. Our considered first-order stationarity condition is defined on the weighted average of the objective value, while the classical Pareto stationarity (please see Fernando et al. (2023, equation (2)) and references therein) is measured on the size of the weighted average of the gradients. The weighting vector in both of these two notations is optimized over a simplex. Therefore, the stationarity condition of our proposed min max formulation can be considered as one variant of Pareto stationarity for nonconvex problems.

