# OpenReview forum: "Min-Max Multi-objective Bilevel Optimization with Applications in Robust Machine Learning"
_ICLR.cc/2023/Conference — ICLR 2023 poster_

### Official Review · Reviewer_fwCP · 2022-10-24

**Confidence:** 5
**Correctness:** 3
**Technical Novelty And Significance:** 3
**Empirical Novelty And Significance:** 2
**Recommendation:** 6

**Clarity, Quality, Novelty And Reproducibility:**

The complexity results proposed in this work is worse than the best-known results. This needs to be clearly discussed and clarified. The authors need to show under which setting, their results are better and why it is specifically worse for the classical case of $n=1$.

**Strength And Weaknesses:**

The problem under consideration is interesting, however, a similar problem already studied by (Hu et al., 2022) with minor differences. Moreover, their sample complexity is in the order of ${\cal O}(1/\epsilon^2)$ which is better than the one in this work ${\cal O}(1/\epsilon^{2.5})$ when $n=1$. Thus, the authors need to discuss why their result is worse for the case of $n=1$. Moreover, a table is required to compare the proposed complexity bounds with the existing ones.

**Summary Of The Paper:**

In this work, the authors consider a class of bilevel optimization problems in which he objective function of the upper level is a maximum over $n$ different functions for each one a separate lower level problem is given. In particular, they propose a single-loop algorithm in which a gradient descent update is performed for both upper and lower level decision variables per iteration. The authors then provide simple complexities of finding a stationary point of the upper level problem and optimal solutions of the lower level ones.

**Summary Of The Review:**

The paper needs more discussion and comparison with the existing works.

---

> ### Author Response · Authors · 2022-11-11
> **Thank you for the review (thread 1/2)**
>
> We thank the reviewer for their review and their comments. We address each of the comments separately as follows:
>
> -----------------------------
>
> > The problem under consideration is interesting, however, a similar problem already studied by (Hu et al., 2022) with minor differences.
>
> We acknowledge that Hu et al. (2022) may *appear* similar to our work at a glance, but we would like to clarify that the differences are not "minor" as we are solving a different problem. We address this on page 4 in Section 2 **Closely related and Concurrent Work**, but we will elaborate further here to make the distinction clearer.
>
> At a high level, the problem in Hu et al (2022) is not multi-objective: the authors explicitly call it multi-block. They are still solving the single-objective min-max problem $\min_x \max_{\alpha} f(x, \alpha)$. Hence the Hu et al. (2022) problem setup cannot solve standard bilevel learning applications such as representation learning and HPO. They choose AUC maximization as their motivating example instead.
>
> Now, we explain what may be the source of confusion: why it seems like they are solving a multi-objective problem. Hu et al. (2022) start with the min-max problem $\min_x \max_\alpha f(x, \alpha)$ with strong concavity in $\alpha$, such as in AUC maximization. Then they make it bilevel to $\min_x \max_\alpha f(x, y^\star (x), \alpha)$ subject to $y^\star(x) = \arg \min_y g(x, y, \alpha)$ by splitting the $x$ variable and then further splitting into multi-block to $\min_x \max_{\alpha_i, i \in [n]} \sum_i f_i(x, y_i^\star(x), \alpha_i)$. Here, each $f_i$ is strongly concave in $\alpha_i$. This is a different problem setup than ours and does not include our problem formulation.
>
> Therefore, the crucial difference is this: they study a single-objective problem $\min_x \max_{\alpha} f(x, \alpha)$, and we consider the robust multi-objective bilevel problem $\min_x \max_i f_i(x, y_i^\star(x))$. Their approach seems similar at first glance because they are solving the single-objective problem in a bilevel, multi-block way, but their problem class does not encompass the multi-objective one we consider.
>
> -----------------------------
>
> > Moreover, their sample complexity is in the order of  $\mathcal{O}$$(1/\epsilon^2)$ which is better than the one in this work  $\mathcal{O}$$(1/\epsilon^{2.5})$ when $n=1$. Thus, the authors need to discuss why their result is worse for the case of $n=1$.
>
>
> We acknowledge the reviewer's concern. As we have elaborated above, we are solving a different problem than theirs, so the sample complexities are not directly comparable.
>
> First, we discuss the novelty in our work: our focus of interest is $n > 1$; $n = 1$ has no notion of task robustness, since there is only one objective. The $\min_x \max_{i \in [n]}$ is equivalent to just $\min_x$ for $n = 1$ and hence not the focus of our problem formulation. The novelty of our problem formulation and the analysis is for the $n > 1$ case where we have to (i) simultaneously establish convergence for all $n$ LL variables $\bar y_i$, and (ii) handle $n$ separate descent equations for the $n$ LL variables to then establish convergence of the UL variable $\bar{x}$ by bounding the non-smooth $\mathbb{E} [ \max_{i \in [n]} \| \| \bar y_i - y^\star(\bar{x}) \|\|^2]$; note that this is especially challenging since a bound on $\mathbb{E} [ \|\| \bar y_i - y^\star(\bar{x}) \|\|^2]$ on each $i \in [n]$ separately does not directly imply a bound on $\mathbb{E} [ \max_{i \in [n]} \| \| \bar y_i - y^\star(\bar{x}) \| \|^2]$. We have explicitly discussed the novelty and the challenges in our analysis after the Theorem 1 statement in Section 3.
>
> Second, note that Hu et al. (2022) consider an unconstrained UL variable $x$, and hence get the improved sample complexity -- this is standard in stochastic bilevel optimization literature where improved results can be achieved for unconstrained UL variable (please see Table 1). The constrained UL setting that is needed for HPO problems is not conducive to the acceleration schemes utilized in most existing papers for improved results.
>
> Finally, we agree that there is potential room for improvement. In the $n=1$ case, our algorithm builds off of the TTSA algorithm [Hong et al., 2020] with a complexity of $\mathcal{O}$$(1/\epsilon^{2.5})$ complexity. The only existing work in the $n=1$ case with a better sample complexity in a single-loop constrained UL case is the extremely recent STABLE (Chen et al. 2022b), achieving  $\mathcal{O}$$(1/\epsilon^2)$. STABLE, has a much more complex update than TTSA using variance reduction techniques. That being said, we are very excited by the idea of extending more complex algorithms like STABLE to the multi-objective bilevel optimization setting. We are optimistic about the potential of these ideas and wish to explore them in future work.

---

> > ### Comment · Reviewer_fwCP · 2022-12-06
> > **Changing the score**
> >
> > Thanks to the authors for their explanation. I raised my score to marginally above the acceptance threshold.

---

> ### Author Response · Authors · 2022-11-11
> **Thank you for the review (thread 2/2)**
>
> > Moreover, a table is required to compare the proposed complexity bounds with the existing ones.
>
> We would like to observe that comparing sample complexities is not appropriate (or at least mathematically rigorous) in our case since we are comparing sample complexities across different problems -- single objective vs multi-objective problems.
>
> -----------------------------
>
> > **Clarity, Quality, Novelty And Reproducibility.** The complexity results proposed in this work is worse than the best-known results. This needs to be clearly discussed and clarified. The authors need to show under which setting, their results are better and why it is specifically worse for the classical case of $n=1$.
> >
> > **Summary.** The paper needs more discussion and comparison with the existing works.
>
> We thank the reviewer for the constructive feedback and review. First, we would like to first point out that the problem under consideration is significantly different from that in Hu et al. (2022), which we elaborate on above. Second, a major contribution of our work is in advancing a single-objective algorithm to work in the multi-objective optimization case, both theoretically and empirically. Our work is the first work to do so. Finally, we agree that TTSA (the algorithm we build off of) is not the one with the best sample complexity. However, it is intuitive, easily parallelizable, and straightforward to implement. We believe that this work is just a starting point in multi-objective algorithms for bilevel optimization: as the reviewer suggests, there is potential for better sample complexity.
>
> In light of this, we respectfully ask the reviewer to consider re-evaluating the merit of our work and increasing their score. We are more than happy to give further clarifications in case anything is still unclear.

---

> ### Author Response · Authors · 2022-12-06
> **Checking in with reviewer fwCP**
>
> Hi Reviewer fwCP: we haven't heard back from you and would like to check in with you to see if our response addresses your concerns. Are there further questions we can answer or clarifications we can make, particularly regarding the novelty of our work and how it differs from the existing works? Once again, we would like to respectfully ask you to consider re-evaluating the merit of our work and increase the score, and we look forward to discussing further!
>
> Best, Authors

---

### Official Review · Reviewer_UZe1 · 2022-10-24

**Confidence:** 5
**Clarity, Quality, Novelty And Reproducibility:** The quality, clarity and originality …
**Correctness:** 3
**Technical Novelty And Significance:** 3
**Empirical Novelty And Significance:** 3
**Recommendation:** 6

**Strength And Weaknesses:**

Pros:
1. This paper builds a frame for such a specific bilevel optimization problem, which has not been considered before. The proposed algorithm is simple and easy to implement with a single-loop structure.

2. In the reformulated problem in (3), the upper level problem turns to optimization a maximization problem over a simplex. This new component requires some new treatments.  Comprehensive convergence rate analysis is provided to justify the algorithmic designs.

3. The algorithm itself is easy to understand and apply. Experiments seem to support the proposed algorithm well.

Cons:
1. Excluding the nicely designed problem itself, the algorithm is not that surprising given existing single-loop algorithms such as Hong et al., 2020 in bilevel optimization. It looks like applying SGD (projected-SGD) to a specific problem. I wonder how the authors implement the hypergradient in practice? Is there any instructions here?

2. For the experiments, e.g., in Fig. 5, why does the variance look so large?

3. The analysis seems to follow from TTSA (Hong et al., 2020), and yields a similar rate of $K^{-2/5}$. It seems that some other SGD types of bilevel methods like stocBiO (Ji et al., 2021) have better complexity. Can it be used here for improved complexity?

4. It seems to me that the complexity still has space to be improved using some strategies like variance reduction. Can the authors elaborate more on this?

**Summary Of The Paper:**

This paper studies robust solution for bilevel optimization under min-max multi-objective form. The main idea is build the problem by a min-max upper-level objective function and using weighted sum of weakly convex functions li instead of a single weakly convex function. To solve such a problem, the SGD(projected SGD) to iterate $y$, $x$ and $\lambda$. Some applications such as robust representation learning and robust hyperparameter selection  are provided to evaluate the performance.

**Summary Of The Review:**

Overall, I feel this work does a good job in designing a novel from of bilevel problem. It provided a precious frame for further and finer solution. I tend to accept this work but with a conservative score given the above questions. However, I am open to change my mind based on other reviewer’s comments and the authors’ feedbacks.

---

> ### Author Response · Authors · 2022-11-11
> **Thank you for the review (thread 1/2)**
>
> We thank the reviewer for their review and their insightful support and comments. We are glad that the reviewer recognizes the novelty of the problem formulation, the simplicity of our proposed algorithm, and the technical quality of our novel analysis.  We address each of the comments separately in the following:
>
> ----
> >  Excluding the nicely designed problem itself, the algorithm is not that surprising given existing single-loop algorithms such as Hong et al., 2020 in bilevel optimization. It looks like applying SGD (projected-SGD) to a specific problem.
>
> As we have discussed in the **Comparison with Related Work** on page 6, while our algorithm is similar to existing single-loop bilevel optimization algorithms, our main contribution lies in establishing convergence rates for a different problem, which requires us to handle various new challenges not studied in existing literature.
>
> ----
> > I wonder how the authors implement the hypergradient in practice? Is there any instructions here?
>
> The analysis does not require the actual hypergradient but rather a stochastic estimate with bounded bias. The standard Hessian inverse approximation using the Neumann series is one way of computing this estimate (as we have discussed at the bottom of page 6). Since we are considering a single-loop algorithm, even a straightforward iterative differentiation [Ji et al., 2021] can provide an sufficiently useful estimate of the hypergradient. We utilize this for our empirical evaluations.
>
> ----
> > For the experiments, e.g., in Fig. 5, why does the variance look so large?
>
> The variance in only high in Figure 5 plotting the gradient norm of the upper level variable; other figures plotting the actual objectives have much smaller variances. First, we would like to note that these are the norms of the **stochastic gradients** with a very small batch size of 8, leading to high variance. But the plots still highlight the convergence trends.
>
> Furthermore, please kindly note that even though the single-loop algorithms, e.g., Hong et al. (2020), have been studied for stochastic bilevel problems, our considered problem actually contains another layer of the maximization, resulting in a triple-sequence design of the algorithm. The stochastic noise errors are coupled as the algorithms proceeds. The technical challenges also include the control of the ascent incurred from the maximization process, which is not present in existing stochastic bilevel optimization literature.
>
> ----
>
> > The analysis seems to follow from TTSA (Hong et al., 2020), and yields a similar rate of $K^{-2/5}$. It seems that some other SGD types of bilevel methods like stocBiO (Ji et al., 2021) have better complexity. Can it be used here for improved complexity?
>
> We agree with the reviewer that the algorithm we use partially follows from TTSA, and therefore naturally has a similar rate. We also agree that using other types of methods like stocBiO could potentially lead to different algorithms for our multi-objective optimization framework with better complexity. However, each algorithm applies in its own setting: as we highlighted in Table 1, stocBiO is a double-loop algorithm for the unconstrained upper-level optimization problem, which is different from the setting that we focus on (single-loop with constrained upper-level). That being said, we thank the reviewer for the idea and highly encourage future work that builds off our analysis to extend other bilevel methods to the multi-objective setting, as we find this direction promising.
>
> ---
> > It seems to me that the complexity still has space to be improved using some strategies like variance reduction. Can the authors elaborate more on this?
>
> We again thank the reviewer for the suggestion of using variance reduction. The success of variance reduction depends on the problem setup: variance reduction techniques are usually leveraged for unconstrained upper-level (UL) problem, while problems like HPO require a constrained UL (which is what we study in this paper). However, a very recent work, STABLE (Chen et al., 2022b), uses a novel strategy to improve the complexity for the single-loop single-objective bilevel case from $\mathcal{O}(K^{-2/5})$ to $\mathcal{O}(K^{-1/2})$ by increasing the computational cost of the lower-level (LL) step. Therefore, we agree that the complexity can potentially be improved and are optimistic about this possibility as follow-up or future work.
>
> ---

---

> ### Author Response · Authors · 2022-11-11
> **Thank you for the review (thread 2/2)**
>
> > **Summary:** Overall, I feel this work does a good job in designing a novel from of bilevel problem. It provided a precious frame for further and finer solution. I tend to accept this work but with a conservative score given the above questions. However, I am open to change my mind based on other reviewer's comments and the authors' feedbacks.
>
> We thank the reviewer for the encouraging comments and support of our proposed new learning framework and algorithm. We hope that our comments clarify the reviewer's concerns regarding the experimental implementation and results. We appreciate the reviewer's suggestions of using other methods to achieve a better complexity and agree that this would be interesting and important future work. Please let us know if you have any further questions, and I hope you would consider increasing the score in light of these clarifications.

---

> > ### Comment · Reviewer_UZe1 · 2022-11-14
> > **Thanks for the response**
> >
> > I thank the authors for the detailed response. My questions are mostly addressed and I will increase my score to 7.
> >
> > Best,
> > Reviewer

---

> > > ### Author Response · Authors · 2022-11-16
> > > **Thank you for your support**
> > >
> > > We are very pleased to know that your concerns have been addressed. Thank you for recognizing the importance of our work and commitment to increasing the score.

---

> > > ### Author Response · Authors · 2022-12-06
> > > **Increasing the score + happy to clarify further**
> > >
> > > Hi Reviewer UZe1, if you still maintain the position that the score is 7, we kindly ask that you increase it in the original review. In addition, if you have any further questions or clarifications to make, we are happy to help with that as well.
> > >
> > > Best, Authors

---

### Official Review · Reviewer_XgUo · 2022-10-24

**Confidence:** 4
**Correctness:** 3
**Technical Novelty And Significance:** 3
**Empirical Novelty And Significance:** 3
**Recommendation:** 5

**Clarity, Quality, Novelty And Reproducibility:**

The authors present the problem formulation and their proposed solution clearly.
The problem formulation and the proposed algorithm are both original works.
Considering the complex parameter settings of the proposed algorithm, It is necessary to have the open-sourced code to reproduce all results in this work.

**Strength And Weaknesses:**

Strength
1. The authors did a thorough literature review w.r.t. robust learning and multi-task learning.
2. The authors created a new MOO bilevel problem, proposed a solution to it, and established its convergence rate.

Weaknesses
1. The main issue with this work is that the paper is trying to create an (unnecessary) multi-objective optimization (MOO) problem and address it. It is challenging to understand the benefit of formulating a MOO problem as a bilevel one instead of summing the objectives and using a single-objective solver or generating multiple Pareto-optimal solutions. Specifically, it is worth comparing the proposed formulation with the Pareto-optimal solutions considering that the proposed one only gives one robust solution and the Pareto front gives a full set of solutions. What is the complexity difference? What is the convergence difference? Besides, the Parent solution can be generalized easily to nonconvex use cases but can the proposed algorithm be extended to the nonconvex cases? How can the proposed one be justified to be beneficial?
2. It is also interesting to convert problem (2) to (3) by replacing the objective in the leader problem with a weighted sum of multiple objectives. Though the equivalence might be straight for the authors, it seems not the case for the majority of readers. It will be good to prove why these two formulations are equivalent.
3. Equation (2) has a quite strong assumption that there are n different objective function pairs (fi, gi), i.e, the number of objectives in the leader problem and the follower problem is the same. This is usually not true considering that the number of leaders and the number of followers are usually not of the same magnitude.
4. The examples given in the introduction part, e.g., safety-critical, scenario-specific, etc are very unclear. What is a concrete example whose formulation is like (2)? This formulation is also different from the existing MOO works in Sinha et al., 2015; Deb & Sinha, 2009; Ji et al., 2017.
5. The authors assume the LL problem is unconstrained strongly convex and the UL problem is constrained weakly convex. This is again a very strong assumption and hardly satisfied in reality considering the main application is in RL and HPO.
6. For this iterative algorithm, how to choose the three learning rates alpha, beta and gamma are critical. The authors only give the chosen numbers at the end of the appendix but did not give any intuition on how they are selected especially how this matches what Theorem 1 suggested regarding these three learning rates.
7. In Theorem 1, the bias norm is bounded by a sequence bk. How to verify that b_k^2 <= alpha in practice?
8. The writing needs to be polished.
a. At the end of the first paragraph, there is typo 'Similar'.
b. In Table 1, it is not fair to list the problem solution and solving strategy together. Specifically, the single loop should not be listed with other columns.

**Summary Of The Paper:**

The authors formulate a new min-max multi-objective bilevel optimization problem and applied it in robust ML and HPO.
They also proposed a new algorithm to find a solution to the proposed new problem and establish its convergence rate and computational complexity.

**Summary Of The Review:**

The framework and algorithms are novel enough but the benefit of the way formulating the problem compared with the existing MOO formulation is skeptical. The strong assumption w.r.t. the leader and follower problems is another concern. Considering this, it is hard to recommend its publication in its present form.

------------------------
I have read the authors' replies. The response has not significantly changed my view of the paper, and thus I would like to keep the same score.

---

> ### Author Response · Authors · 2022-11-11
> **Thank you for the review (Thread 1/4)**
>
> We thank the reviewer for their thorough review and their insightful questions and comments. We address each of these comments separately in the following (alongside the reviewer's original comment):
>
> ---------------------
> > The main issue with this work is that the paper is trying to create an (unnecessary) multi-objective optimization (MOO) problem and address it. It is challenging to understand the benefit of formulating a MOO problem as a bilevel one instead of summing the objectives and using a single-objective solver or generating multiple Pareto-optimal solutions.
>
> Bilevel optimization problems are ubiquitous in machine learning applications such as representation learning and hyperparameter optimization, which is difficult to formulate as a single-level problem. We consider standard stochastic bilevel problems such as these, formulating a natural robust multi-objective version of these problems inspired by the benefits of robust multi-objective learning highlighted in Mehta, et al. (2009) and Collins et al. (2020). These papers consider the robust multi-objective view but do not study stochastic bilevel learning problems, which we do. Existing bilevel optimization problems, however, are all single-objective rather than multi-objective.
>
> The advantages of taking single objective problems and formulating them as robust multi-objective problems have been highlighted in various works (see the literature cited in Section 2, Min-max Robust Optimization in Machine Learning of our paper). To summarize, the main advantage is that we can get guarantees on the worst-case performance instead of the usual average case performance (see for example our generalization guarantees in Appendix B). If we just summed the objectives and solved a single-objective problem, we would only be able to establish guarantees for the average-case performance: maybe we would find a solution that is good for most tasks, but might do extremely poorly on some. Moreover, at the lower level (LL) problem, there are different objectives for the learners as the individual problem structures and data distributions are different, again forming a natural multi-objective optimization (MOO) problem.
>
> Much like our motivating existing literature on robust multi-objective learning, we focus on a single robust solution instead of a set of Pareto optimal solutions since, in various applications, we finally need select a single solution, and the robust ($\min\,\max$) solution provides stronger worst-case guarantees than any Pareto-optimal solution, which is our main motivation.
>
> ---------------------
>
> > Specifically, it is worth comparing the proposed formulation with the Pareto-optimal solutions considering that the proposed one only gives one robust solution and the Pareto front gives a full set of solutions. What is the complexity difference? What is the convergence difference?
>
> As we mentioned earlier, we focus on a single robust solution instead of a set of Pareto optimal solutions since, in various applications, we finally need select a single solution, and the robust ($\min\,\max$) solution provides stronger worst-case guarantees than any Pareto-optimal solution, which is our main motivation.
>
> Existing literature on multi-objective bilevel optimization does not study the stochastic bilevel setup relevant in machine learning problems, and hence does not study the finite-time convergence guarantees or sample complexities like we do (and the existing literature on single-objective stochastic bilevel optimization does). Since such algorithms and analyses do not exist, we cannot compare to them.
>
> ---------------------
>
> > Besides, the Parent solution can be generalized easily to nonconvex use cases but can the proposed algorithm be extended to the nonconvex cases? How can the proposed one be justified to be beneficial?
>
> Our analysis considers a weakly convex upper-level (UL) objective which definitely includes certain nonconvex objectives (but not all) depending the level of nonconvexity (see $\mu_\ell$ Appendix A.1, Assumption 1, equation (13)). It is necessary to have some handle on the level of non-convexity $\mu_\ell$. The larger $\mu_\ell$ is, the more non-convex the objective is, and we pay the price in the convergence rate, as explicitly presented in Appendix A.2, Theorem 2, equation (29). As we mentioned above, the existing multi-objective bilevel optimization schemes do not consider the stochastic optimization setup and do not establish finite-time convergence rates for any class of objectives. The main benefit of our formulation is the guarantees we can establish on the worst-case performances enabled by the robust optimization problem, which are stronger guarantees than the average-case performances.

---

> ### Author Response · Authors · 2022-11-11
> **Thank you for the review (Thread 2/4)**
>
> > It is also interesting to convert problem (2) to (3) by replacing the objective in the leader problem with a weighted sum of multiple objectives. Though the equivalence might be straight for the authors, it seems not the case for the majority of readers. It will be good to prove why these two formulations are equivalent.
>
> Here we provide further details on the equivalence between problems (2) and (3) which we will incorporate in the main text. When there is an unique $\arg \max_{i \in [n]} f_i$, the solution to the problem
> $$
> \max_{\lambda \in \Delta_n} \sum_{i \in [n]} \lambda_i f_i, \text{
> where } \Delta_n = \{ \lambda \in \mathbb{R}^n \colon \lambda_i \geq 0 \forall i \in [n], \sum_{i=1}^n \lambda_i = 1\} \ \ (n\text{-simplex})
> $$
> is given exactly as
> $$
> \max_{\lambda \in \Delta_n} \sum_{i \in [n]} \lambda_i f_i = \max_{i \in [n]} f_i,
> $$
> for any $\{f_1, \ldots, f_n\}$ since the $\max_{\lambda \in \Delta_n}$ is achieved when we set the $\lambda_{i^\star} = 1$ for the $i^\star = \arg \max_{i \in [n]} f_i$, and $\lambda_i = 0 \forall i \not= i^\star \in [n]$. In case of non-unique solutions, the equivalence still holds by setting the $\lambda_{i^\star} \forall i^\star \in \arg \max_{i \in [n]} f_i$ appropriately. This equivalence between (2) and (3) has been formally shown in Proposition 3.2 in Lu et al. (2020). This allows us to replace (2) with the equivalent (3).
>
>
>
> -----------------------------
>
> > Equation (2) has a quite strong assumption that there are $n$ different objective function pairs ($f_i$, $g_i$), i.e, the number of objectives in the leader problem and the follower problem is the same. This is usually not true considering that the number of leaders and the number of followers are usually not of the same magnitude.
>
> This concern is not clear to us. Every bilevel problem would have a pair of objectives $f_i, g_i$ -- for example, problem specific training loss $g_i$ and problem specific validation loss $f_i$. And we are considering $n$ such pairs of objectives. This is standard in robust learning problems such as representation learning and HPO.
>
> We would like to understand what some applications are where this is not true. If there are applications where there are some problems $i \in [n]$ with no LL problem/objective $g_i$, this can still be emulated in our setup where the problem $i$'s LL variable $y_i$ is the null set and the LL optimization is a no-op. If there are problems $i \in [n]$ with no UL objective $f_i$, it is not clear how such a problem is even formulated.
>
> -----------------------------
>
>
> > The examples given in the introduction part, e.g., safety-critical, scenario-specific, etc are very unclear. What is a concrete example whose formulation is like (2)? This formulation is also different from the existing MOO works in Sinha et al., 2015; Deb \& Sinha, 2009; Ji et al., 2017.
>
> 'Safety-critical' refers to applications where worst-case performance guarantees are necessary. 'Scenario-specific' refers to the case where multiple objectives share a single variable while also having objective specific variables. For example, the shared variables $x$ can be the parameters of a representation network or hyperparameters, while the problem (or scenario-specific) variables correspond to the model parameters for each objective $y_i$ (given the shared variable $x$). We provide precise examples of formulation (2) in section 4 equation (11), where we optimize for robust representation parameters, and equation (12), where we optimize for robust hyperparameters.
>
> As mentioned earlier, much like our motivating existing literature on robust multi-objective learning, we focus on a single robust solution instead of a set of Pareto optimal solutions since, in various applications, we finally need select a single solution, and the robust solution provides stronger worst-case guarantees than any Pareto-optimal solution, which is our main motivation. Hence, it differs from the above cited multi-objective optimization (MOO) work.
>
> -----------------------------

---

> ### Author Response · Authors · 2022-11-11
> **Thank you for the review (Thread 3/4)**
>
> > The authors assume the LL problem is unconstrained strongly convex and the UL problem is constrained weakly convex. This is again a very strong assumption and hardly satisfied in reality considering the main application is in RL and HPO.
>
> The "strongly convex LL problem" is a standard assumption in every recent stochastic bilevel optimization paper that provides rigorous convergence guarantees. "Weakly convex" implies nonconvex. The strong convexity allows for a unique stationary point $y^\star(x)$ for the lower level (LL) problem for any given upper level (UL) variable $x$. If there are multiple stationary points for the LL problem, then the (hyper)gradient of the upper level optimization variable is not well defined, and can be shown to lead to NP-hardness of even finding a stationary point for the UL problem (see Theorem 5 in Zhang et al. (2020)). To the best of our knowledge, the current theoretical guarantees for the stochastic bilevel optimization algorithms all operate under a strongly convex LL assumption.
> This assumption is exactly true for the HPO problem we study empirically in equation (12) and true for the representation learning problem with large enough $\ell_2$ penalty $\rho$ in equation (11).
>
> *Zhang, Jingzhao, et al. "Complexity of finding stationary points of nonconvex nonsmooth functions." International Conference on Machine Learning. PMLR, 2020.*
>
>
> -----------------------------
>
> > For this iterative algorithm, how to choose the three learning rates alpha, beta and gamma are critical. The authors only give the chosen numbers at the end of the appendix but did not give any intuition on how they are selected especially how this matches what Theorem 1 suggested regarding these three learning rates.
>
> We choose these learning rate by following the theoretical analysis, but we need to tune the initial values of these learning rates properly, which can give good performance/convergence behaviors for the min-avg and min-max problem. For our empirical evaluation, we first select $\alpha, \beta$ that give good performance/convergence for the min-avg problem (the baseline). We do a hyperparameter search to choose these parameters. Then we fix $\alpha, \beta$ and only select $\gamma$ that provides good convergence for the min-max problem (our proposed scheme).
>
>
> -----------------------------
>
> > In Theorem 1, the bias norm is bounded by a sequence $b_k$. How to verify that $b_k^2 \leq \alpha$ in practice?
>
> First, we would like to emphasize that these are conditions under which we can establish our theoretical convergence rate: usually we don't need/try to satisfy this condition and rather select learning rates that give good empirical convergence. However, this condition can be satisfied empirically if desired. If we can explicitly compute the hypergradient (which is possible in some problems), the above condition is satisfied trivially. If we are using the Hessian inverse approximation (for example, utilized in Ghadimi and Wang, 2018) to compute the hypergradient, this condition can be satisfied empirically given a value for $\alpha$ (as we have discussed at the bottom of page 6).
>
>
> -----------------------------
>
> > The writing needs to be polished. a. At the end of the first paragraph, there is typo 'Similar'. b. In Table 1, it is not fair to list the problem solution and solving strategy together. Specifically, the single loop should not be listed with other columns.
>
>
> We will incorporate your feedback in our writing. However, we would like to mention that  "single-loop'' can be a property of the problem -- this property tells us what kind of access is available to the stochastic oracles for the two levels. This "single-loop" property also implies the simplicity of the designed algorithm to find the solution as the single-loop structure does not require any convergence metric or termination criteria for the inner loop learning process (necessary for double-loop algorithms), especially in the stochastic algorithms.
>
> -----------------------------

---

> ### Author Response · Authors · 2022-11-11
> **Thank you for the review (Thread 4/4)**
>
> > **Clarity, Quality, Novelty And Reproducibility.** The authors present the problem formulation and their proposed solution clearly. The problem formulation and the proposed algorithm are both original works. Considering the complex parameter settings of the proposed algorithm, It is necessary to have the open-sourced code to reproduce all results in this work.
>
>
> Our implementation follows the proposed straightforward min-max algorithm and the min-avg baseline with additional steps that track all the relevant statistics during the optimization. Our code is available on GitHub but we have not linked to it as per the double-blind requirements.
>
> -----------------------------
>
> > **Summary.**  The framework and algorithms are novel enough but the benefit of the way formulating the problem compared with the existing MOO formulation is skeptical. The strong assumption w.r.t. the leader and follower problems is another concern. Considering this, it is hard to recommend its publication in its present form.
>
> We hope that we have highlighted the benefit of our robust min-max multi-objective formulation and our proposed scheme and novel analysis further in the rebuttal. Specifically, bilevel optimization problems are widespread in machine learning, and transforming single-objective problems into robust multi-objective versions has been shown to increase robustness. We are bridging the connections between these two techniques here to enable robust optimization of a new class of problems, namely bilevel ones including hyperparameter optimization and representation learning. We sincerely ask that the reviewer reconsider their evaluation and consider increasing their score based on our response. We are happy to answer any further questions the reviewer would have, especially if the benefits of the formulation is still not clear.

---

> > ### Comment · Reviewer_XgUo · 2022-11-28
> > **Keep the same score**
> >
> > Thanks for your response and I appreciate your detailed explanations.
> >
> > 1. It is still unclear why the robust optimization formulation is necessary for e.g., hyperparameter optimization.
> > In fact, the Pareto frontier can give a much more informative solution curve clearly demonstrating the tradeoff between multiple objectives, and cannot offer a worse result than robust optimization.
> >
> > 2. How to tune $\alpha$, $\beta$ and $\gamma$ is critical. I strongly suggest adding a separate section to explain this in the revision. Also, it is worth explaining how to tune $\alpha$ to guarantee that $b_k^2 \leq \alpha$.
> >
> > 3. Besides, I disagree that weakly convexity implies non-convexity. The former mostly refers to convexity and is much narrower than the latter.

---

> > > ### Author Response · Authors · 2022-12-01
> > > **Comments on reviewer's XgUo's response (Thread 1/3)**
> > >
> > > Thank you for the questions and your commitment to understanding our work. We would like to address your concerns in more detail below, hoping to further clarify the novelty of our work.
> > >
> > > > It is still unclear why the robust optimization formulation is necessary for e.g., hyperparameter optimization. In fact, the Pareto frontier can give a much more informative solution curve clearly demonstrating the tradeoff between multiple objectives, and cannot offer a worse result than robust optimization.
> > >
> > > Concretely, consider the problem in [R1]: we are given a set of $n$ tasks, and the goal is to find a *single* good hyperparameter configuration $\lambda$ shared among all the tasks. Suppose, further, that it is important to have good performance across all the tasks **and** that it is important that the same single hyperparameter configuration $\lambda$ would generalize well to similar unseen tasks (this property is formalized in our results in Appendix B for an application). The minmax  robust multi-objective optimization formulation in our equation (2) precisely seeks such a $\lambda$. In our notation, the UL variable $x$ corresponds to the hyperparameter configuration $\lambda$.
> > >
> > > [R1] Franceschi, et al. Bilevel Programming for Hyperparameter Optimization and Meta-Learning, 2018.

---

> > > ### Author Response · Authors · 2022-12-01
> > > **Comments on reviewer's XgUo's response (Thread 2/3)**
> > >
> > > >In fact, the Pareto frontier can give a much more informative solution curve clearly demonstrating the tradeoff between multiple objectives, and cannot offer a worse result than robust optimization.
> > >
> > > We agree with the reviewer that Pareto frontiers are useful. However, we would like to note that there are various forms of solutions in multi-objective optimization. There are Pareto optimal solutions, but also "possibly optimal" solutions [R2], convex coverage set of solutions [R3], and minmax robust solution. The appropriate form of solution(s) would depend on the application, and we are focusing on minmax applications, motivated by existing work such as [R4] and [R5] since a minmax solution can be shown to have good generalization guarantees (as we have also shown in Appendix B).
> > >
> > > We agree that ``the Pareto frontier can give a much more informative solution curve clearly demonstrating the tradeoff between multiple objectives'', but note that, this curve is mostly intuitive with obvious tradeoffs for $n=2$ objectives. With $n > 3$ objectives, the Pareto frontier cannot even be visualized, and one has to resort to pairwise comparisons, making it hard to reason about the tradeoffs between objectives even for moderately high $n$ since we will have to consider $n^2$ such comparisons (for example $n \sim O(10)$). Therefore, given a Pareto front of solutions, it is not clear which of the Pareto optimal solutions we should select.
> > >
> > > One advantage of the minmax formulation (our equation (2)) is that it tries to seek a single solution instead of a set of solutions. This allows us to use the solution for a new related problem (like for a new related task in representation learning application or hyperparameter optimization application in [R1]), we can use the robust minmax solution -- we select the robust solution for the shared UL variable $x$ (the representation network or the  hyperparameter configuration). With a Pareto front, it is not clear which solution to pick for a new task since we would have a set of solutions, without the knowledge of which one would be useful for a new task/objective.
> > >
> > > Furthermore, regarding the point that "the Pareto frontier [...] cannot offer a worse result than robust optimization", please kindly note that, while a solution on the Pareto frontier implies that there is no other solution that "dominates" it, to the best of our knowledge, *there is no guarantee that some solution on the **obtained Pareto frontier** achieves the optimal value for the robust minmax objective $\max_i \min_x f_i(x)$* **unless** the Pareto frontier is completely dense, which is never the case. Multi-objective optimizers can return a set of solutions on the Pareto frontier, but even uniformly covering the Pareto frontier requires the size of the solution set to grow exponentially in the number of objectives $n$. The validity of the claim that "the Pareto frontier [...] cannot offer a worse result than robust optimization" relies heavily on the density of the Pareto frontier obtained by the optimizer, and hence is not generally true.
> > >
> > > Finally, for nonconvex objective functions, the Pareto frontier refers to the Pareto stationarity rather than Pareto optimality. Our considered first-order stationarity condition is defined on the weighted average of the objective value, while the classical Pareto stationarity (please see equation (2) in the very recent paper [R6] and references therein) is measured on the size of the weighted average of the gradients. The weighting vector in both of these two notations is optimized over a simplex. Therefore, the stationarity condition of our proposed min-max formulation can be considered as one variant of Pareto stationarity for nonconvex problems.
> > >
> > > [R2] Wilson, et al. "Computing possibly optimal solutions for multi-objective constraint optimisation with tradeoffs." International Joint Conference on Artificial Intelligence. 2015.
> > >
> > > [R3] Yang, et al. "A generalized algorithm for multi-objective reinforcement learning and policy adaptation." Neural Information Processing Systems. 2019.
> > >
> > > [R4] Mehta, et al. "Minimax multi-task learning and a generalized loss-compositional paradigm for MTL." Neural Information Processing Systems. 2012.
> > >
> > > [R5] Collins, et al. "Task-robust model-agnostic meta-learning." Neural Information Processing Systems (NeurIPS). 2020.
> > >
> > > [R6] H. Fernnado, et al, "Mitigating gradient bias in multi-objective learning: A provably convergent stochastic approach." 2022.

---

> > > ### Author Response · Authors · 2022-12-01
> > > **Comments on reviewer's XgUo's response (Thread 3/3)**
> > >
> > > > How to tune $\alpha, \beta$  and  $\gamma$ is critical. I strongly suggest adding a separate section to explain this in the revision. Also, it is worth explaining how to tune $\alpha$ to guarantee that $b_k^2 \leq \alpha$.
> > >
> > > We agree that the (initial) learning rates are critical, and hence, we tune it. At this point of the discussion, we are unable to submit a revision. We will update the future revision as per the suggestion (when we are able to make revisions).
> > >
> > > Please kindly note that learning rate tuning is very common for solving both general optimization and machine learning problems. The orders of shrinking the step sizes have been given in our theory. What we need to tune are only the initial step sizes. Note that the LL problem is strongly convex, the maximization at the UL is concave, and the minimization at the UL is nonconvex, so in general, we select the initial rate of $\gamma$ as the largest one, $\beta$ as the medium, and $\alpha$ as the smallest since the strongly convex LL problems is relatively easy, then convex, and finally nonconvex.
> > >
> > > Regarding the $b_k^2 \leq \alpha$ condition, if the implicit gradient is computed exactly, the condition $b_k^2 \leq \alpha$ is *trivially satisfied* since $b_k = 0$. When we are using the (standard in stochastic bilevel optimization literature) sampling-based Hessian inverse approximation utilizing the Neumann series, please note that $\alpha = \mathcal O(K^{-3/5})$ while $b^2_k=\mathcal{O}((1-\mu_g/L_g)^t)$ utilizing $t$ samples (please see Lemma 3.2. in [R7] or Lemma 1 in [R8]), where $\mu_g$ and $L_g$ represent the strong convexity parameter and gradient Lipschitz continuity constant respectively. Therefore, with $t \geq \mathcal{O}(\log K)$ we have $b_k^2 \leq \alpha$.
> > >
> > > [R2] Ghadimi, et al. Approximation Methods for Bilevel Programming, 2018.
> > >
> > > [R3] Hong, et al. A Two-Timescale Stochastic Algorithm Framework for Bilevel Optimization: Complexity Analysis and Application to Actor-Critic, 2020.
> > >
> > > > Besides, I disagree that weakly convexity implies non-convexity. The former mostly refers to convexity and is much narrower than the latter.
> > >
> > > We apologize for the miscommunication: we meant that weak convexity captures a class of non-convex problems. Weakly convex functions are not convex -- note difference in the following definitions (also in Appendix A.1, Assumptions 1 and 2). For any convex function $\tau$, there exists a $\mu \geq 0$ such that, for any $x, x'$ ($x \not= x'$)
> > >
> > > $$
> > > \tau(x') \geq \tau(x) + \langle \nabla_x \tau(x), x' - x \rangle + \mu \| x' - x \|^2,
> > > $$
> > >
> > > whereas, for a weakly-convex function $\kappa$, there exists $\nu > 0$ such that, for any $x, x'$ ($x \not= x'$)
> > >
> > > $$
> > > \kappa(x') \geq \kappa(x) + \langle \nabla_x \kappa(x), x' - x \rangle - \nu \| x' - x \|^2.
> > > $$
> > >
> > > Note the "$-\nu$" for a weakly-convex $\kappa$ instead of the "$+\mu$" for a convex $\tau$ in the third term on the right hand side of the above two inequalities. So $\kappa$ is clearly not convex. Moreover, note that the $\|x' - x\|^2$ term on the right-hand side of the inequality for the weakly-convex function is strictly positive, implying that, for large enough $\nu$, the inequality will be true for **any** function. We provide convergence results which depend on the coefficient of weak-convexity (for our UL function in question, it is the $\mu_\ell$), with slower rates for larger coefficients.
> > >
> > > **We hope that this addresses and clarifies the concerns raised by the reviewer. We again appreciate their feedback and their consideration.**

---

> > > ### Author Response · Authors · 2022-12-06
> > > **Checking in for further questions/clarifications**
> > >
> > > Hi Reviewer XgUo: we would like to check in with you to see if our response makes sense and addresses your concerns about the significance of the formulation, step size tuning, and our assumptions. Do you have any further questions or clarifications to make? If so, we would be happy to clarify further.
> > >
> > > Best, Authors

---

### Decision · Program_Chairs · 2023-01-20

**Decision:**

Accept: poster

**Justification For Why Not Higher Score:**

I did not see enough enthusiasm from the reviewers about the technical strength/novelty.

**Justification For Why Not Lower Score:**

I think it is worthwhile to publish papers with good conceptual contributions (instead of just looking for delta improvements to well established problems).

**Metareview: Summary, Strengths And Weaknesses:**

The paper considers robust, multi-objective min-max bilevel optimization problems. Such problems are motivated by applications such as multi-agent reinforcement learning and hyperparameter optimization. Their study with multiple objectives is initiated by this work. The specific assumptions about the upper level (UL) and lower level (LL) problems involve weak convexity for UL and strong convexity for LL. For this setting, the paper introduces an algorithm based on (projected) SGD applied to the two level problems. The paper proves that the proposed algorithm converges as $\tilde{O}(\sqrt{n}K^{-2.5}),$ where $n$ is the number of objectives in the formulation and $K$ is the number of iterations. The paper further illustrates how the proposed techniques can be used to address problems in robust representation learning and robust hyperparameter optimization.

The paper is written well and was judged technically sound, with non-trivial technical results. However, its relationship to TTSA (Hong et al, 2020) was not completely clear and should be discussed more in a revised version. The AC and the reviewers appreciated the originality and novelty of the overall approach, and hope that this paper will stimulate more research on related problems.

**Note From Pc:**

if the above contains the word "oral" or "spotlight" please see: "oral" presentation means -> notable-top-5% and "spotlight" means -> notable-top-25%. As stated in our emails, we are disassociating presentation type from AC recommendations

**Summary Of Ac-Reviewer Meeting:**

Two of the reviewers (UZe1 and fwCP) showed up to the online meeting and the third reviewer (XgUo) followed up over email. This is the summary of the XgUo's position:

"--I have read the other reviewers' reviews and the authors' responses. They do not really change my view of this work.
Basically, the authors formulated a specific problem (combination of robust optimization + MOO + BL) and solved it with a method similar to TTSA (Hong et al, 2020).
--I am not convinced that the proposed robust MOO bilevel formulation helps solve real problems like hyperparameter optimization.
--But I agree that the authors have justified the proposed method (to address this specific problem) theoretically well (though I did not check all the proof details).

Feel free to make a decision as per your understanding and consensus on this work (i.e., if all of you align on the final decision, I would not challenge it) but I would like to keep the same score considering all my concerns so far for your reference."

In the discussion with the other two reviewers, it did not seem like they were too enthusiastic about the technical contributions and could understand well the relationship to the TTSA algorithm; however, the reviewers did think that the novel problem formulation in the paper was interesting and could conceivably lead to more work in this area. They shared that their 'real' score would be closer to 7; however there was no such option for the reviewers to enter and they felt an 8 was too high. Hence, my inclination was to suggest the acceptance, even if the technical results are not very strong, due to the originality of the approach and potential for leading to more work in the area. I also read through the discussion and found the author responses reasonable and well grounded.